# Feedback between a retinoid-related nuclear receptor and the *let-7* microRNAs controls the pace and number of molting cycles in *C. elegans*

Ruhi Patel[1†], Himani Galagali[2†], John K Kim[2*], Alison R Frand[1*]

[1]Department of Biological Chemistry, David Geffen School of Medicine, University of California, Los Angeles, Los Angeles, United States; [2]Department of Biology, Johns Hopkins University, Baltimore, United States

**Abstract** Animal development requires coordination among cyclic processes, sequential cell fate specifications, and once-a-lifetime morphogenic events, but the underlying timing mechanisms are not well understood. *Caenorhabditis elegans* undergoes four molts at regular 8 to 10 hour intervals. The pace of the cycle is governed by PERIOD/*lin-42* and other as-yet unknown factors. Cessation of the cycle in young adults is controlled by the *let-7* family of microRNAs and downstream transcription factors in the heterochronic pathway. Here, we characterize a negative feedback loop between NHR-23, the worm homolog of mammalian retinoid-related orphan receptors (RORs), and the *let-7* family of microRNAs that regulates both the frequency and finite number of molts. The molting cycle is decelerated in *nhr-23* knockdowns and accelerated in *let-7(−)* mutants, but timed similarly in *let-7(−) nhr-23(−)* double mutants and wild-type animals. NHR-23 binds response elements (ROREs) in the *let-7* promoter and activates transcription. In turn, *let-7* dampens *nhr-23* expression across development via a complementary *let-7*-binding site (LCS) in the *nhr-23* 3′ UTR. The molecular interactions between NHR-23 and *let-7* hold true for other *let-7* family microRNAs. Either derepression of *nhr-23* transcripts by LCS deletion or high gene dosage of *nhr-23* leads to protracted behavioral quiescence and extra molts in adults. NHR-23 and *let-7* also coregulate scores of genes required for execution of the molts, including *lin-42*. In addition, ROREs and LCSs isolated from mammalian *ROR* and *let-7* genes function in *C. elegans*, suggesting conservation of this feedback mechanism. We propose that this feedback loop unites the molting timer and the heterochronic gene regulatory network, possibly by functioning as a cycle counter.

**\*For correspondence:**
jnkim@jhu.edu (JKK);
afrand@mednet.ucla.edu (ARF)

[†]These authors contributed equally to this work

**Competing interest:** The authors declare that no competing interests exist.

## Editor's evaluation

The manuscript nicely advances our understanding of the roles of heterochronic genes and the NHR-23 nuclear receptor transcription factor in the regulation of the temporal dynamics of molting behavior in *C. elegans* larval development. The data reveals direct regulatory feedback between let-7 family microRNAs and nhr-23, and shows that this circuit contributes to the regulation of developmental pace. The findings should be of interest to the field studying heterochronic genes and microRNAs in developmental timing, and to the broader field of chronobiology, particularly the regulation of complex oscillatory gene regulatory networks.

## Introduction

Timekeeping is a critical component of animal development. Developmental timers are a subset of biological clocks that govern the frequency of cyclic processes such as the formation of vertebrate somites and insect body segments (*Diaz-Cuadros et al., 2021*; *Keyte and Smith, 2014*; *Uriu, 2016*). Heterochronic gene pathways, like the microRNA–mRNA networks in *Caenorhabditis elegans* and other organisms, regulate sequential events and orchestrate the timing of development across tissue types (*Ambros and Ruvkun, 2018*; *Galagali and Kim, 2020*; *Holguera and Desplan, 2018*). The mechanisms by which developmental clocks and heterochronic pathways interact to coordinate repeated developmental processes with cell fate transitions remain unknown.

Studies of circadian rhythm have provided a framework for understanding how biological clocks schedule rhythmic processes. The circadian clock is a physiologic clock that governs daily feeding–fasting and sleep–wake cycles, as well as diurnal, organ-specific metabolic cycles. Entrainment of the circadian clock coordinates the cellular and molecular processes with predictable 24 hour (h) changes in the environment (*Takahashi, 2016*). The period of developmental clocks, unlike physiologic clocks, may vary in response to external conditions, such as temperature, nutrition, and growth factors. Many developmental clocks that regulate morphogenic processes also stop after a finite number of iterations (*Konopka and Benzer, 1971*; *Rensing et al., 2001*; *Tsiairis and Großhans, 2021*).

The mechanistic basis for both developmental and physiological clocks are molecular-genetic oscillators. Cyclic expression of the core components of oscillators and their target genes together underlie biological rhythms. Molecular-genetic oscillators are comprised of interconnected feedback loops among the core components. Experimental and theoretical studies indicate that negative feedback loops with intrinsic time delays or interdependent positive and negative feedback loops with intrinsic time delays set up most self-sustaining genetic oscillators (*Johnson and Day, 2000*; *Novák and Tyson, 2008*; *Tsiairis and Großhans, 2021*). In both cases, time delays are caused in part by unequal rates of RNA versus protein synthesis and degradation. For example, during somitogenesis, the Hes7 transcription factor represses its own transcription, producing a self-sustaining oscillator (*Bessho et al., 2003*).

The key components of the circadian clock in mammals also consist of transcriptional activators and repressors interacting through interlocked feedback loops. During the day, CLOCK and BMAL1 activate the transcription of *PERIOD/PER* and other genes. During the night, PER proteins interact with CLOCK and BMAL1 and repress their own transcription (*Partch et al., 2014*; *Takahashi, 2016*; *Takahashi, 2017*). The short half-life of the PER protein, in combination with the continued transcriptional repression of *PER*, results in decrease of PER proteins late in the night. The decrease in the levels of PER is accompanied by increase in the levels of CLOCK and BMAL1 early in the morning. CLOCK and BMAL1 also activate transcription of *REV-ERBα* and *REV-ERBβ*. The competition between the transcriptional repressors, REV-ERBs, and the transcriptional activators, the Retinoid-related Orphan Receptors (RORs), for the same binding sites in the *BMAL1* promoter regulates rhythmic expression of *BMAL1* in peripheral organs and the central nervous system (*Cook et al., 2015*; *Zhang et al., 2017*).

The components of the circadian clock are also subject to post-transcriptional and post-translational regulation. The *bantam* microRNAs regulate the temporal expression of *Drosophila clock* by directly binding the *clock* 3' UTR and repressing translation (*Kadener et al., 2009*). In mice, the microRNAs miR-24 and miR-30 regulate stability of *Per2* mRNAs and repress their translation by interacting with the *Per2* 3' UTR (*Yoo et al., 2017*). A few other microRNAs regulate the expression of core clock components. However, the prevalence of microRNA-mediated post-transcriptional feedback loops among developmental clocks is not yet known (*Alvarez-Saavedra et al., 2011*; *Chen et al., 2013*; *Du et al., 2014*).

Molting in *C. elegans* is a reiterated and periodic developmental process. Under favorable conditions, *C. elegans* develop through four larval stages, L1 to L4. Larvae molt from one stage to the next at regular 8- to 10-h intervals and then emerge as adults. *C. elegans* enter and exit a state of behavioral quiescence, termed lethargus, during each molt (*Figure 1A*). Across lethargus, epithelia detach from the old cuticle and synthesize the new larger cuticle for the upcoming life stage. The animal then escapes from the old cuticle at ecdysis. Newly emerged larvae forage and feed during the intermolt. We previously identified PER/LIN-42 as a key component of the underlying molting cycle timer (*Monsalve et al., 2011*).

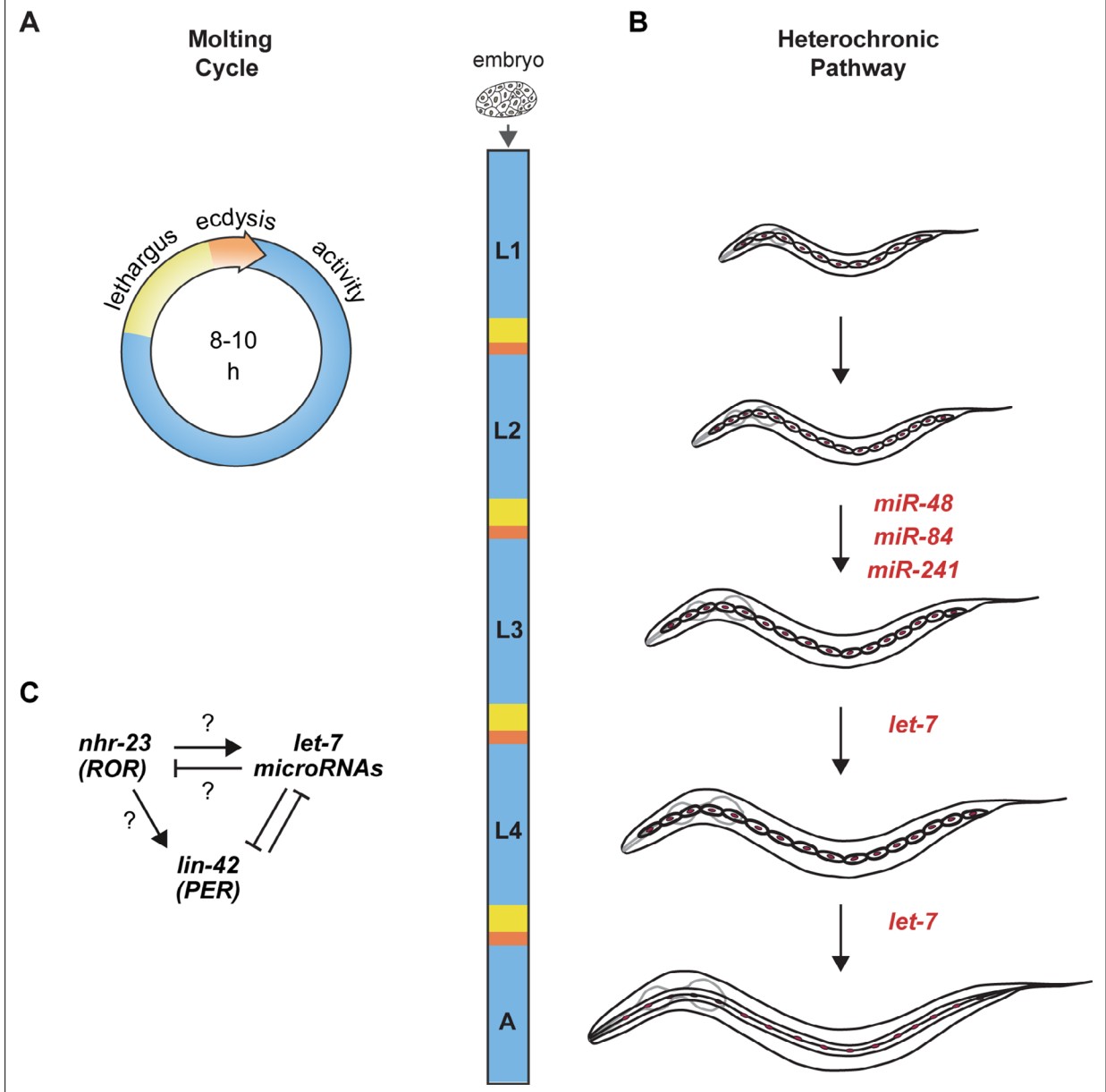

**Figure 1.** Animal development requires coordination between reiterated processes and successive transitions in cell fate. (**A**) Stages of the molting cycle of *C. elegans*, emphasizing the regular intervals of lethargus (yellow), ecdysis (orange), and physical activity (blue). Upon hatching, the embryo grows and develops through four larval stages that are punctuated by molts. (**B**) Successive transitions in the fate of the lateral epidermal stem cells, called seam cells (red nuclei), in developing worms. The *let-7* family of microRNAs, consisting of miR-48, miR-84, miR-241, and *let-7*, promote transitions in the fate of seam cells; miR-48, miR-84, and miR-241 are specific to the L2 stage, while *let-7* is associated with the L3 and L4 stages. The adult stage is characterized by the presence of cuticular structures called alae. (**C**) Schematic depiction of interactions between ROR/*nhr-23*, *let-7* microRNAs, and PER/*lin-42* in *C. elegans*. Arrowheads and bars signify positive and negative regulation, respectively. Question marks signify regulatory events evaluated in this study.

More recent work has identified more than 3700 *C. elegans* genes with oscillatory expression across larval development, including 257 linked to specific aspects of molting (*Hendriks et al., 2014*; *Kim et al., 2013*; *Meeuse et al., 2020*). Transcript levels of these genes oscillate with the same frequency as the molting cycle, and the waveforms have a phase-locked peak once per larval stage, that is, the genes peak at the same relative timepoint within each larval stage. Computational models suggest that a single genetic oscillator governs the reiterative expression of all 3700 genes (*Meeuse*

*et al., 2020*; *Tsiairis and Großhans, 2021*). It is not known whether the PER-based molting cycle timer and this theoretical genetic oscillator represent the same timekeeping mechanism.

The heterochronic gene pathway regulates the timing of unidirectional cell fate transitions during the development of *C. elegans*. Key heterochronic genes include the conserved *let-7* microRNA, its paralogs, and stage-specific targets of the *let-7* family (*Abbott et al., 2005*; *Ambros and Ruvkun, 2018*; *Reinhart et al., 2000*; *Figure 1B*). Each larval stage is marked by stereotypic divisions of the lateral epidermal stem cells, called seam cells. The *let-7* paralogs *mir-48*, *mir-84*, and *mir-241* specify the L2 fate of the seam cells, wherein the seam cells undergo one symmetric and one asymmetric division (*Abbott et al., 2005*). The *let-7* microRNA specifies later L3 and L4 fates, which include homotypic fusion of the seam cells into lateral syncytia and secretion of three long ridges on the adult-stage cuticle called alae (*Ambros, 1989*; *Reinhart et al., 2000*; *Vadla et al., 2012*).

NHR-23, the only *C. elegans* homolog of mammalian ROR transcription factors (*Antebi, 2015*), is expressed in the larval epidermis during each larval stage (*Frand et al., 2005*; *Kostrouchova et al., 1998*). Predicted targets of NHR-23 are enriched for genes associated with molting, including cuticle collagens and enzymes necessary for synthesis and degradation of the cuticle (*Kouns et al., 2011*). LIN-42, the *C. elegans* homolog of the core circadian clock protein and tumor suppressor PERIOD (*Jeon et al., 1999*), sustains the 8 h intervals between molts (*Monsalve et al., 2011*). LIN-42 and the *let-7* family mutually inhibit one another (*Figure 1C*; *McCulloch and Rougvie, 2014*; *Perales et al., 2014*; *Reinhart et al., 2000*; *Van Wynsberghe et al., 2014*). Moreover, homologs of genes involved in the maintenance of circadian rhythm in *Drosophila* interact genetically with *let-7* and regulate the L4-to-adult transition in *C. elegans* (*Banerjee et al., 2005*).

Further evidence of crosstalk between the molting cycle timer and the heterochronic pathway comes from the observation that the levels of primary *let-7* family transcripts cycle in phase with the molts (*McCulloch and Rougvie, 2014*; *Van Wynsberghe et al., 2011*). However, the transcriptional activator(s) responsible for the oscillatory expression of *let-7s* remains unknown.

Here, we show that both NHR-23 and the *let-7* family of microRNAs (the *let-7s*) are key components of a simple regulatory circuit that operates within the molecular-genetic oscillator underlying the molting cycle and also within the heterochronic gene regulatory network. Using longitudinal studies of the biorhythm of molting in relevant genetic backgrounds, molecular and cell biological analyses, and bioinformatic approaches, we show that NHR-23 transcriptionally activates the *let-7s* and, in turn, the *let-7s* post-transcriptionally repress *nhr-23* mRNA. In addition, NHR-23 positively autoregulates its own transcription. Together, NHR-23/ROR and the *let-7s* establish a transcriptional–post-transcriptional feedback loop that governs the pace and extinction of the molting cycle after four iterations. As both the key components and *cis*-regulatory elements comprising this feedback loop are conserved from nematodes to mammals, our findings may apply to some developmental and tissue-specific circadian clocks of humans, and help elucidate related pathologies, including birth defects, cancers, sleep disorders, and metabolic syndromes (*Oyama et al., 2017*; *Patke et al., 2017*; *Puram et al., 2016*; *Roenneberg and Merrow, 2016*).

## Results

### Larval molting cycles lengthen in *nhr-23* knockdowns and shorten in *let-7* family mutants

To determine the role of *nhr-23* and *let-7* in timing the molting cycle, we measured and compared the length of molting cycles in *nhr-23* knockdowns, *let-7* mutants, and control larvae through a series of longitudinal studies. Each experiment captured one iteration of the molting cycle. The full set captured emergence of L2s, L3s, L4s, and young adults. In each experiment, we measured (1) the interval of physical activity in the target stage (defined as the time elapsed between successive episodes of lethargus); (2) the interval of lethargus associated with the molt; and (3) the wake-to-wake interval (defined as the time elapsed between two sequential transitions from lethargus to activity) (*Figure 2A*).

Feeding L1 stage hatchlings bacteria that express dsRNAs complementary to *nhr-23* (*nhr-23(RNAi)*) usually leads to severe molting defects and larval arrest in the L2 stage. To circumvent L2 arrest and determine how knockdown of *nhr-23* affects the timing of the L3 and L4 stages, we maintained worms, starting at the L1 stage on control bacteria for 6 h and 14 h, respectively, and then moved them to

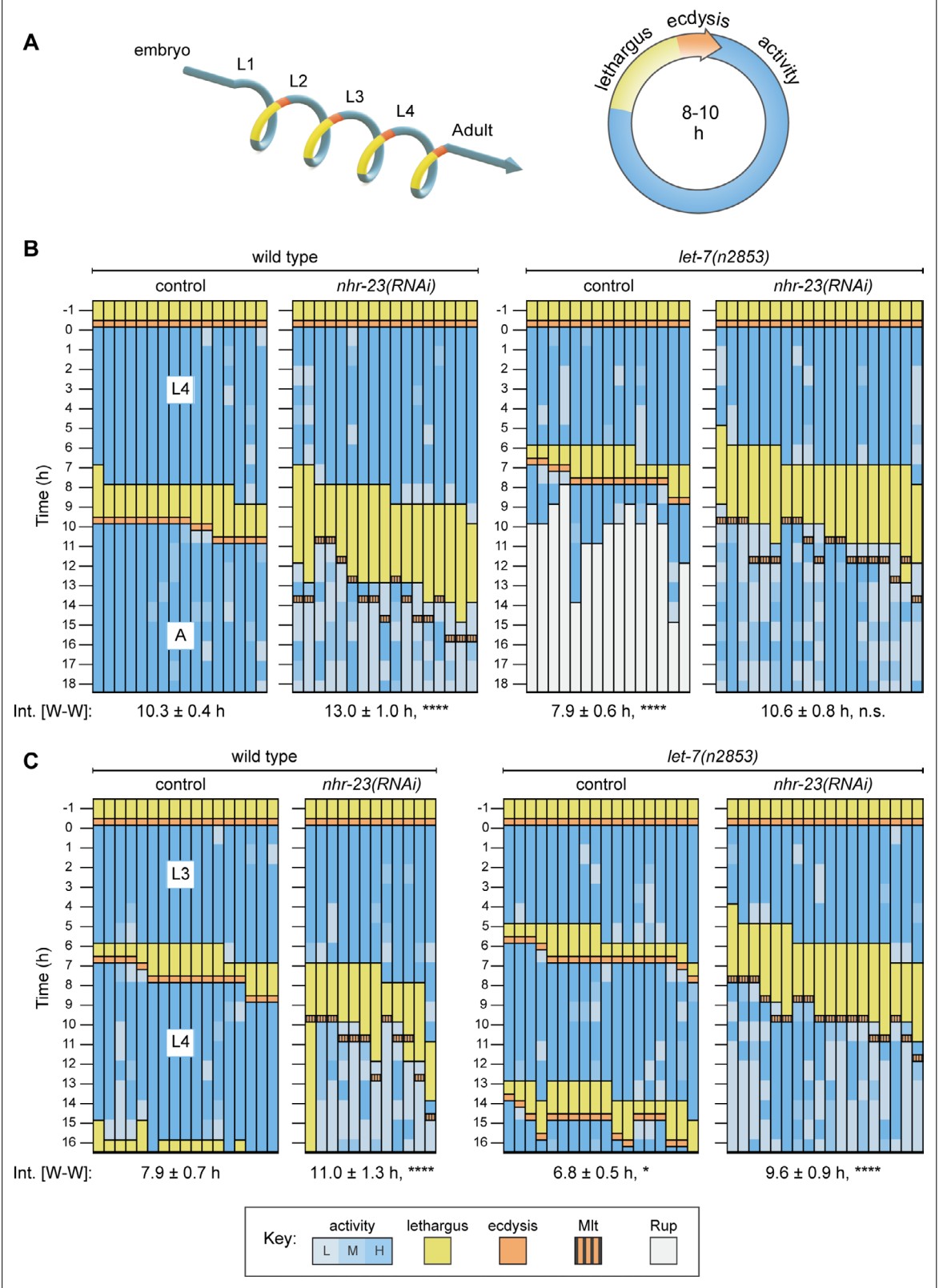

**Figure 2.** Opposite and codependent effects of *nhr-23* and *let-7* on molting biorhythms. (**A**) Stages of the life cycle (left) and the molting cycle (right) of *C. elegans*. (**B**) Actograms depict the behavior, life stage, and phenotype of worms observed at 1h intervals from emergence in L4 onward. Each chart combines records from two independent trials. The records within each column correspond to a single worm. In addition, the molting-defective phenotype is labeled only when first detected. See key at the bottom of *Figure 2* for details; activity is designated as low (L), medium (M), or high (H);

*Figure 2 continued on next page*

Figure 2 continued

Mlt and Rup indicate molting and rupture phenotypes, respectively. The wake-to-wake time intervals (W–W) are indicated. Relevant scoring rubrics are further defined in Results and Materials and methods. ****p ≤ 0.0001; ordinary one-way analysis of variance (ANOVA) with Bonferroni's correction for multiple comparisons. (**C**) Same as B, except that the actograms depict animals observed from emergence in L3 onward. ****p ≤ 0.0001, *p ≤ 0.05; ordinary one-way ANOVA with Bonferroni's correction for multiple comparisons. *Supplementary file 1* includes the active, lethargic, and W–W intervals of these eight cohorts, as well as additional cohorts of both *let*-7 family mutants and *nhr-23* knockdowns.

*nhr-23(RNAi)* bacteria. This strategy ensured that all test subjects emerged in the target stage superficially normal, but none of the test subjects fully shed the cuticle from the ensuing molt, signifying complete penetrance of the molting-defective (Mlt) phenotype associated with *nhr-23(RNAi)*. Age-matched, wild-type larvae fed the same bacterial strain transformed with an empty vector served as controls.

The actograms in *Figure 2* display the results of these longitudinal studies. Each actogram corresponds to an isogenic cohort of animals. Therein, each column represents a single animal that emerged in the target stage (L4 in *Figure 2B* and L3 in *Figure 2C*), developed, and underwent the ensuing molt. Each animal was isolated during the preceding molt to achieve stringent synchronization at the outset. After it emerged, the worm was observed for approximately 1 min at regular 1 h intervals. At each timepoint, the worm was 'active' if both pharyngeal muscle contractions (pumps) and sinusoidal locomotion were observed. Conversely, the worm was 'lethargic' if neither pharyngeal pumps nor sinusoidal locomotion were observed, and its body posture resembled a hockey stick (*Iwanir et al., 2013*; *Raizen et al., 2008*). Separation of the preexisting cuticle from the body and detection of the shed cuticle on the culture plate signified the commencement and completion of ecdysis, respectively (*Singh and Sulston, 1978*).

As expected, the cohort of wild-type (control) animals first emerged as L4s, then entered and exited lethargus, shed the larval cuticle (ecdysed), and emerged as young adults almost synchronously (*Figure 2B* and *Supplementary file 1*). Approximately 50% (8 out 17 animals) of the cohort of *nhr-23(RNAi)* animals entered lethargus 9–10 h after emerging as L4s as compared with 19% of the control cohort (3 out of 16 animals). Strikingly, the cohort of *nhr-23(RNAi)* animals remained lethargic for twice as long as the control cohort. All of the *nhr-23(RNAi)* animals began to pump and locomote once again, but oftentimes at lower rates than wild-type adults. In principle, this intermittent sluggishness might result from incomplete arousal or hindrance by unshed parts of the L4-stage cuticle. Regardless, the wake-to-wake interval of the L4-stage *nhr-23(RNAi)* cohort was 13 ± 1.1 h as compared with 10.3 ± 0.4 h for the control cohort (p ≤ 0.0001). Following this trend, the L3-stage cohort of *nhr-23(RNAi)* larvae entered lethargus 1.4 ± 0.8 h later and remained in lethargus twice as long as the age-matched wild-type cohort (*Figure 2C*). Similarly, the cohort of *nhr-23(RNAi)* larvae molting from L2 to L3 were in lethargus three times longer than the age-matched controls (*Supplementary file 1*). Thus, *nhr-23(RNAi)* animals developing through three larval stages entered lethargus slightly later and remained lethargic for much longer than wild-type animals.

To evaluate the role of the *let-7s*, we tracked cohorts of *let-7(n2853)*, *let-7(mg279)*, and *let-7(mg279); mir-84(tm1304)* double mutants across late larval stages. Both *n2853* and *mg279* are associated with lower levels of mature *let-7*, relative to wild-type animals. However, *n2853* is a substitution in the seed sequence, whereas *mg279* is a 27-bp deletion upstream of the mature microRNA (*Bracht et al., 2004*; *Reinhart et al., 2000*). The null allele of *mir-84* enhances relevant phenotypes associated with *let-7(mg279)* (*Hayes and Ruvkun, 2006*). We also tracked *mir-48 mir-241(nDf51); mir-84(n4037)* triple mutants across L2, when the corresponding microRNAs are expressed but mature *let-7* is not yet detected (*McCulloch and Rougvie, 2014*). In contrast to animals subjected to *nhr-23(RNAi)*, *let-7(n2853)* mutants both entered and exited lethargus more quickly than wild-type animals. For example, the wake-to-wake interval for the *let-7(n2853)* cohort developing from L4s into adults was only 7.9 ± 0.6 h, an acceleration of 2.9 ± 0.7 h relative to the wild-type cohort (*Figure 2B*). All of the *let-7(n2853)* animals subsequently ruptured at the vulva, a hallmark of this strong loss-of-function allele (*Reinhart et al., 2000*; *Ecsedi et al., 2015*). In complementary studies, L4-stage cohorts of both *let-7(mg279)* single and *let-7(mg279); mir-84(tm1304)* double mutants also entered lethargus ahead of wild-type L4s (*Supplementary file 1*). Moreover, the cohort of *let-7(n2853)* mutants observed from emergence in L3 onward passed through two consecutive lethargic phases and emerged as young adults ahead of the entire wild-type cohort (*Figure 2C*). As such, repetition of the L3 stage, a retarded heterochronic

phenotype, cannot explain the acceleration of the L4 stage observed in *let-7(n2853)* mutants, because both the L3 and L4 stages of the mutants were shorter than those of wild-type larvae. Thus, lethargus was advanced and larval development was accelerated in three distinct mutants of the *let-7* family.

The altered pace of molting exhibited by either *nhr-23(RNAi)* or the *let-7(n2853)* mutant was partially suppressed in the *nhr-23(RNAi) let-7(n2853)* double mutant (*Figure 2B, C* and *Supplementary file 1*). Strikingly, none of the *let-7(n2853)* mutants ruptured on *nhr-23(RNAi)*, suggesting that the *let-7*-mediated suppression of *nhr-23* regulates both lethargus and the morphogenesis of the vulva (*Figure 2B*). Approximately 70% of the L4-stage cohort of *nhr-23(RNAi) let-7(n2853)* double mutants entered lethargus 7–8h after the L3-to-L4 molt, while the rest entered lethargus 5–6 h after the L3-to-L4 molt. In contrast, only a third of the L4-stage *let-7(n2853)* mutants entered lethargus 7 h after the L3-to-L4 molt; the rest did so after only 6 h. Thus, a majority of the *nhr-23(RNAi) let-7(n2853)* double mutants became lethargic 1 h after the majority of the *let-7(n2853)* single mutants. The double mutants also emerged from lethargus earlier than *nhr-23(RNAi)* single mutants (p < 0.01, ordinary one-way analysis of variance [ANOVA] with Bonferroni's correction for multiple comparisons). As a result, the wake-to-wake interval of the L4-stage cohort of *nhr-23(RNAi) let-7(n2853)* double mutants was 10.6 ± 0.8 h, similar to the value of the wild-type cohort (p ≥ 0.9). Notably, *nhr-23(RNAi) let-7(n2853)* double mutants underwent aberrant ecdysis, indicating that the role of *nhr-23* in lethargus and ecdysis is genetically separable.

Suppression of the altered pace of molting in the *nhr-23(RNAi) let-7(n2853)* double mutants was also apparent during the L2 and L3 stages. The wake-to-wake interval of the *nhr-23(RNAi) let-7(n2853)* double mutants during the L3 stage was 1.4 ± 1.4 h shorter than *nhr-23(RNAi)* alone (p = 0.0002). Moreover, the triple knockout of the *let-7* sisters, *mir-48 mir-241(nDf51); mir-84(n4037)*, partially suppressed the prolonged lethargy associated with *nhr-23(RNAi)* across the L2/L3 molt, shortening the lethargic interval by 0.9 ± 1.0 h (p = 0.002, *Supplementary file 1*) to that of wild-type animals.

Taken together, the mutual suppression of the behavioral phenotypes of *nhr-23(RNAi)* or the *let-7(n2853)* mutant in the *nhr-23(RNAi) let-7(n2853)* double mutant suggests that NHR-23 and *let-7s* act in the same genetic pathway. Based on these results, we propose a model whereby NHR-23 accelerates the molting cycle in part by regulating the expression of the *let-7* family of microRNAs and the *let-7s* decelerate the cycle in part by regulating the expression of *nhr-23*.

## NHR-23 promotes oscillatory expression of primary *let-7* and its paralogs

RNA polymerase II transcribes microRNA genes to generate primary microRNAs (pri-miRNAs). The microprocessor complex, consisting of Drosha and Pasha, processes pri-miRNAs into precursor miRNAs (pre-miRNAs). Pre-microRNAs are exported into the cytoplasm and processed further by Dicer. The resulting microRNA duplex is unwound and the mature microRNA strand is loaded into the effector Argonaute protein. Cofactors can regulate the biogenesis of microRNAs at each of these levels (*Ambros and Ruvkun, 2018*; *Galagali and Kim, 2020*).

Based on the longitudinal studies described above, we hypothesized that the transcription factor NHR-23 may directly regulate transcription of *let-7*. Consistent with this hypothesis, a binding peak for NHR-23 was reported within ~300 bp upstream of primary *let-7* from the modENCODE Consortium (*Figure 3A*; *Celniker et al., 2009*; *Gerstein et al., 2010*). Nuclear hormone receptors typically bind DNA response elements as homotypic or heterotypic dimers (*Evans and Mangelsdorf, 2014*). NHR-23 and its mammalian counterpart RORα are among the few that bind the consensus sequence 5′-(A/G) GGTCA-3′ as monomers to activate transcription of target genes (*Giguère et al., 1994*; *Kouns et al., 2011*). We identified three occurrences of this sequence, called the ROR response element (RORE), within the reported NHR-23-binding peak (*Figure 3A*). Additionally, the 300 bp region containing the ROREs is contained within a previously characterized enhancer element required for *let-7* transcription (*Johnson et al., 2003*; *Kai et al., 2013*).

To validate NHR-23-binding upstream of primary *let-7* during L3 and L4, we appended the coding sequence for a 3xFLAG affinity tag to the endogenous *nhr-23* gene using the CRISPR-Cas9 system (*Paix et al., 2015*) and performed chromatin immunoprecipitation with anti-FLAG antibody followed by gene-specific, quantitative polymerase chain reactions (ChIP-qPCR). The signal flanking RORE3 was enriched 4-fold during the L3 stage and 21-fold during the L4 stage in *nhr-23::3xflag* samples as compared with wild-type (N2; no tag) samples. In contrast, signal from the promoter of *col-19*, which

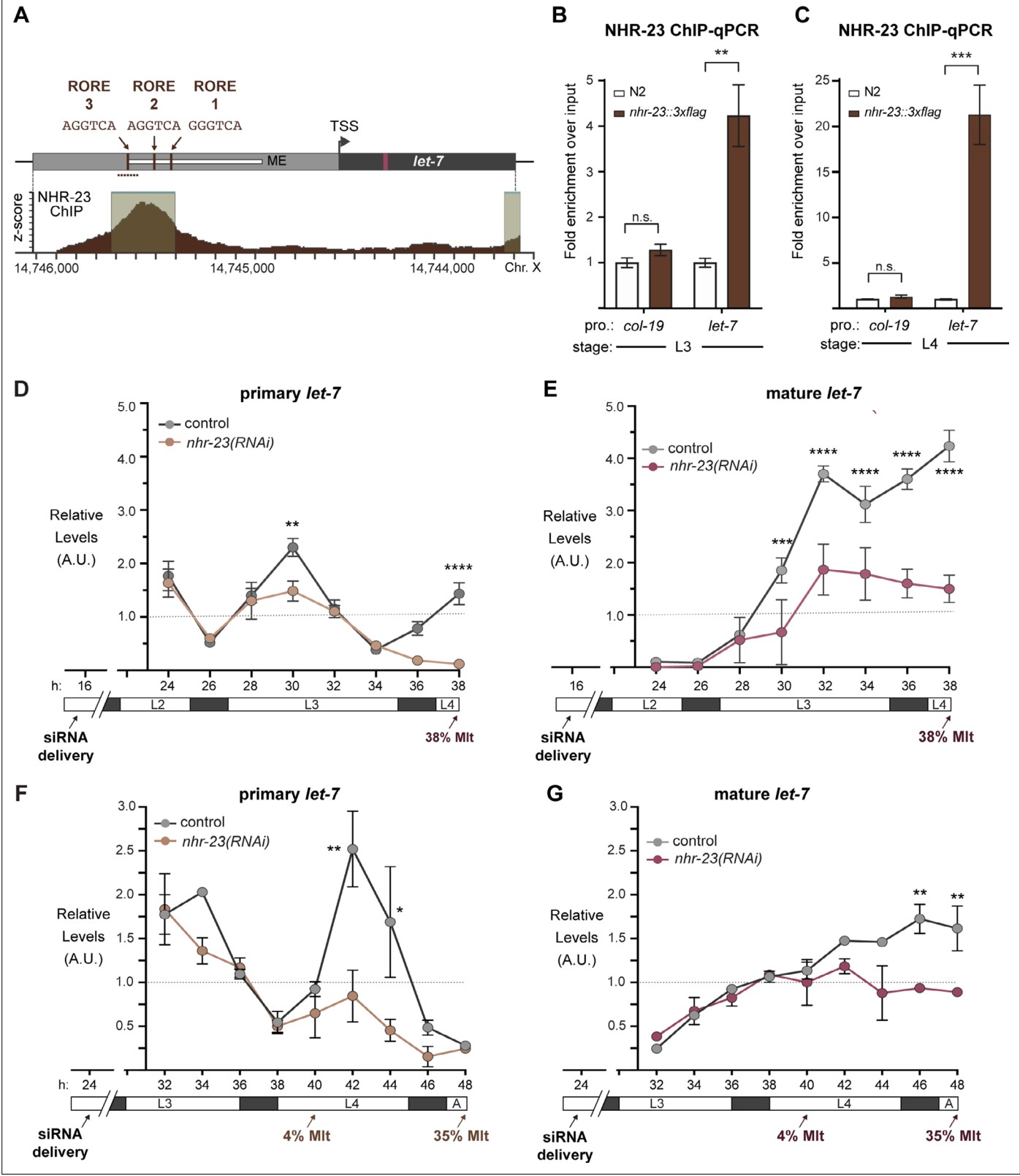

**Figure 3.** NHR-23 promotes transcription of primary *let-7*. (**A**) Schematic of the *let-7* locus in *C. elegans* (top) and corresponding NHR-23 ChIP-seq peaks (bottom). Top: Mature *let-7* (magenta); pri-*let-7* (black); upstream DNA sequences (dark gray) present in the *let-7p::gfp* transcriptional reporter (*Kai et al., 2013*) the minimal seam-specific enhancer (light gray) (MP) (*Johnson et al., 2003*); consensus ROREs (brown). TSS indicates the major transcriptional start site for pri-*let-7* (*Kai et al., 2013*). Dotted line represents the amplicon quantified by ChIP-qPCR. Bottom: The NHR-23 ChIP-seq

*Figure 3 continued on next page*

*Figure 3 continued*

peaks reported by the modENCODE Consortium are indicated. (**B, C**) ChIP-qPCR analysis of NHR-23::3xFLAG enrichment at the *let-7* promoter in L3- and L4-stage wild-type (untagged control) and *nhr-23::3xflag* larvae. The promoter of *col-19*, which had no detectable NHR-23 ChIP-seq peaks, was used as a negative control. Values represent the mean ± standard error of the mean (SEM) of three independent trials, each of which included three technical replicates. Values for the amplicon of interest in QK159 [*nhr-23::3xflag*] and N2 (wild type) were first normalized to the respective input. The average fold enrichment in QK159 samples was then normalized to the average fold enrichment in N2 samples within each trial. **p ≤ 0.01, ***p ≤ 0.001, two-way analysis of variance (ANOVA) with Bonferroni's correction for multiple comparisons. (**D**) Levels of primary *let-7* transcript determined by TaqMan RT-qPCR in *nhr-23(RNAi)* and mock-treated larvae developing from the late L2 stage until the L3-to-L4 molt. Each value was normalized to *ama-1* transcript levels in the same sample. Values were then normalized to the average of all control time samples. Symbols represent the mean and range from two biological replicates. The *x*-axis indicates time elapsed (h) on food. The underlying bar depicts developmental stages; gray boxes therein signify lethargi. The times of initial exposure to *nhr-23* siRNAs and the appearance of molting-defective *nhr-23(RNAi)* larvae are indicated. ****p ≤ 0.0001, **p ≤ 0.01; two-way ANOVA with Bonferroni's correction for multiple comparisons. (**E**) Same as D, except the levels of mature *let-7* transcripts, normalized to levels of the snoRNA U18, are shown. ****p ≤ 0.0001, ***p ≤ 0.001, two-way ANOVA with Bonferroni's correction for multiple comparisons. (**F, G**) Same as D and E, respectively, except the larvae were collected from the early L3 stage until the L4-to-adult stage. **p ≤ 0.01, *p ≤ 0.05, two-way ANOVA with Bonferroni's correction for multiple comparisons.

The online version of this article includes the following figure supplement(s) for figure 3:

**Figure supplement 1.** NHR-23 interacts with ROREs upstream of *let-7* and its paralogs.

**Figure supplement 2.** NHR-23 regulates the expression of the *let-7* family during the L3 stage.

is not targeted by NHR-23, was not detectably enriched in either strain (**Figure 3B, C**). Together, the data show that NHR-23 binds one or more ROREs in the promoter of *let-7* during two sequential larval stages. Using the same combination of bioinformatic and biochemical approaches, we also found that NHR-23 occupies the promoters of three *let-7* sisters (*mir-48*, *mir-241*, and *mir-84*) in both L3 and L4 larvae (**Figure 3—figure supplement 1**; *Johnson et al., 2003*).

We next asked whether *nhr-23* regulates the temporally reiterated expression from the promoter of *let-7*. To address this question, we measured and compared the abundance of nuclear-localized GFP expressed from the *let-7* promoter (*Kai et al., 2013*) in stage-specific *nhr-23* knockdowns and age-matched control animals via quantitative fluorescence microscopy (**Figure 3—figure supplement 2A, B**). In preliminary studies, we tracked the cycling signal associated with this particular *let-7p::nls-gfp* fusion gene and detected peaks early in the third and fourth molts. Accordingly, nuclei in the lateral epidermis were imaged within the first hour of the L3/L4 and L4/adult molts. The signal intensity in hyp7 nuclei was 2.3 ± 1.3-fold (mean ± standard deviation [SD]) lower in *nhr-23(RNAi)* than control animals. Levels of GFP detected in seam nuclei were more variable during the L3-to-L4 molt than the L4-to-adult molt, possibly due to continuation of the cell cycle. Even so, the mean signal intensity in the seam was consistently lower in *nhr-23* knockdowns than control animals (**Figure 3—figure supplement 2A, B**).

To determine the extent to which *nhr-23* promotes the reiterated expression of endogenous *let-7*, we used TaqMan RT-qPCR to detect primary (pri-) *let-7* and mature *let-7* in successive samples of *nhr-23* knockdowns and mock-treated, wild-type animals developing from L2 to L4 or L3 to young adults (**Figure 3D–G**). Attenuation of the RNAi of *nhr-23* enabled the collection of hundreds of *nhr-23(RNAi)* animals late in larval development, as <40% of *nhr-23(RNAi)* animals exhibited molting defects by the endpoint. Under these conditions, peak levels of *nhr-23* transcripts were 4.1-fold lower in *nhr-23(RNAi)* than wild-type animals (data not shown). Transcript levels of pri-*let-7* in wild-type animals peaked in L3 and once again in L4 (**Figure 3D, F**). In contrast, pri-*let-7* levels in *nhr-23(RNAi)* animals were 1.5-fold lower at L3 (30 h) and 3-fold lower at L4 (42 h) than the peak value detected in age-matched, control larvae (**Figure 3D, F**). Levels of mature *let-7* stagnated in *nhr-23(RNAi)* animals but rose continuously in wild-type controls collected across the L3-to-L4 and larval-to-adult transitions (**Figure 3E, G**). In both L3 and L4 stages, molting-defective larvae were first observed as levels of *let-7* plateaued, consistent with the attribution of the phenotype to knockdown of *nhr-23*. The levels of the primary transcripts of the other members of the *let-7* family, miR-48, miR-84, and miR-241, were similarly reduced in *nhr-23(RNAi)* larvae developing across the L3 stage, as compared with age-matched control larvae (**Figure 3—figure supplement 2C, D**). Collectively, these findings strongly suggest that NHR-23 directly and repeatedly activates the transcription of the primary transcripts of the *let-7* family of microRNAs.

## Scrambling the ROREs reduces NHR-23 binding at *let-7* promoters and henocopies *let-7 loss-of-function* (*lf*) mutants

To test the physiological relevance of the three consensus ROREs in the promoter of *let-7*, we used CRISPR/Cas9-mediated gene editing to scramble the ROREs in pairs (*Figure 4A*). The GC content of the scrambled region was kept the same as in the wild-type ROREs. Mutant RORE strains were outcrossed multiple times and then subjected to molecular assays and phenotypic analyses. For technical reasons, we were only able to generate *let-7(scRORE1,2)* and *let-7(scRORE1,3)* strains.

To determine the extent to which the ROREs were necessary for NHR-23 occupancy at the promoter of *let-7*, we performed ChIP-qPCR in *let-7(xk41-scRORE1,2)*, *let-7(xk39-scRORE1,3)* and wild-type animals during the L4 stage. The level of enrichment of the wild-type *let-7* promoter in the *nhr-23::3xflag* samples was 25-fold higher, relative to control animals. In contrast, the enrichment was only ~5-fold higher in both *let-7(scRORE1,2)* and *let-7(scRORE1,3)* mutants relative to the control animals (*Figure 4B*, *Figure 4—figure supplement 1*). The level of enrichment of the *let-7* promoter in *let-7(scRORE1,2)* and *let-7(scRORE1,3)* mutants was still above background, suggesting that the remaining RORE not scrambled in each of the *let-7(scRORE)* mutants may contribute to some binding by NHR-23.

Next, we queried the levels of primary and mature *let-7* transcripts in *let-7(scRORE1,2)*, *let-7(scRORE1,3)*, and wild-type animals immediately following the L2/L3 molt (*Figure 4C, D*). At the peak of expression (22 h), the levels of pri-*let-7* in *let-7(scRORE1,2)* and *let-7(scRORE1,3)* animals were decreased by 2.4- and 1.7-fold, respectively, relative to wild type (*Figure 4C*). Correspondingly, the levels of mature *let-7* at the same timepoints were reduced by 2.7- and 2.5-fold in the *let-7(scRORE1,2)* and *let-7(scRORE1,3)* animals, respectively, relative to wild-type animals (*Figure 4D*). However, no significant difference was detected in the accumulated levels of mature *let-7* by the L4 stage in *let-7(scRORE1,2)* and *let-7(scRORE1,3)* animals compared to wild-type animals (data not shown). Thus, reduced binding of NHR-23 is accompanied by reduced transcription and slower accumulation of *let-7* in these strains.

To characterize heterochronic phenotypes associated with scrambling the ROREs, we scored the number of seam cell nuclei in the *let-7(scRORE1,2)* and *let-7(scRORE1,3)* mutants and wild-type animals. At least two independent isolates of each scrambled mutant were analyzed. As positive controls, we included two *let-7* hypomorphs, *let-7(n2853)* and *let-7(mg279)*, since these mutants have higher numbers of seam cells than wild-type animals (*Chan and Slack, 2009*; *Reinhart et al., 2000*). Seam cell nuclei were identified and scored in adult animals based on the fluorescent reporter gene *Pscmgfp*, which was crossed into each strain prior to scoring. All lines of the *let-7(scRORE1,2)* and *let-7(scRORE1,3)* strains exhibited significantly increased number of seam cells relative to wild-type adults (*Figure 4E*). The extent of seam cell hyperplasia detected in the *let-7(scRORE)* mutants was comparable to *let-7(mg279)*, but less severe than *let-7(n2853)* (*Figure 4E*).

To examine how the ROREs, and by extension, NHR-23-mediated activation of *let-7*, affect the biorhythm of molting, we conducted longitudinal behavioral studies on *let-7(scRORE1,2)*, *let-7(scRORE1,3)*, and wild-type animals developing from the L3-to-L4 molt until young adulthood (*Figure 4F*). All four mutant strains (i.e., two independent alleles of *let-7(scRORE1,2)* and *let-7(scRORE1,3)*) were found to enter into and emerge from the L4-to-adult molt significantly earlier than wild type, similar to previous findings with *let-7(lf)* mutants. Therefore, scrambling the ROREs is sufficient to increase the speed of development, consistent with our model that the pace of the molting cycle is controlled, in part, by NHR-23-mediated activation of *let-7s*.

Thus, reduced occupancy of the *let-7* promoter by NHR-23, reduced levels of primary *let-7* transcripts, seam cell hyperplasia, and quicker pace of the molting cycle are all associated with the *let-7(scRORE1,2)* and *let-7(scRORE1,3)* mutants. It is likely that the kinetics of accumulation of mature *let-7* strongly affects development of the seam and the pace of molting, consistent with prior reports on the time sensitive nature of *let-7* function (*Reinhart et al., 2000*).

## The *nhr-23* 3′ UTR contains a functional *let-7* consensus site

To determine if NHR-23 and *let-7* constitute a feedback loop, we next asked whether the *let-7* family of microRNAs downregulates *nhr-23* transcript abundance in developing larvae. We identified a single element in the 3′ UTR of *nhr-23* (*Mangone et al., 2010*; *Roach et al., 2020*) that perfectly complements the 5′ seed sequence of *let-7* and partially complements the remainder of the mature microRNA

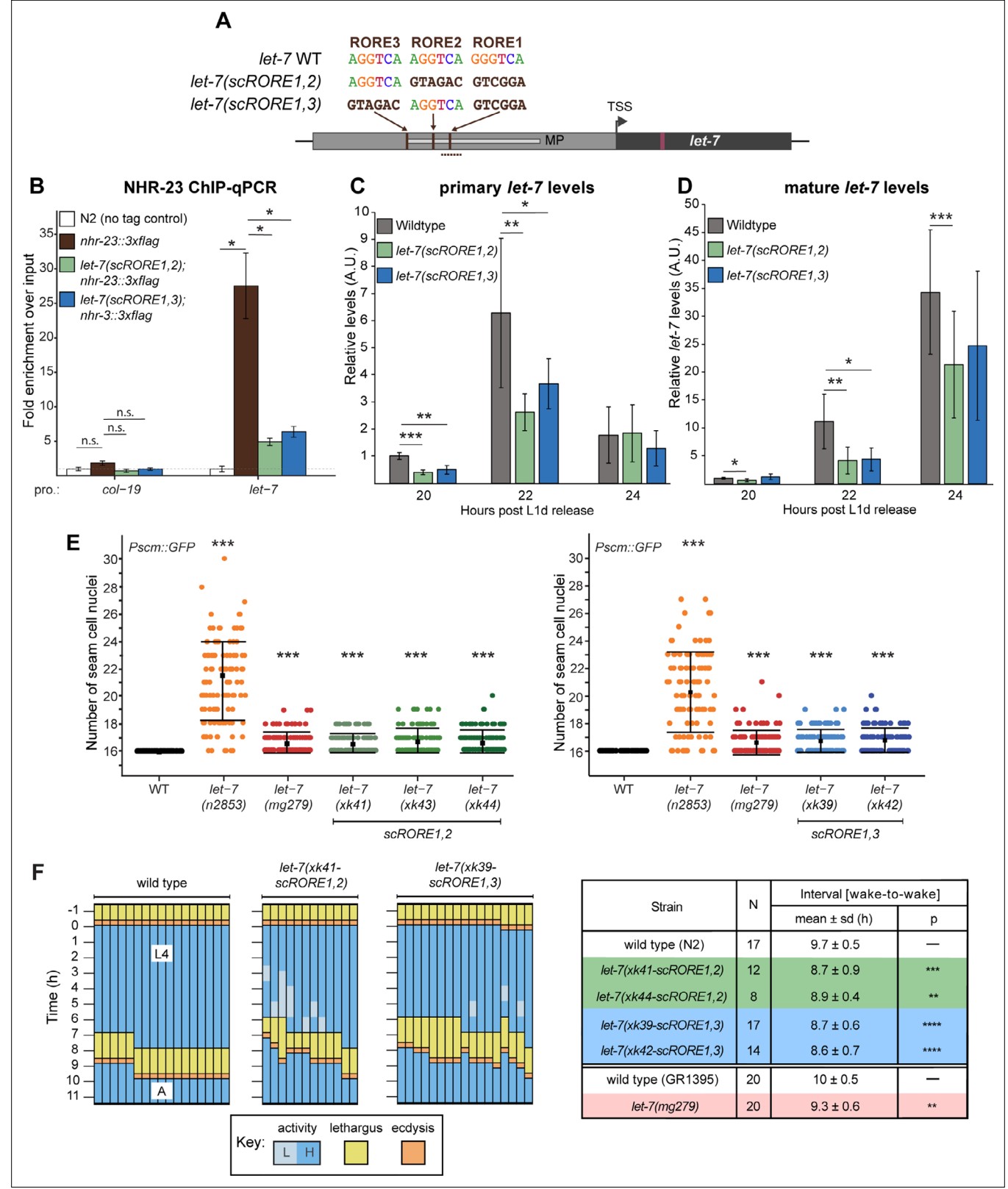

**Figure 4.** Scrambling the ROREs reduces NHR-23 occupancy at the *let-7* promoter and phenocopies *let-7(lf)* mutants. (**A**) The RORE sites in the *let-7* promoter were scrambled in pairs using CRISPR/Cas9-mediated genome editing. The dotted line represents the amplicon quantified by ChIP-qPCR. (**B**) ChIP-qPCR analysis of NHR-23 binding to the *let-7* promoter in wild-type *nhr-23::3xflag* strain and *nhr-23::3xflag; let-7(xk41-scRORE1,2)* and *nhr-23::3xflag; let-7(xk39-scRORE1,3)* mutants. The promoter of *col-19*, which had no NHR-23 ChIP-seq peak, was used as a negative control. One biological

*Figure 4 continued on next page*

*Figure 4 continued*

replicate of ChIP-qPCR from L4 animals is shown. The bar graph represents the mean of two technical replicates. The error bars represent mean ± standard deviation (SD). Values for the amplicon of interest were first normalized to their respective input samples. The average fold enrichment for each genotype was then normalized to the average fold enrichment in N2 (no tag control) samples; n.s. is not significant; *p < 0.05, two-tailed Student's unpaired *t*-test. A second biological replicate is shown in *Figure 4—figure supplement 1*. (**C**) Levels of primary *let-7* transcript determined by RT-qPCR in *let-7(xk41-scRORE1,2)* and *let-7(xk39-scRORE1,3)* mutants immediately after the L2-to-L3 molt. Three biological replicates, with two technical replicates each, are shown. The bar graph represents the mean of the six values first normalized to the levels of *eft-2* and then to the value of the N2 sample at 20 h. The error bars represent mean ± SD; *p ≤ 0.05, **p ≤ 0.01, ***p < 0.001; two-tailed Student's paired *t*-test. (**D**) Same as C, except levels of mature *let-7* determined using Taqman RT-qPCR were first normalized to U18 snoRNA and then to the value of the N2 sample at 20 h; *p ≤ 0.05, **p ≤ 0.01, ***p < 0.001, two-tailed Student's paired *t*-test. (**E**) The number of seam cell nuclei in three independent lines (*xk41*, *xk43*, and *xk44*) of *let-7(scRORE-1,2)* and two independent lines (*xk39* and *xk42*) *let-7(scRORE-1,3)* mutants grown at 25°C are shown. The *let-7(n2853)* and *let-7(mg279)* were scored as controls. All scoring was done in the background of JR672 (*Pscm::GFP*). Mean ± SD shown; *N* ≥ 100 for each strain; ***p < 0.001, one-way analysis of variance (ANOVA). (**F**) Left: Actograms depict the behavior and life stage of single wild-type or mutant animals observed at regular 1 h intervals, as described in *Figure 2B*. Here, high activity (L) refers to continuous pharyngeal pumping, whereas low activity (L) refers to intermittent pharyngeal pumping at the time of observation. Right: Table with wake-to-wake intervals for multiple independent isolates; ****p ≤ 0.0001, ***p ≤ 0.001, **p ≤ 0.01, Mann–Whitney test.

The online version of this article includes the following figure supplement(s) for figure 4:

**Figure supplement 1.** Scrambling the ROREs in the *let-7* promoter reduces NHR-23 occupancy.

sequence. Hereafter, this element is called the *let-7* consensus site (LCS). Three other sequences in the 3′ UTR of *nhr-23* partially complement the *let-7s* with mismatches to the seed (*Figure 5A* and *Supplementary file 2*).

To assess the significance of the LCS on *nhr-23* mRNA levels, we designed and utilized a set of bicistronic reporters for post-transcriptional *cis*-regulatory elements, each housed in a distinct extra-chromosomal array and unique transgenic strain (*Figure 5B*). Briefly, the coding sequence of tandem (td) Tomato was fused with the 3′ UTR of *nhr-23*, whereas the coding sequence of GFP was fused with the 3′ UTR of *unc-54*, which is not targeted by the *let-7s*. An SL2 trans-spliced leader sequence bridged the two fusion genes. The promoter of *dpy-7* drove expression of the operon in the hypodermis. The readout was the ratiometric signal of tdTomato to GFP detected in the lateral epidermis (*Figure 5—figure supplement 1*). This approach controlled for potential differences in gene expression associated with particular arrays or mosaic animals rather than the test 3′ UTR.

*Figure 5C* shows the merged and individual signals detected in transgenic animals in the L4-to-adult molt, at which time both *let-7* and *dpy-7* are highly expressed. The ratiometric signal for the *nhr-23* 3′ UTR reporter was ~6-fold lower than the negative control *unc-54* 3′ UTR reporter (*Figure 5D*). Similarly, the ratiometric signal for the positive control reporter, the 3′ UTR of the known *let-7* target *lin-41* (*Slack et al., 2000*), was 3-fold lower than the negative control. It is unlikely that the 3′ UTR fused to tdTomato affects the efficiency of trans-splicing or causes nonsense-mediated decay of the polycistronic pre-mRNA because the absolute intensities of GFP of all three constructs were equivalent.

We next systematically deleted each of the four predicted *let-7*-binding sites in the *nhr-23* 3′ UTR and compared their reporter signals with the signal detected from the wild-type reporter for *nhr-23* 3′ UTR. Excision of the LCS led to a 2-fold increase in the ratio of tdTomato/GFP signals, relative to the average ratio associated with the reporter for the full-length 3′ UTR of *nhr-23* (*Figure 5E*). In contrast, deletions of the other predicted *let-7*-binding sites (*Δ26–42*, *Δ227–249*, and *Δ623–646*) in the *nhr-23* 3′ UTR reporters did not increase the ratio of the tdTomato/GFP signals. The decrease in the ratio of the tdTomato/GFP signals in the *Δ26–42* and *Δ623–646* deletion mutants of the *nhr-23* 3′ UTR reporters suggests the involvement of other post transcriptional regulatory mechanisms. Thus, the LCS is the only bona fide *let-7*-binding site tested in the *nhr-23* 3′ UTR. Consistent with this result, a high-throughput approach to catalog targets of microRNAs identified the 3′ UTR of *nhr-23* among cellular transcripts associated with ALG-1, the primary Argonaute of the worm microRNA RISC complex (*Broughton et al., 2016*; *Grishok et al., 2001*). Taken together, these data support the hypothesis that *let-7* represses *nhr-23* by directly binding the LCS in its 3′ UTR.

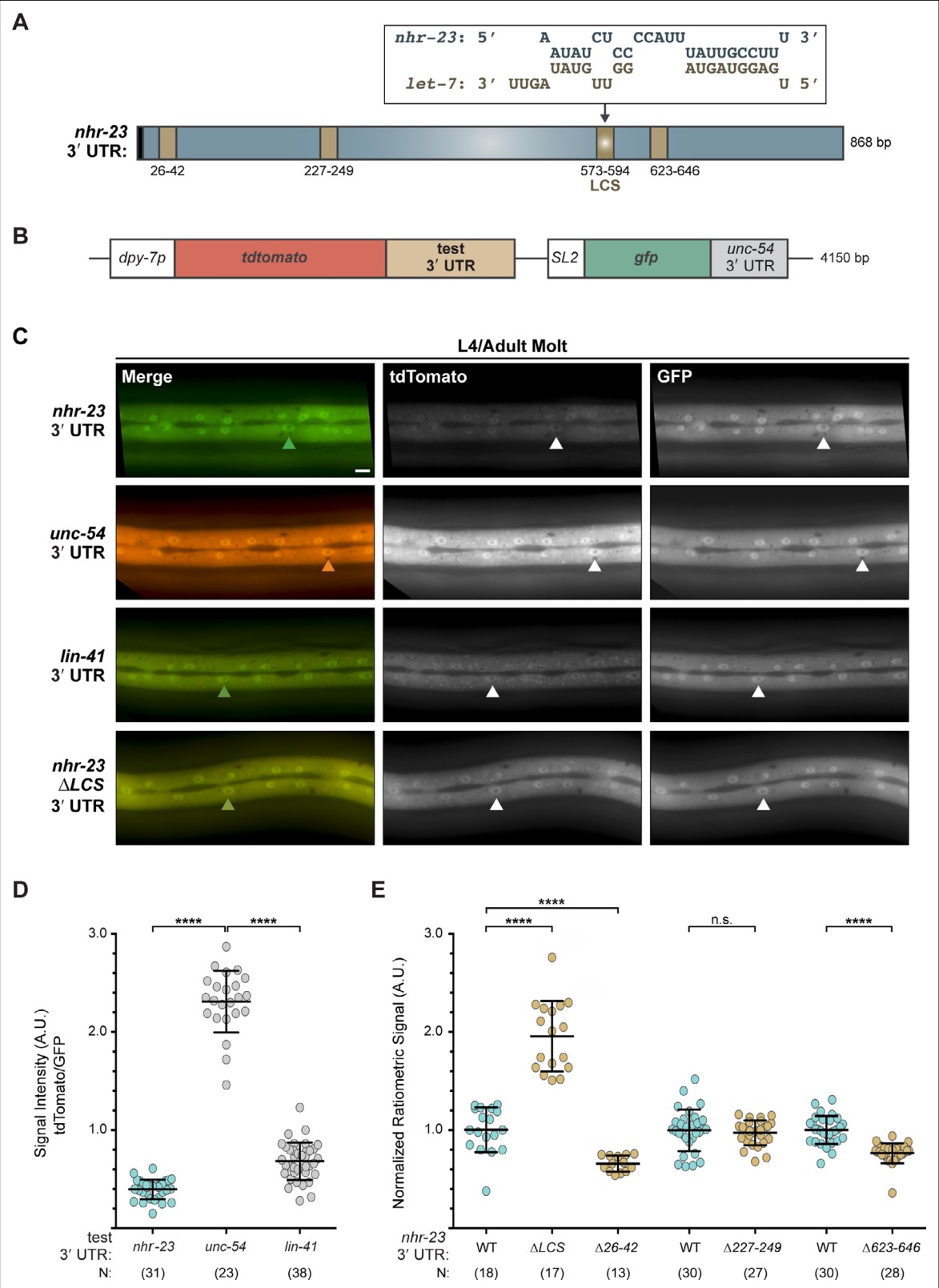

**Figure 5.** The 3′ UTR of *nhr-23* contains a functional *let-7* consensus site (LCS). (**A**) Predicted base pairing between the LCS in the *nhr-23* 3′ UTR and mature *let-7*. Schematic also shows three other predicted *let-7*-binding sites (brown box) and the stop codon (black box). (**B**) Design of bicistronic reporters for 3′ UTR-mediated gene regulation. (**C**) Rows of representative fluorescence images show merged and individual signals from tdTomato and GFP coexpressed in the lateral epidermis of the same worm. Labels indicate the 3′ UTR fused to tdTomato in the corresponding reporter. Arrowheads

*Figure 5 continued on next page*

*Figure 5 continued*

point to hyp-7 nuclei. Scale bar = 10 µm. All images were captured with an exposure time of 10 ms. (**D**) Quantitation of the ratiometric signal (tdTomato/ GFP) associated with each 3' UTR reporter detected. Each symbol represents the average value of three region-of-interests or ROIs per worm. *N* indicates the cumulative sample size from two independent experiments. Bars signify mean ± standard deviation (SD) for the cumulative sample; ****p ≤ 0.0001, ordinary one-way analysis of variance (ANOVA) with Tukey's correction for multiple comparisons. (**E**) As in D, except that ratiometric values were normalized to same-day controls. The full-length (WT) *nhr-23* construct is depicted in blue; deletion constructs in brown.

The online version of this article includes the following figure supplement(s) for figure 5:

**Figure supplement 1.** Design and performance of bicistronic reporters for *cis*-regulatory elements in 3' UTRs of interest.

## Both the LCS and *let-7s* contribute to dampening the expression of *nhr-23*

We next deleted the endogenous LCS of *nhr-23* by CRISPR/CAS9 (*Paix et al., 2015*) to generate the *nhr-23(aaa20-ΔLCS)* strain. We then detected and compared temporal waves in the abundance of *nhr-23* transcripts among wild-type animals and both *nhr-23(ΔLCS)* and *let-7(n2853)* mutants developing from late L2s into young adults by TaqMan RT-qPCR. To stage each strain, we inspected and scored the behavior of ~100 worms as active or quiescent at each timepoint prior to collection of the sample. Lethargi, and by extension the molts, were identified post hoc based on these measurements. Wild-type larvae developed more slowly than the gain-of-function (*gf*) *nhr-23(ΔLCS)* or loss-of-function (*lf*) *let-7(n2853)* mutants in this particular experiment. However, we captured oscillatory expression of *nhr-23* across the target stages among the time samples of each strain (*Figure 6A* and *Figure 6— figure supplement 1A*). Additionally, we used the program Metacycle (*Wu et al., 2016*) to calculate the amplitude and phase of the expression curves of *nhr-23* and performed manual calculations to determine the rates of accumulation and decay of *nhr-23* transcripts (*Figure 6A'*). Peak levels of *nhr-23* were typically detected one-third to one-half of the way through the L2, L3, and L4 stages in wild-type time samples. However, the peak values of sequential waves dropped by a regular increment of ~1.5-fold from one life stage to the next, an indication of dampening (*Figure 6A'*, *Supplementary file 3*).

Three metrics of the expression curves for *nhr-23* – amplitude, peak value and rising slope – were consistently higher both in *nhr-23(ΔLCS)* mutants and *let-7(n2853)* mutants, as compared with wild-type animals, across both the L3 and L4 stages in two independent biological replicates (*Figure 6A, A'*, *Figure 6—figure supplement 1A, A'*, *Supplementary file 3*). For instance, the peaks in *nhr-23* transcript levels that were detected early in L3 and L4 were ~1.6-fold higher in *nhr-23(ΔLCS)* samples than in wild-type samples, despite the dampening (*Figure 6A*, *Supplementary file 3*). Similarly, the amplitude, peak value, and slope of *nhr-23* curves in *let-7(n2853)* mutants were also both significantly higher relative to wild type during the L3 and L4 stages. The phases of the *nhr-23* waveforms differed among the three cohorts but were not consistently earlier in either mutant relative to wild-type animals, across both life stages and biological replicates (*Supplementary file 3*). Interestingly, an extra pulse of *nhr-23* expression was detected in both *nhr-23(ΔLCS)* and *let-7(n2853)* time samples collected after the fourth molt, suggesting the potential for a supernumerary molt (see arrows, *Figure 6A*).

We used a similar approach to determine the extent to which the *let-7s* repress the expression of *nhr-23* during the L2 stage. We compared the abundance of *nhr-23* transcripts in regular time samples of *nhr-23(ΔLCS)* single mutants, *mir-48 mir-241(nDf51); mir-84(n4037)* triple mutants, and wild-type larvae developing from late L1s into early L3s (*Figure 6—figure supplement 1B, B'*). The L2-stage expression curves detected in both mutants were at least 3-fold steeper than those detected in wild-type larvae. Additionally, the amplitude of *nhr-23* expression was 2.3-fold higher in *nhr-23(ΔLCS)* mutants than in control animals.

In complementary studies, we tracked the abundance of NHR-23 protein expression in epidermal nuclei as indicated by the signal associated with the NHR-23::GFP fusion protein. Protein levels also cycled from the L2 through the L4 stage. For example, the signal peaked 2 h after emergence in the L4 stage but was not detected 3 h later (*Figure 6—figure supplement 2A, B*). Both the extent and kinetics of protein increase and decrease corresponded well with the expression curves for *nhr-23* transcripts detected in wild-type larvae. We next asked if the *let-7s* regulate the abundance of NHR-23 proteins by comparing the abundance of the NHR-23::GFP fusion protein in the *let-7(mg279); mir-84(tm1304)* double mutant and wild-type animals (*Figure 6—figure supplement 2C*). GFP was detected in the epidermal nuclei of the *let-7(mg279); mir-84(tm1304)* mutant molting from L4s to

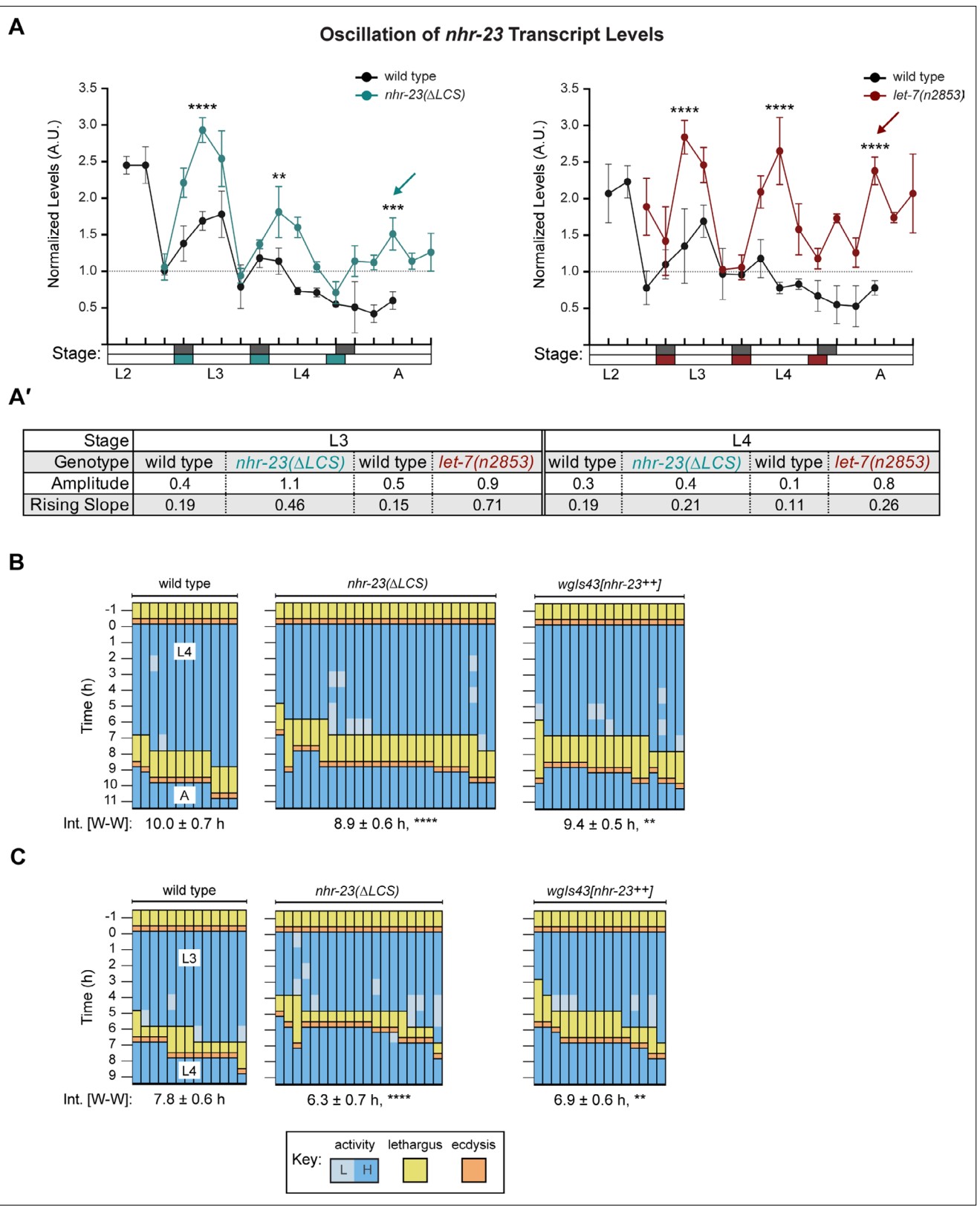

**Figure 6.** Steeper waveforms of *nhr-23* expression and increased pace of development are both associated with the *nhr-23(ΔLCS)* (*aaa20* allele) and *let-7* mutants. (**A**) Levels of *nhr-23* transcripts detected by TaqMan RT-qPCR in regular time samples of wild-type, *nhr-23(ΔLCS)*, and *let-7(n2853)* animals collected from late L2 through early adulthood. Shaded rectangles beneath the *x*-axis signify lethargi in wild-type (dark gray), *nhr-23(ΔLCS)* (teal), and *let-7(n2853)* (maroon) animals; white rectangles signify intervals of physical activity. Transcript levels for *nhr-23* were first normalized to *ama-1* transcripts

*Figure 6 continued on next page*

Figure 6 continued

within each time sample and then further normalized to the mean of all wild-type time samples, represented by the dashed gridline. Dots and error bars represent the mean and range from three technical replicates, respectively. As the rates of development of the three genotypes differ from one another, the waveforms for the wild-type samples were shifted to the left by 4 h in both graphs to align the mutant and wild-type samples by developmental stage. The p values shown are for comparisons between wild-type and mutant values at the indicated timepoints; ****p ≤ 0.0001, ***p ≤ 0.001, **p ≤ 0.01, two-way analysis of variance (ANOVA) with Tukey's correction for multiple comparisons. *Figure 6—figure supplement 1A* shows the results of an independent biological replicate. Arrows point to the supernumerary peaks in *nhr-23* transcript levels detected in both *nhr-23(ΔLCS)* and *let-7(n2853)* animals. Wild-type animals were sampled 24–50 h after release from L1 diapause; *nhr-23(ΔLCS)* and *let-7(n2853)* animals were sampled 22–48 h after L1 diapause. (**A'**) Metrics used to compare the sequential waves of *nhr-23* expression associated with each of the indicated genotypes: *nhr-23(ΔLCS)*, *let-7(n2853)*, and wild type. The amplitude of the waveforms was calculated using Metacycle. The rising slope refers to the rate at which transcript levels ascend from the trough detected before or during the preceding molt to the peak detected within the specified stage. (**B, C**) Actograms depict the behavior and life stage of single animals observed at regular 1 h intervals, as described in *Figure 2*. In this case, high or low activity refers to continuous or sporadic pharyngeal pumping observed during the time sample. As previously described, *Supplementary file 1* has the active, lethargic, and W–W intervals of the cohorts in these studies. ****p ≤ 0.0001, **p ≤ 0.01, ordinary one-way ANOVA with Bonferroni's correction for multiple comparisons.

The online version of this article includes the following figure supplement(s) for figure 6:

**Figure supplement 1.** Both the functional *let-7* consensus site (LCS) in the 3' UTR of *nhr-23* and *let-7*-family miRNAs limit the abundance of *nhr-23* transcripts across larval development.

**Figure supplement 2.** The NHR-23 levels cycle across larval development and are elevated in *let-7(lf)* mutants relative to wild type.

adults but was not readily detected in wild-type molting animals. The signal from NHR-23::GFP became bright in the *let-7(mg279); mir-84(tm1304)* mutant that had emerged as adults but remained dim in wild-type adults. Interestingly, the corresponding 3.4-fold increase in fluorescence intensity matched the 3.4-fold increase in abundance of *nhr-23* transcripts detected in *let-7(n2853)* versus wild-type samples collected at a comparable timepoint. Of note, the native 3' UTR of *nhr-23* was fused to *nhr-23::gfp* in the genetic reagent used in our study, whereas the ectopic 3' UTR of *unc-54*, which is not a target of the *let-7s*, was fused to *nhr-23::gfp* in a distinct reagent used in previous research (*Hayes et al., 2006*; *Kostrouchova et al., 1998*). Thus, the current study is the first to report that the *let-7s* likely directly repress *nhr-23* through association with the LCS in the *nhr-23* 3' UTR to prevent the accumulation of *nhr-23* transcripts and proteins in wild-type adults.

To study how the LCS, and by extension, *let-7*-mediated repression of *nhr-23*, affects the biorhythm of molting, we tracked cohorts of *nhr-23(ΔLCS)* larvae across both the L3 and L4 stages (*Figure 6B, C*). As a complementary approach, we also tracked larvae that expressed multiple copies of *nhr-23* from an integrated, tandem array (*wgIs43[nhr-23++]*) across the same life stages (*Celniker et al., 2009*; *Gerstein et al., 2010*). The majority of *nhr-23(ΔLCS)* L3 larvae entered lethargus and emerged as L4 larvae before most wild-type L3 larvae began to molt. The wake-to-wake interval of the *nhr-23(ΔLCS)* L3-stage cohort was 1.5 ± 0.9 h shorter than that of wild-type L3s. Likewise, the majority of *wgIs43[nhr-23++]* larvae, which overexpress *nhr-23*, entered lethargus and emerged in the next life stage faster than age-matched, wild-type animals (*Figure 6B, C*). The wake-to-wake interval was 6.9 ± 0.6 h for the *wgIs43[nhr-23++]* cohort developing from L3 to L4, compared with 7.8 ± 0.6 h for the wild-type cohort (p ≤ 0.01). Combining *wgIs43[nhr-23++]* with *let-7(n2853)* led to larval lethality and prohibited a similar analysis. Thus, both derepression and increased dosage of *nhr-23* were associated with advanced lethargus and faster cycles, similar to our findings with the *let-7(lf)* mutants (*Figure 2*).

Together, these findings show that the endogenous LCS in the *nhr-23* 3' UTR is indeed a *cis*-regulatory, repressive element, and strongly suggest that *let-7* and its paralogs bind this functional LCS and negatively regulate the expression of *nhr-23* transcripts and proteins, while larvae transit the molts and emerge in the subsequent life stage. Therefore, these data are consistent with a model whereby NHR-23 and the *let-7s* form a transcriptional–post transcriptional feedback loop that regulates the duration of the molt. Immediately following the molt, NHR-23 activates transcription of the *let-7s* early during the larval stage. The post transcriptional repression of *nhr-23* by the *let-7s* keeps the levels of *nhr-23* below a particular threshold, preventing early entry into the next molt.

## Forced expression of *nhr-23* is sufficient to trigger supernumerary molts

As described above, there was no detectable dampening of *nhr-23* transcript levels in *let-7(n2853)* mutants, whereas the phenomenon was obvious in wild-type animals (*Figure 6A*, *Figure 6—figure*

supplement 1A). Mutations in *let-7* were originally characterized as retarded heterochronic mutants that underwent supernumerary molts (*Hayes et al., 2006*; *Reinhart et al., 2000*). Considering this, we hypothesized that *let-7*-dependent dampening of the oscillatory expression of *nhr-23* effectively counts down the number of molts and ultimately extinguishes the molting cycle.

To test this idea, we tracked and compared instances of molting-associated behaviors and animal viability between wild-type adults and age-matched gain-of-function (*gf*) mutants where *nhr-23* is overexpressed: *nhr-23(ΔLCS)* (*Figure 6A*, *Figure 6—figure supplement 1A*) and *wgIs43[nhr-23++]* (*Celniker et al., 2009*; *Gerstein et al., 2010*). First, we inspected partially synchronized populations at regular timepoints 2–5 days after the emergence of adults. Behavioral quiescence, defined by a lack of detectable pharyngeal pumping or locomotion, was more common among both *nhr-23(ΔLCS)* and *wgIs43[nhr-23++]* adults than wild-type animals across this time interval. Moreover, the percentage of quiescent *nhr-23(gf)* adults peaked and significantly exceeded the percentage of quiescent wild-type adults during three to four successive time samples (*Figure 7A*). We next asked whether quiescent *nhr-23(ΔLCS)* and *wgIs43[nhr-23++]* adults observed at those particular timepoints were in fact undergoing lethargi associated with supernumerary molts rather than transient, satiety-induced quiescence (*You et al., 2008*). To distinguish between these two possibilities, we singled quiescent adults into three respective cohorts per genotype and tracked the animals within each cohort for an additional 12 h (*Figure 7B*). In parallel, we singled and tracked quiescent wild-type adults. The over-whelming majority of singled *nhr-23(gf)* adults were quiescent for several hours and then attempted to ecdyse, a sequence of events indicative of a supernumerary molt. Most animals shed entire cuticles or parts thereof, but nonetheless died (*Figure 7—videos 2 and 3*). The *nhr-23(ΔLCS)* adult shown in *Figure 7—video 3* is one such example. The animal was quiescent for 6 h, then exhibited inter-mittent twitches of the grinder, a behavior that accompanies ecdysis, and ultimately bagged, likely because unshed cuticle occluded the vulva. In contrast, all quiescent wild-type adults regained activity and only one animal died during the period of observation (*Figure 7B*, *Figure 7—video 1*). By the abovementioned criteria, 97% (*n* = 34) of singled *nhr-23(ΔLCS)* adults and 91% (*n* = 33) of singled *wgIs43[nhr-23++]* adults underwent supernumerary molts whereas none (*n* = 11) of the wild-type adults did so (p < 0.0001, chi-square test). *Figure 7C* shows one example each of an *nhr-23(ΔLCS)* and a *wgIs43[nhr-23++]* adult that underwent aberrant molts and became trapped in partly shed cuticles. Both animals had eggs in the uterus. However, the *nhr-23(ΔLCS)* animal had an old cuticle attached to its tail. Also, alae were visible on both the lateral surface of the extant cuticle and the partly shed cuticle, implying that the epidermis had terminally differentiated prior to the attempted molt. These results show that forced expression of *nhr-23* is sufficient to initiate a supernumerary molt but not sufficient to properly complete the molt. Taken together, these data suggest that artificially increasing the abundance of NHR-23 relative to the *let-7s* drives additional iterations of the molting cycle.

## Dynamic levels of *nhr-23* and the *let-7s* shape expression curves of many effectors of the molting cycle

NHR-23 and *let-7* may act as core components of a molecular-genetic oscillator that regulates the onset and duration of the molts. Other biological clocks generate and sustain orderly waves in the expression of both core clock components and groups of 'clock-controlled genes' (CCGs) that encode coordinated effectors of the biorhythm. Consistent with this model, genes that are depleted in *nhr-23* knockdowns are strongly enriched for oscillating genes (*Tsiairis and Großhans, 2021*). From this perspective, we considered how the negative feedback loop between *nhr-23* and the *let-7s* might affect the expression of genes that oscillate in phase with different events linked to the molting cycle. To test this idea, we chose two oscillatory genes linked to molting: (1) *fbn-1*, which encodes a compo-nent of the sheath that encloses and protects animals during each molt (*Katz et al., 2021*); and (2) *mlt-10*, which encodes a component of the cuticle (*Frand et al., 2005*; *Meli et al., 2010*). We then queried the expression levels of each of the above transcripts in *nhr-23(RNAi)*, *nhr-23(ΔLCS)*, and *let-7(n2853)* mutants and control animals collected at regular intervals from late L2 through young adulthood using RT-qPCR (*Figure 8A–D*). As described earlier, the amplitude and phase of each waveform were deter-mined using Metacycle, while the slope of each waveform was calculated manually.

As expected, peak levels of *fbn-1* were detected early in the L3 and L4 stages in control animals (*Figure 8A, B*). Knockdown of *nhr-23* during both stages reduced the peak level of *fbn-1* expression by 2-fold (*Figure 8A*, *Supplementary file 3*). The slope of the *fbn-1* waveform was 4.9-fold lower in

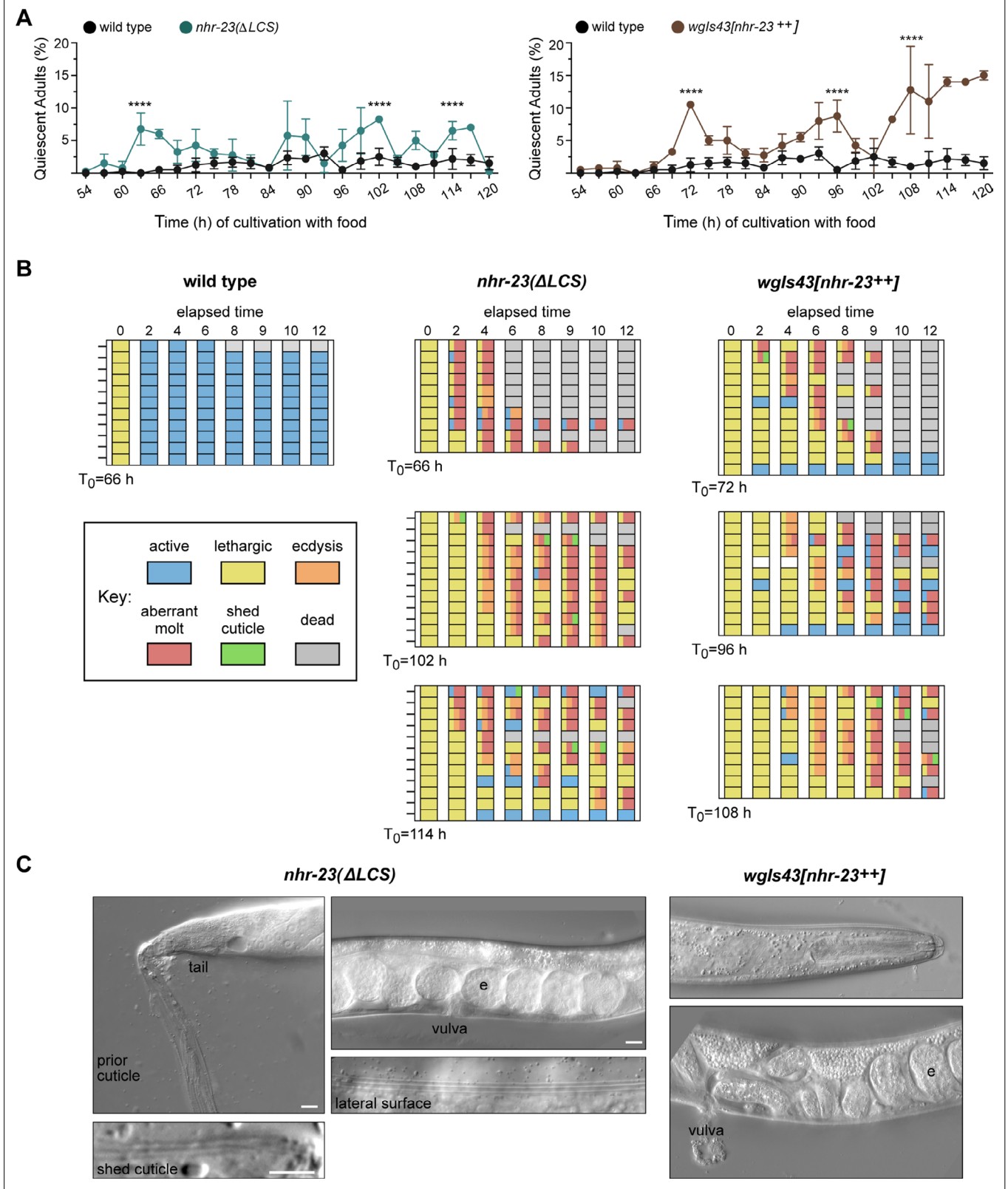

**Figure 7.** Both derepression and ectopic overexpression of *nhr-23* trigger additional molts in reproductively mature animals. (**A**) The percentage of adults in wild-type, *nhr-23(ΔLCS)*, and the *nhr-23* overexpression strain (*wgIs43[nhr-23++]*) that appeared quiescent at regular timepoints 54–120 h after release from diapause and cultivation with food. Values represent the mean ± standard deviation (SD) from two independent trials, with cumulative sample sizes of 300–400 animals per timepoint. The values for the wild-type cohort were repeated in both graphs for ease of comparison. Significant

*Figure 7 continued on next page*

Figure 7 continued

peaks in the prevalence of quiescent animals are marked by asterisks. The corresponding values significantly exceeded the values for age-matched, wild-type animals (p < 0.0001, chi-square test, $\chi^2$ ranged from 39 to 223). (B) Actograms depict the behavior and fate of quiescent adults singled at each timepoint marked by an arrow and then observed at regular 2 h intervals. Records within each row correspond to a single worm. The behavior of a worm at a particular timepoint was scored as active, lethargic, or ecdysing as described in Materials and methods. Aberrant molts and death were also recorded (see Materials and methods). (C) DIC micrographs show examples of adults that attempted to molt. Arrows point to former cuticles dislodged from the tail or head; arrowheads point to alae on both the passing and emergent cuticles. The letter 'e' denotes fertilized embryos within the uterus. Scale bars = 10 μm.

The online version of this article includes the following video for figure 7:

**Figure 7—video 1.** Behavior and fate of a quiescent wild-type adult.

https://elifesciences.org/articles/80010/figures#fig7video1

**Figure 7—video 2.** Behavior and fate of a quiescent *nhr-23(ΔLCS)* adult.

https://elifesciences.org/articles/80010/figures#fig7video2

**Figure 7—video 3.** Behavior and fate of a quiescent *wgIs43[nhr-23++]* adult.

https://elifesciences.org/articles/80010/figures#fig7video3

*nhr-23* knockdowns during the L3 stage and 1.3-fold lower during the L4, relative to age-matched control animals. In contrast, both LCS deletion and *let-7* mutations increased the peak level of *fbn-1* transcripts by ~1.5- to 2-fold in L3- and L4-stage animals, as compared with age-matched controls (*Figure 8B*, *Supplementary file 3*). The slope of the *fbn-1* expression curves was twofold higher in *nhr-23(ΔLCS)* mutants and 3-fold higher in *let-7(n2853)* mutants than wild type (*Supplementary file 3*). Peak values of *fbn-1* expression were detected slightly earlier in both *nhr-23(ΔLCS)* and *let-7(n2853)* mutants developing through the L4 stage, relative to control animals (*Figure 8B*); this finding was replicated in a second, independent trial (*Figure 8—figure supplement 1A*).

Peak levels of *mlt-10* transcripts were detected late in each larval stage, right before animals enter the molt (*Figure 8C*). In L4-stage *nhr-23(RNAi)* larvae, the peak level of *mlt-10* was reduced by 2-fold, relative to control animals (*Supplementary file 3*). Additionally, knockdown of *nhr-23* reduced the slope of the *mlt-10* expression curve to 0.1, compared with 0.4 in control animals, suggesting that *nhr-23* likely affects the rate of accumulation of *mlt-10* transcripts (*Supplementary file 3*). In *nhr-23(ΔLCS)* and *let-7(n2853)* mutants, the peak value of *mlt-10* expression was about 2-fold higher and the rising slope was about 4-fold higher than wild type (*Figure 8D* and *Supplementary file 3*). Again, peaks in *mlt-10* expression were detected slightly earlier during the L4 stage, in both mutants relative to wild-type animals, in two independent trials (*Figure 8D* and *Figure 8—figure supplement 1B*). Thus, the cyclical expression profiles of *fbn-1* and *mlt-10* are altered in *nhr-23* knockdowns, and in *nhr-23(gf)* and *let-7(lf)* mutants. As we describe below, both genes have predicted *cis*-regulatory binding elements for NHR-23 and the *let-7s* in their promoters and 3′ UTRs, respectively, suggesting direct transcriptional activation by NHR-23 and direct repression by the *let-7s*. The feedback loop likely sculpts the temporal expression profiles of *fbn-1* and *mlt-10*, as well as other genes linked to molting.

To determine whether joint regulation by NHR-23 and *let-7s* was a signature of oscillatory genes that are linked to molting, we used a bioinformatics approach. We selected a set of potential target genes of the molting timer based on two criteria: (1) expression of the gene oscillates with a period of 8–10 h across larval development (*Hendriks et al., 2014*; *Kim et al., 2013*); and (2) activity of the gene affects one of the many distinct but interdependent steps within the molting cycle. We consider these genes as CCGs. Collectively, the 67 selected CCGs encode transcription factors, signaling molecules, enzymes, and matrix proteins that are involved in the synthesis and removal of cuticles, and neuropeptides that regulate quiescence and arousal (*Supplementary file 4*). Next, we systematically and independently evaluated each CCG as a potential target of NHR-23 or *let-7s* through meta-analyses of published datasets mentioned below and original bioinformatic approaches. A CCG classified as a direct target of NHR-23 met at least two of the following criteria: (1) NHR-23 occupied the 5′ regulatory region of the gene in vivo, as annotated in a modENCODE ChIP-seq dataset of NHR-23 (*Celniker et al., 2009*; *Gerstein et al., 2010*); (2) the same regulatory region contained more ROREs than expected by chance; and (3) knockdown of *nhr-23* resulted in lower transcript levels (*Kouns et al., 2011*). A CCG classified as a target of the *let-7s* met two criteria: (1) ALG-1 bound the 3′ UTR of the respective mRNA in vivo, as reported in an ALG-1 iCLIP dataset (*Broughton et al., 2016*) and (2) the 3′ UTR contained more LCSs than expected by chance (*Rehmsmeier et al., 2004*).

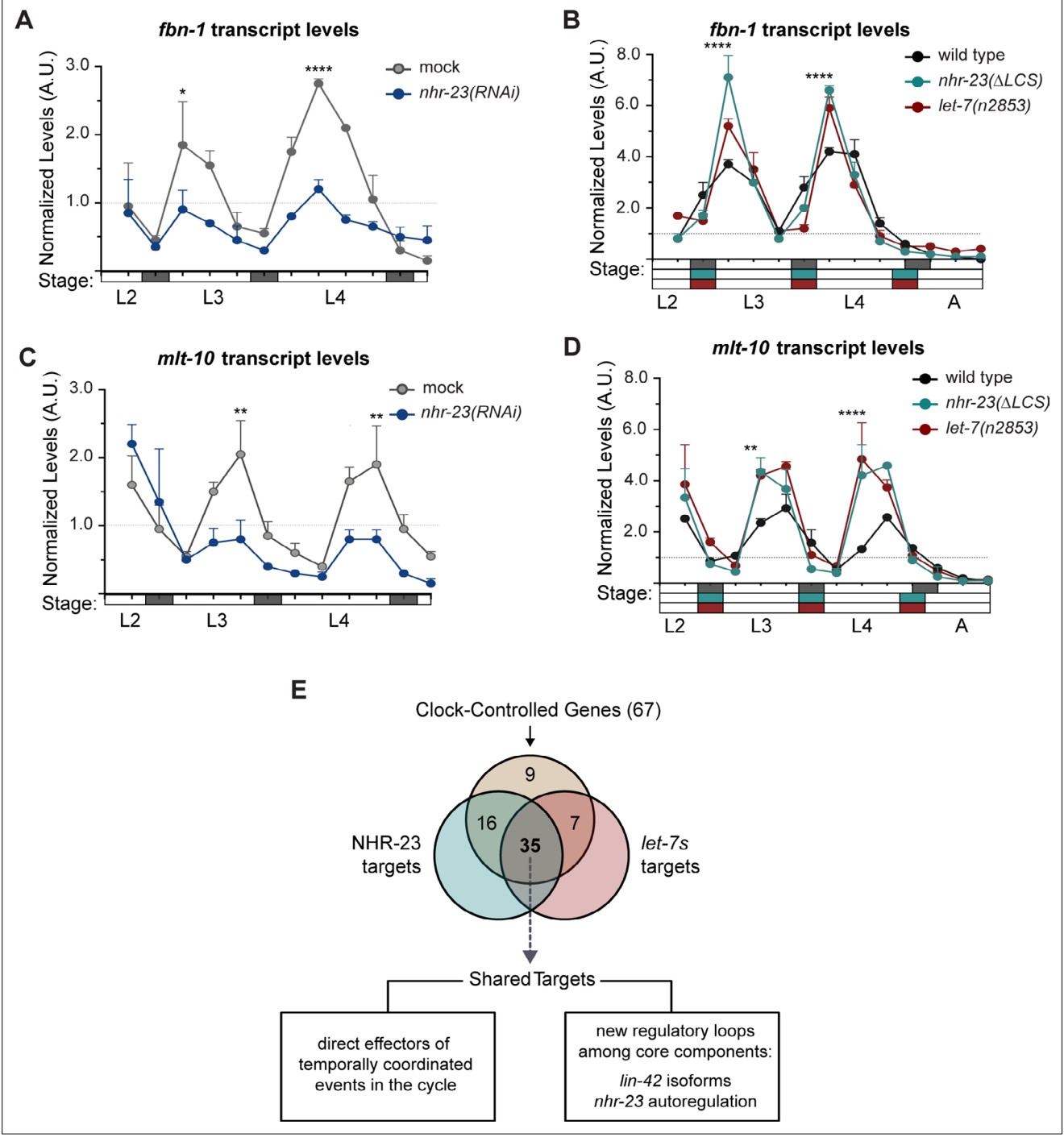

**Figure 8.** Most genes regulated by the molting clock are shared targets of both NHR-23 and *let-7s*. (**A**) Levels of *fbn-1* transcripts detected by TaqMan RT-qPCR in regular time samples of mock-treated and *nhr-23(RNAi)* animals collected from the late L2 through young adulthood. The data were collected from distinct experiments: one set, which comprised two independent trials, covered the late L2 stage until the L3-to-L4 molt; and the other set, which also comprised two independent trials, covered the late L3 stage until the L4-to-adult molt. Lethargus (dark gray boxes) and intervals of physical activity (white boxes) are denoted beneath the x-axis. Values for *fbn-1* were first normalized to *ama-1* transcripts within each same time sample and then normalized to the mean of all wild-type time samples, represented by the dashed gridline. Dots and error bars represent the mean and range from two biological replicates, respectively; ****p ≤ 0.0001, *p < 0.05, two-way analysis of variance (ANOVA) with Bonferroni's correction for multiple comparisons. (**B**) Same as A, except that the levels of *fbn-1* transcripts were measured in wild-type, *nhr-23(ΔLCS)* and *let-7(n2853)* animals collected from late L2 through early adulthood. Shaded rectangles beneath the x-axis depict the molts in in wild-type (dark gray), *nhr-23(ΔLCS)* (teal), and *let-7(n2853)* (magenta) animals; ****p ≤ 0.0001, two-way ANOVA with Bonferroni's correction for multiple comparisons. Wild-type animals were sampled 24–50 h after release from L1 diapause; *nhr-23(ΔLCS)* and *let-7(n2853)* animals were sampled 22–48 h after L1 diapause. (**C, D**) Same as B and C, respectively,

*Figure 8 continued on next page*

eLife Research article

Developmental Biology | Genetics and Genomics

*Figure 8 continued*

except the levels of *mlt-10* transcripts were measured using Taqman RT-qPCR. ****p ≤ 0.0001, **p < 0.01, two-way ANOVA with Bonferroni's correction for multiple comparisons. (**E**) Venn diagram summarizes the classification of 67 clock-controlled genes (CCGs) as direct targets of NHR-23, *let-7*, both, or neither based on original bioinformatic approaches and meta-analyses of published ChIP-seq (*Gerstein et al., 2010*), comparative microarray (*Kouns et al., 2011*), and ALG-1-iCLIP datasets (*Broughton et al., 2016*). *Supplementary file 4* provides the detailed information used to classify each gene of interest. Relevant scoring rubrics are fully described in Results and Materials and methods. The flowchart beneath the Venn diagram shows examples of prospective components of the molting timer and effectors of specific subroutines of the molting cycle that emerged as dual targets from the meta-analysis.

The online version of this article includes the following figure supplement(s) for figure 8:

**Figure supplement 1.** The genes *fbn-1* and *mlt-10* are shared targets of NHR-23 and *let-7s*.

By these criteria, 35 of 67 CCGs (57%) were classified as shared targets of both NHR-23 and the *let-7s* (including *fbn-1* and *mlt-10*), 16 CCGs (24%) as targets of only NHR-23, 7 CCGs (10%) as targets of only *let-7s*, and 9 CCGs (13%) as targets of neither factor (*Figure 8E*, *Supplementary file 4*). Notably, multiple response elements for NHR-23 were identified in the promoters of almost all CCGs classified as *let-7* targets and vice versa, even though NHR-23 or ALG-1 were not enriched at those genomic locations in the abovementioned ChIP-seq or iCLIP datasets, respectively. Therefore, 57% of CCGs as shared targets may be an underestimate and more outputs of the molting timer may ultimately be recognized as targets of both NHR-23 and the *let-7s*. Only 10% of twenty randomly selected genes that are not known to cycle in expression were classified as shared targets of both NHR-23 and *let-7s*, suggesting that NHR-23 and the *let-7s* together may specifically regulate the expression of oscillatory genes that drive molting. These findings suggest that partly interdependent waves in the abundance of NHR-23 and the *let-7s* sculpt the temporal expression profiles of *fbn-1*, *mlt-10*, and possibly many additional effectors of the molting timer.

## NHR-23 and *let-7s* govern the temporal expression profile of other key clock genes

The bioinformatics analysis described above provided more evidence for regulatory interactions among other key components of the oscillator. Our analysis suggested that NHR-23 promotes the expression of both *lin-42/PER* and the *let-7s*, whereas the *let-7s* repress the expression of both *lin-42/PER* and *nhr-23* transcripts. Three major spliced isoforms of *lin-42* are recognized to encode regulators of the molting cycle and components of the heterochronic pathway (*Edelman et al., 2016*; *Jeon et al., 1999*; *Monsalve et al., 2011*). We identified three ROREs in the unique promoter of *lin-42a* and three additional ROREs in the shared promoter of *lin-42b* and *lin-42c* (*Figure 9—figure supplement 1A*). The ROREs in both promoters correspond to sites of NHR-23 enrichment detected in a ChIP-seq dataset from the modENCODE Consortium (*Celniker et al., 2009*; *Gerstein et al., 2010*). Consistent with the data from the modENCODE Consortium, NHR-23 ChIP-qPCR analysis during L3 showed that the *lin-42a* promoter was enriched by 5-fold and the *lin-42b* promoter was enriched 7-fold in the *nhr-23::3xflag* samples, relative to background (*Figure 9A*). To characterize further the extent to which NHR-23 activates the pulsatile expression of *lin-42*, we measured and compared the levels of *lin-42* transcripts across the L4 stage in attenuated *nhr-23* knockdowns and control animals (*Figure 9B*). As expected, levels of *lin-42* in control samples peaked in L3 and once again in L4. No such peak was detected in *nhr-23(RNAi)* at the L4 stage. The transcript levels of *lin-42* detected in *nhr-23* knockdowns at the L4 stage (42 h) were 2.6-fold lower than the peak value detected in age-matched, control larvae. Likewise, the slope of *lin-42* expression was 6-fold lower in L4-stage *nhr-23(RNAi)* larvae, relative to age-matched controls. Moreover, we identified a single RORE site 827–833 bp upstream of the start codon of human *PER2*, suggesting that NHR-23/ROR-mediated transcriptional activation of *lin-42/Per* may be conserved in mammals.

We identified four LCSs, including one with perfect complementarity to the *let-7* seed region, in the shared 3′ UTR of *lin-42a* and *b*, suggesting that the *let-7s* directly repress both *lin-42* isoforms (*Figure 9—figure supplement 1D*). Although *lin-42* was previously described as containing sites complementary to the *let-7s*, the specific *cis*-regulatory elements were not well defined (*Reinhart et al., 2000*). No LCSs were detected in the 3′ UTR of *lin-42c*, which is modeled as a dominant negative (*Monsalve et al., 2011*). To determine how *let-7* affects the expression of *lin-42*, we measured the levels of *lin-42* transcripts in *let-7(n2853)* and wild-type animals (*Figure 9C*). We also included samples

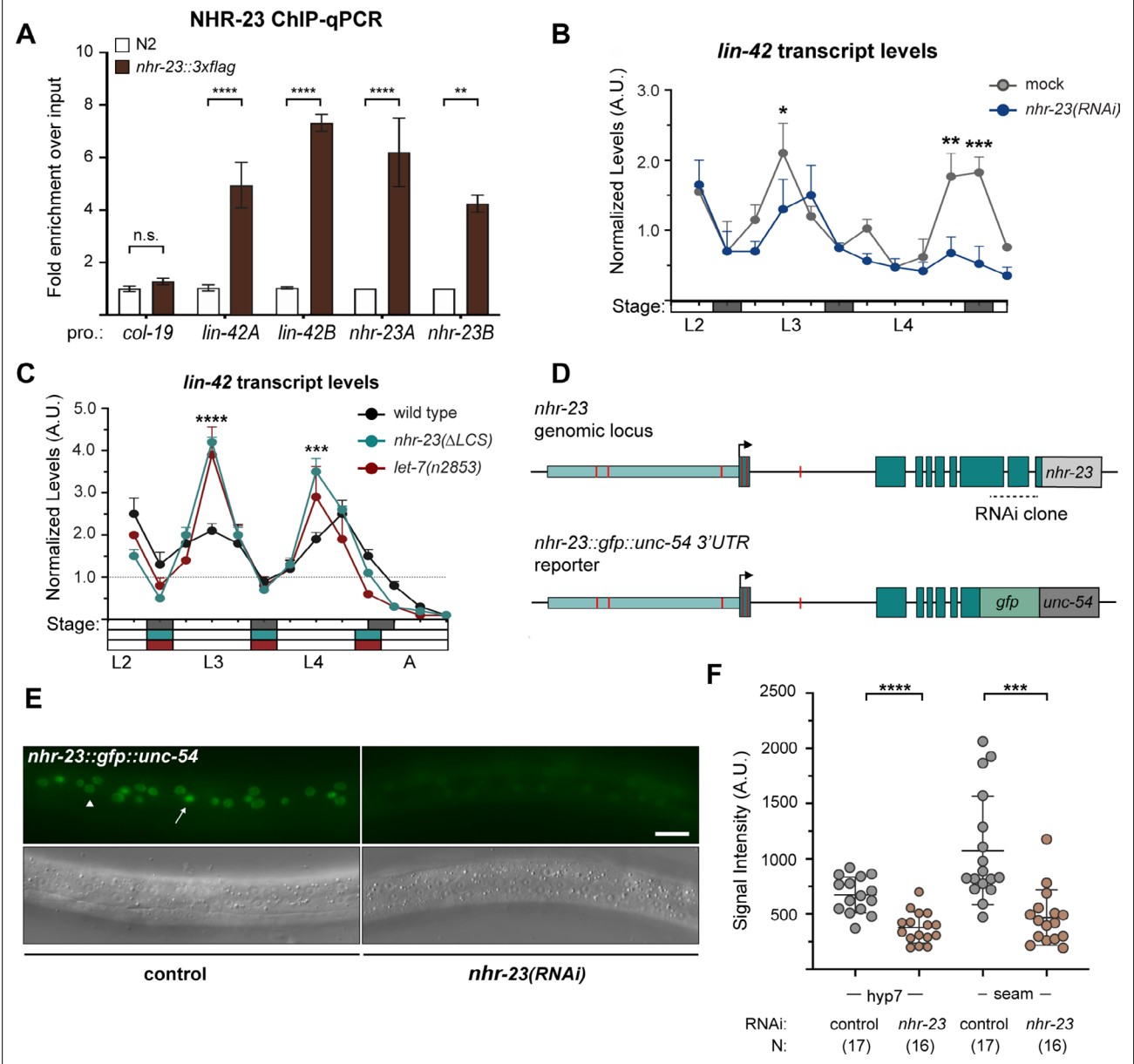

**Figure 9.** NHR-23 and *let-7s* govern the temporal expression profile of other key clock genes. (**A**) ChIP-qPCR for NHR-23 enrichment at *lin-42* and *nhr-23* promoters in N2 (untagged NHR-23) or *nhr-23::3xflag* L3-stage larvae. Values represent the mean ± standard error of the mean (SEM) of three independent trials, each of which included two technical replicates. Values for the amplicon of interest in each sample was first normalized to the respective input and then the average values for QK159 [*nhr-23::3xflag*] were normalized to the average value of the amplicon in N2 within each trial. ****p ≤ 0.0001, **p ≤ 0.01, two-way analysis of variance (ANOVA) with Bonferroni's correction for multiple comparisons. (**B**) Levels of *lin-42* transcripts detected by TaqMan RT-qPCR in regular time samples of mock-treated and *nhr-23(RNAi)* animals collected from the late L2 through young adulthood, as described in *Figure 8A*. ***p ≤ 0.001, *p < 0.05, two-way ANOVA with Bonferroni's correction for multiple comparisons. (**C**) Levels of *lin-42* transcripts detected by TaqMan RT-qPCR in regular time samples of wild-type, *nhr-23(ΔLCS)*, and *let-7(n2853)* animals collected from late L2 through early adulthood, as described in *Figure 8B*. ****p ≤ 0.0001, ***p ≤ 0.001, two-way ANOVA with Bonferroni's correction for multiple comparisons. (**D**) Schematic of the *nhr-23* genomic locus (top) and the *nhr-23::gfp* fusion gene (bottom) that was used to determine the extent to which NHR-23 regulates the expression of itself. The region 2.5 kb upstream (light teal) of the start codon of *nhr-23* isoform A, *nhr-23* exons (dark teal), and the *nhr-23* 3' UTR (light gray) are shown in the genomic locus. In the *nhr-23::gfp::unc-54* reporter, the last two and half exons of *nhr-23* were replaced by the coding sequence for *gfp* as shown. The *nhr-23* 3' UTR was replaced by the *unc-54* 3' UTR (dark gray). The ROREs are shown as red boxes. The dotted line represents the region targeted by *nhr-23* RNAi clone. (**E**) Fluorescence and DIC micrographs show the signal from GFP detected in the lateral epidermis of mid-L4-stage animals that express the *nhr-23::gfp::unc-54* reporter in the mock-treated control animal (left) and *nhr-23(RNAi)* animal (right). The arrow points to a seam cell nucleus and the arrow head points to a hyp7 nucleus. (**F**) Quantified signal intensities of the *nhr-23::gfp::unc-54* reporter as in E.

*Figure 9 continued on next page*

Figure 9 continued

Each circle represents the average of three separate nuclei within the same worm, and error bars indicate the mean ± standard deviation (SD). Scale bar = 15 μm; exposure time = 25 ms. ****p ≤ 0.0001, ***p ≤ 0.001, one-way ANOVA with Bonferroni's correction for multiple comparisons.

The online version of this article includes the following figure supplement(s) for figure 9:

**Figure supplement 1.** NHR-23 and *let-7s* govern the temporal expression profile of other key clock genes.

from *nhr-23(ΔLCS)* mutants in the analysis. The peak levels of *lin-42* expression were 1.5-fold higher in both *let-7(n2853)* and *nhr-23(ΔLCS)* mutants, in the L3 stage, relative to the age-matched control animals (*Figure 9C*). We also detected earlier peaks in *lin-42* expression in both *nhr-23(ΔLCS)* and *let-7(n2853)* mutants developing through the L4 stage, across two independent replicates (*Figure 9C*, *Figure 9—figure supplement 1C*, *Supplementary file 3*). Going further, we identified two LCSs perfectly complementary to the *let-7* seed in the 3′ UTR of human *Per2* transcripts (*Figure 9—figure supplement 1D*), suggesting that the regulatory interactions between LIN-42 and the *let-7s* may be conserved in humans.

Our bioinformatics analysis predicts positive autoregulation of *nhr-23*. We found eight putative ROREs within the upstream regulatory region of *nhr-23*. Two of these ROREs were occupied by NHR-23 in vivo, as indicated by ChIP-seq data from the modENCODE Consortium (*Figure 9—figure supplement 1B*). NHR-23 ChIP-qPCR during L3 showed that the promoter of the *nhr-23* gene was enriched in *nhr-23::3xflag* samples (*Figure 9A*), further substantiating the hypothesis of autoregulation. To test whether NHR-23 promotes its own expression, we used a fusion gene wherein the last two and a half exons of *nhr-23* were replaced with *gfp* fused to the 3′ UTR of *unc-54* (*Figure 9D*). We compared the expression of this fusion gene in the lateral epidermis of *nhr-23* knockdown and control animals during the mid-L4 stage (*Figure 9E, F*). In this assay, the dsRNAs used to downregulate *nhr-23* expression specifically target the last two and half exons and thus, in theory, should affect expression of only endogenous *nhr-23* and not the *nhr-23::gfp::unc-54* transgenic reporter. Fluorescence signal was easily detectable in the epidermis of control animals, but not in *nhr-23(RNAi)* larvae (*Figure 9E*). The intensity of GFP detected in hyp7 of *nhr-23(RNAi)* animals was ~2-fold lower than mock-treated animals (*Figure 9F*). The intensity of GFP in the seam was similarly lowered upon knockdown of *nhr-23*. These data suggest that NHR-23 activates its own expression. Together, these data suggest that the positive autoregulation of *nhr-23*, in combination the NHR-23–*let-7* negative feedback loop, may contribute to a self-sustaining molecular-genetic oscillator. Key components of the molting cycle timer, including *lin-42* and CCGs, may be regulated by both NHR-23 and *let-7s*.

## Reciprocal regulatory elements may be conserved in mammalian *ROR* and *let-7* genes

We next asked whether the feedback loop between NHR-23/ROR and the *let-7s* may be conserved between nematodes and vertebrates. Using bioinformatic approaches, we searched for ROREs upstream of the homologs of *let-7* in the sequenced and annotated genomes of humans, mice, and zebrafish. We inspected the genomic region 3 kb upstream of the precursor *let-7* microRNA and identified one to five distinct ROREs in all homologs (*Figure 10—figure supplement 1A*). *Figure 10A* depicts the ROREs found upstream of selected homologs of *let-7*. In each example, more ROREs were found than predicted by chance.

To determine the extent to which NHR-23/ROR could promote the expression of mammalian homologs of *let-7*, we generated a transgenic worm strain harboring a plasmid composed of the promoter of *M. musculus let-7a-1* fused with *gfp*. Using fluorescence microscopy, we then detected and compared the expression of this reporter gene in *nhr-23(RNAi)* and control animals undergoing the L4-to-adult molt (*Figure 10B, C*). Fluorescence signal of the GFP reporter in control animals was bright in the pharynx, a tissue where *nhr-23* is normally expressed (*Figure 10B*; *Kostrouchova et al., 1998*). In contrast, negligible signal was detected in the pharynx of *nhr-23(RNAi)* animals. Quantitative analyses of the GFP fluorescence signal show that the intensity in *nhr-23(RNAi)* animals was ~2-fold lower, on average, than control animals (*Figure 10C*). Thus, these findings suggest that NHR-23 regulates the expression of GFP from the promoter of *M. musculus let-7a-1* in *C. elegans* and that the positive arm of the NHR-23–*let-7* feedback loop may be conserved to mammals.

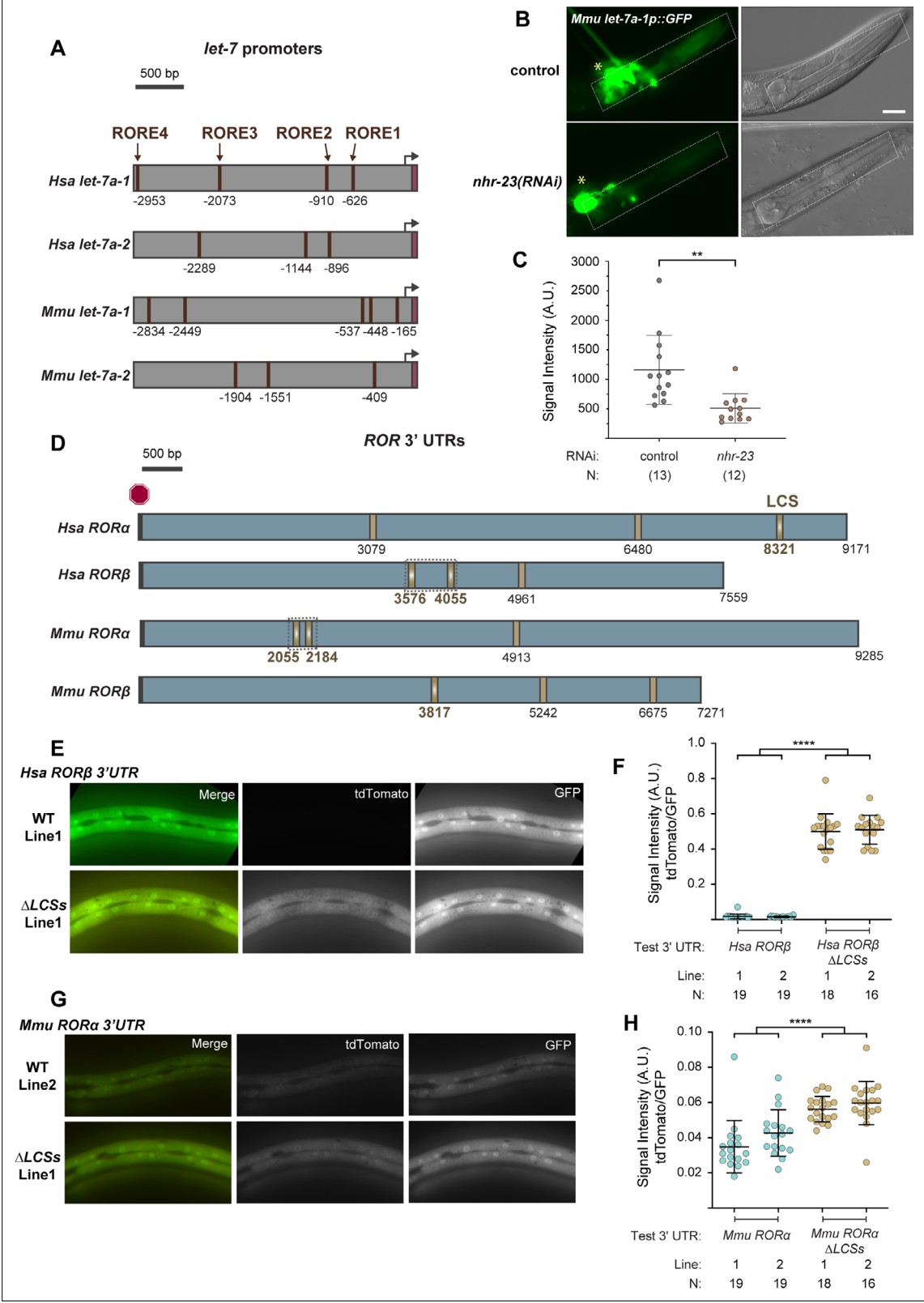

**Figure 10.** RORE and *let-7* consensus site (LCS) elements may be conserved in mammalian *let-7* and ROR genes, respectively. (**A**) Each schematic depicts the 3-kb region upstream of selected homologs of *let-7*. ROREs identified upstream of mature *let-7* (magenta) in the indicated species are shown. Black arrows indicate probable, but not experimentally verified, *let-7* transcriptional start sites. (**B, C**) A transgenic *C. elegans* strain expressing a GFP reporter driven by a 3-kb fragment upstream of *M. musculus let-7* was used to determine the extent to which *C. elegans nhr-23* regulates the

*Figure 10 continued on next page*

*Figure 10 continued*

expression of *M. musculus let-7*. Fluorescence and DIC micrographs show expression of the GFP reporter in the pharynx (dotted rectangle) of mock-treated and *nhr-23(RNAi)* animals undergoing the L4-to-adult molt. Quantification of fluorescence intensity is shown in the graph in C. Each data point represents the mean of three region-of-interests or ROIs measured within the pharynx and error bars depict the mean ± standard deviation (SD) in measurements across the entire sample. The yellow asterisk represents the GFP signal in the neurons attributed to the *ttx-3::gfp* coinjection marker. Scale bar = 15 µm; exposure time = 200 ms. **p ≤ 0.01, one-way analysis of variance (ANOVA) with Bonferroni's correction for multiple comparisons. (**D**) Predicted LCSs (gold) in the 3′ UTRs of four annotated homologs of *nhr-23/ROR*. The red hexagon depicts the stop codon. Gradients and bold labels distinguish sites with perfect complementarity to the seed sequence of *let-7s*. The dotted boxes indicate regions tested in E–H. Each 3′ UTR was retrieved from the UCSC genome browser and verified by comparison with curated ESTs; LCS-*let-7* duplexes were also examined by RNAhybrid. *Supplementary file 2* provides additional information about the prospective duplexes between each of these LCSs and *let-7*. Accession numbers for the related ESTs and genomic sequences are included in the Key Resources Table. (**E–H**) Representative images and quantitation of the ratiometric signal of tdTomato/GFP from bicistronic 3′ UTR reporters (as in *Figure 5B*) of *H. sapiens RORβ* and *M. musculus RORα* 3′ UTRs and variants thereof that lack the LCSs. Each data point in F and H represents the average value of three ROIs per worm. *N* indicates the total sample size from two independent experiments. Bars signify the mean ± SD for the sample. ****p ≤ 0.0001, ordinary one-way ANOVA with Bonferroni's correction for multiple comparisons. Representative images from multiple independent isolates are shown in *Figure 10—figure supplement 1*.

The online version of this article includes the following figure supplement(s) for figure 10:

**Figure supplement 1.** Reciprocal regulatory elements may be conserved in mammalian RORs and *let-7*.

Next, we searched for LCSs in the 3′ UTRs of all 13 homologs of *nhr-23/ROR* annotated in the reference genomes of mice and humans (*Figure 10D*), as well as flies, frogs, zebrafish, and chickens (*Supplementary file 2*). We first aligned and compared the nucleotide sequence of the query 3′ UTR with the sequences of corresponding ESTs. In two cases – zebrafish RORβ and RORγ – multiple ESTs supported longer 3′ UTRs than those presently annotated on the UCSC Genome Browser (see Key Resources Table). We found one to three LCSs perfectly complementary to the seed sequence of *let-7* within 3′ UTRs of 10 of the *nhr-23/ROR* homologs. We also found one or two more LCSs with a single mismatch to the seed sequence of *let-7* in 6 of the corresponding 3′ UTRs. For example, we identified one perfectly complementary LCS in the center of the validated 3′ UTR of human RORβ, flanked by two more sites with respective single nucleotide mismatches to the seed of *let-7* (*Figure 10D*). Similar LCSs with at most a single mismatch to the seed sequences of the *let-7s* were found in the 3′ UTRs for each of the remaining four homologs (*Supplementary file 2*).

To test the significance of the LCSs detected in the mammalian ROR 3′ UTRs, we focused on the 3′ UTRs of *H. sapiens RORβ* and *M. musculus RORα*. Each 3′ UTR has two predicted LCSs that perfectly match the seed region of *let-7*: sequences starting at nucleotide positions 3576 and 4055 in *H. sapiens RORβ*, and 2055 and 2184 in *M. musculus RORα* (*Figure 10D*). For further study, we selected a 590-bp fragment of the *H. sapiens RORβ* 3′ UTR containing two LCSs and a 281-bp fragment of the *M. musculus RORα* 3′ UTR also containing two LCSs (*Figure 10D*, boxed regions). We fused each fragment, as well as variants thereof that lack both LCSs, to *tdTomato* in the bicistronic reporter system described in *Figure 5B*. Using fluorescence microscopy, we measured the intensity of tdTomato and GFP signals in the lateral epidermis of animals undergoing the L4-to-adult molt. When the *H. sapiens RORβ* 3′ UTR was fused to *tdTomato*, the intensity of tdTomato signal was barely detectable in the epidermis, whereas the intensity of GFP signal in the same animal was easily detected (*Figure 10E*, *Figure 10—figure supplement 1B*). Deletion of both LCSs from the *H. sapiens RORβ* 3′ UTR increased the intensity of tdTomato signal by ~30-fold relative to the wild-type constructs (*Figure 10F*). Similar findings were made with the bicistronic reporter constructs that housed the *M. musculus RORα* 3′ UTR (*Figure 10G, H*, *Figure 10—figure supplement 1C*). These data suggest that *C. elegans let-7* can repress the expression of specific mammalian ROR 3′ UTRs. Taken together, these findings imply that the feedback loop between NHR-23/ROR and the *let-7s* may be conserved and may regulate the cyclical expression of target genes in mammalian tissues.

## Discussion

The principal findings of this study unite two distinct time keeping mechanisms mutually dependent on a transcriptional–post-transcriptional feedback loop between NHR-23 and the *let-7s*: the heterochronic genetic pathway that controls the singular switch from larval to adult fates and the genetic oscillator that controls the biorhythm of the molting cycle.

## NHR-23 transcriptionally activates *let-7* and *let-7* post-transcriptionally represses *nhr-23* to form a molecular-genetic oscillator

In *C. elegans*, the primary transcripts of the *let-7* family of microRNAs (miR-48, miR-84, miR-241, and *let-7*) oscillate in abundance and peak once in every larval stage (*McCulloch and Rougvie, 2014*; *Van Wynsberghe et al., 2011*). This oscillatory gene expression pattern is regulated at the transcriptional level (*Kai et al., 2013*). A few negative and positive regulators of *let-7* transcription have been identified previously, although none completely account for the oscillatory expression of the primary *let-7s*. The transcription factor HBL-1 negatively regulates the transcription of *let-7* in the hypodermis until the L3 stage (*Roush and Slack, 2009*). Similarly, the transcription factor LIN-14 restricts the transcription of *mir-48*, *mir-84*, and *mir-241* until the L2-to-L3 transition (*Tsialikas et al., 2017*). The Period protein homolog LIN-42 binds the promoter of *let-7* and dampens the amplitude of primary *let-7* oscillations to prevent early accumulation of mature *let-7* (*McCulloch and Rougvie, 2014*; *Perales et al., 2014*; *Van Wynsberghe et al., 2014*). LIN-42 also dampens oscillations of primary miR-48 transcripts and prevents the early accumulation of mature miR-48. However, *lin-42* is not required for the oscillatory expression pattern of the primary *let-7* transcripts (*McCulloch and Rougvie, 2014*). The transcription factor DAF-12 is involved in a complex regulatory network that couples environmental and developmental signals to regulate *let-7* family transcription. During unfavorable conditions, unliganded DAF-12 represses the expression of the *let-7s*. However, during favorable conditions, and in the presence of its ligand, DAF-12 binds the promoters of *mir-84* and *mir-241* and mildly upregulates transcription (*Bethke et al., 2009*; *Hammell et al., 2009*). The GATA transcription factor ELT-1 binds upstream of *let-7* and promotes transcription during the L4 stage. ELT-1 also acts redundantly with DAF-12 to positively regulate the levels of miR-48, miR-84, and miR-241 (*Cohen et al., 2015*). However, both *daf-12* and *elt-1* are expressed at nearly constant levels during post embryonic development (*Hendriks et al., 2014*; *Kim et al., 2013*) and hence, cannot completely explain the oscillatory expression pattern of the primary transcripts of the *let-7* family.

We have shown that NHR-23 binds upstream of *let-7*, *mir-48*, *mir-84*, and *mir-241* during L3 and L4 stages (*Figure 3B, C*, *Figure 3—figure supplement 1C,D*) and is required for the transcriptional activation of these microRNAs of the *let-7* family (*Figure 3D–G*, *Figure 3—figure supplement 2*). Importantly, we show that NHR-23 is necessary for the cyclic expression profile of primary *let-7*, miR-48, and miR-84, as the oscillations in the primary transcript levels of these microRNAs are dampened upon knockdown of *nhr-23* (*Figure 3D, F*, *Figure 3—figure supplement 2C*). We demonstrate that scrambling the RORE sites in the *let-7* promoter results in decreased binding by NHR-23 (*Figure 4B*, *Figure 4—figure supplement 1*), decreased primary *let-7* levels (*Figure 4C*), and slower accumulation of mature *let-7* (*Figure 4D*). The increased number of seam cells (*Figure 4E*) and the quicker pace of development of *let-7(scRORE1,2)* and *let-7(scRORE1,3)* animals (*Figure 4F*) also supports our conclusion that RORE-dependent transcriptional activation of *let-7* by NHR-23 plays important physiological roles in seam cell fate specification and the timing of developmental events.

A previous study proposed that *let-7* and miR-84 negatively regulated *nhr-23* by an indirect mechanism, independent of the 3′ UTR of *nhr-23* (*Hayes et al., 2006*). Here, we show that *let-7* directly represses *nhr-23* in a manner dependent on a *let-7* complementary sequence (LCS) in the *nhr-23* 3′ UTR (*Figures 5 and 6A, A′*, *Figure 6—figure supplement 1A, A′*). However, our data do not rule out the possibility of additional regulatory pathways that are dependent on *let-7* but independent of the *nhr-23* 3′ UTR. During the adult stage, the level of inappropriately expressed *nhr-23* in *let-7(n2853)* was higher than the level of *nhr-23* in the *nhr-23(ΔLCS)* strain, suggesting the involvement of more than one pathway in the repression of *nhr-23* by *let-7* (*Figure 6A*, *Figure 6—figure supplement 1A*). During the juvenile-to-adult transition, *let-7* represses *lin-41*, which encodes an RNA-binding protein, and this allows the translation of the LIN-41 target, *lin-29* (*Reinhart et al., 2000*; *Slack et al., 2000*). Mutants of *lin-29* exhibit increased expression of NHR-23 during adulthood, suggesting that LIN-29 represses transcription of *nhr-23* (*Harris and Horvitz, 2011*). Therefore, the *lin-29*-mediated inhibition may be the 3′ UTR-independent pathway by which *let-7* represses *nhr-23*.

Thus, NHR-23 and *let-7* form a transcriptional–post-transcriptional negative feedback loop. Within a given larval stage, NHR-23 promotes the expression of the *let-7s* and the *let-7s* repress *nhr-23*. The expression levels of primary *let-7* peak after *nhr-23*, resulting in an intrinsic delay between the accumulation of *nhr-23* and primary *let-7* (*Figures 3D, F and 6A*). Across development, the *let-7s* dampen the relative amplitude of *nhr-23* expression from one larval stage to the next (*Figure 6A*, *Figure 6—figure*

*supplement 1A*). Furthermore, NHR-23 autoregulates its own expression (*Figure 9A, D–F*). Together, these interconnected feedback loops set up a self-sustained molecular-genetic oscillator that is extinguished in adulthood.

## Negative feedback between NHR-23 and *let-7s* sets the pace of the molting cycle

The feedback loop between NHR-23 and the *let-7s* functions in an oscillator-based mechanism to regulate the duration of the molting cycle, in part by driving waves in expression of both key clock components, as well as output/target genes. We have shown that the Period homolog *lin-42* is a transcriptional target of NHR-23 (*Figure 9A–C*, *Figure 9—figure supplement 1C*). The *let-7* family also post-transcriptionally represses *lin-42*. (*McCulloch and Rougvie, 2014*; *Perales et al., 2014*; *Van Wynsberghe et al., 2014*). Similarly, other genes necessary for molting, including *fbn-1* and *mlt-10*, are shared targets of NHR-23 and *let-7* (*Figure 8*, *Supplementary file 4*). The transcription factor NHR-25, which is required for molting, may also be a shared target of NHR-23 and *let-7* (*Supplementary file 4*; *Hayes et al., 2006*). We propose that the NHR-23–*let-7* feedback loop acts within the LIN-42/PER-based molting cycle timer, alongside other as-yet unidentified components.

Based on our findings, we propose a model to explain how the feedback loop between NHR-23 and the *let-7s* controls the speed of the molting cycle. Early in each larval stage, as animals commit to a forthcoming molt, NHR-23 first reaches a functional concentration at the promoters of genes with relatively higher numbers of ROREs, such as *fbn-1* and *noah-1*, and initiates the gene expression programs leading to the biogenesis of the sheath (*Figure 11A*). The sheath is a temporary exoskeleton that encapsulates molting animals and is thought to protect the body of the worm from rupturing while the old cuticle is released and a new one is synthesized (*Katz et al., 2021*). At the same time, NHR-23 also promotes accumulation of the *let-7* family of microRNAs. As NHR-23 continues to accumulate, it begins to activate the expression of genes with relatively fewer ROREs such as *mlt-10* and *osm-11*, which, respectively, encode components of the cuticle and lethargus-promoting peptides (*Meli et al., 2010*; *Singh et al., 2011*). At this time, NHR-23 might also promote the expression of another core clock component, *lin-42*. In this manner, NHR-23 might schedule the start of cuticle biogenesis and onset of lethargus. Then, *let-7*-mediated repression of the same CCGs and *lin-42* likely governs both the end of cuticle remodeling and lethargus. Repression of *nhr-23* delays accumulation of the NHR-23 protein in the next life stage and the onset of any subsequent molt (*Figures 6 and 7* and *Figure 6—figure supplement 1A*). Autoregulation by NHR-23 likely contributes to the rate of accumulation of *nhr-23* transcripts and proteins (*Figure 9D–F*).

While the levels of primary *let-7* oscillate with one peak per larval cycle, the levels of mature *let-7* accumulate starting at early L3. Consistent with previous reports (*Van Wynsberghe et al., 2011*; *McCulloch and Rougvie, 2014*), no oscillations were observed in the levels of mature *let-7*. The association with Argonaute proteins likely stabilizes mature *let-7* levels (*Grishok et al., 2001*; *Winter and Diederichs, 2011*). However, as microRNA levels were measured in whole animals, the possibility of oscillations of mature *let-7* levels in a subset of tissue types cannot be ruled out. Nevertheless, the action of mature *let-7* is responsible for the appropriate timing and rhythmic expression of its target genes like *mlt-10* and *fbn-1*. Similarly, previous reports have shown that the nonoscillatory expression of *bantam* and miR-279 microRNAs is also important for the appropriate rhythmicity of the circadian rhythm-associated genes and related periodic behaviors in flies (*Kadener et al., 2009*; *Luo and Sehgal, 2012*; *Vodala et al., 2012*).

Thus, negative feedback between NHR-23 and the *let-7s* regulates the pace of the molting cycle in part by controlling the rate at which *nhr-23* transcripts accumulate and the amplitude of *nhr-23* expression. This model is consistent with the earlier onset of lethargus and accelerated development observed in both *nhr-23(ΔLCS)* and *let-7(n2853)* mutants (*Figures 2B and 6B*). Both mutants have steeper curves and higher amplitude of *nhr-23* expression (*Figure 6A* and *Figure 6—figure supplement 1A*). In contrast, delayed and protracted lethargus are observed in *nhr-23(RNAi)* animals (*Figure 2B*), which have shallower curves and lower amplitude of *nhr-23* expression. The determination of whether the pace of the molting cycle is regulated by the amplitude of *nhr-23* expression versus the rate of accumulation of *nhr-23* transcripts will require future experiments wherein the two factors are manipulated independently of one another.

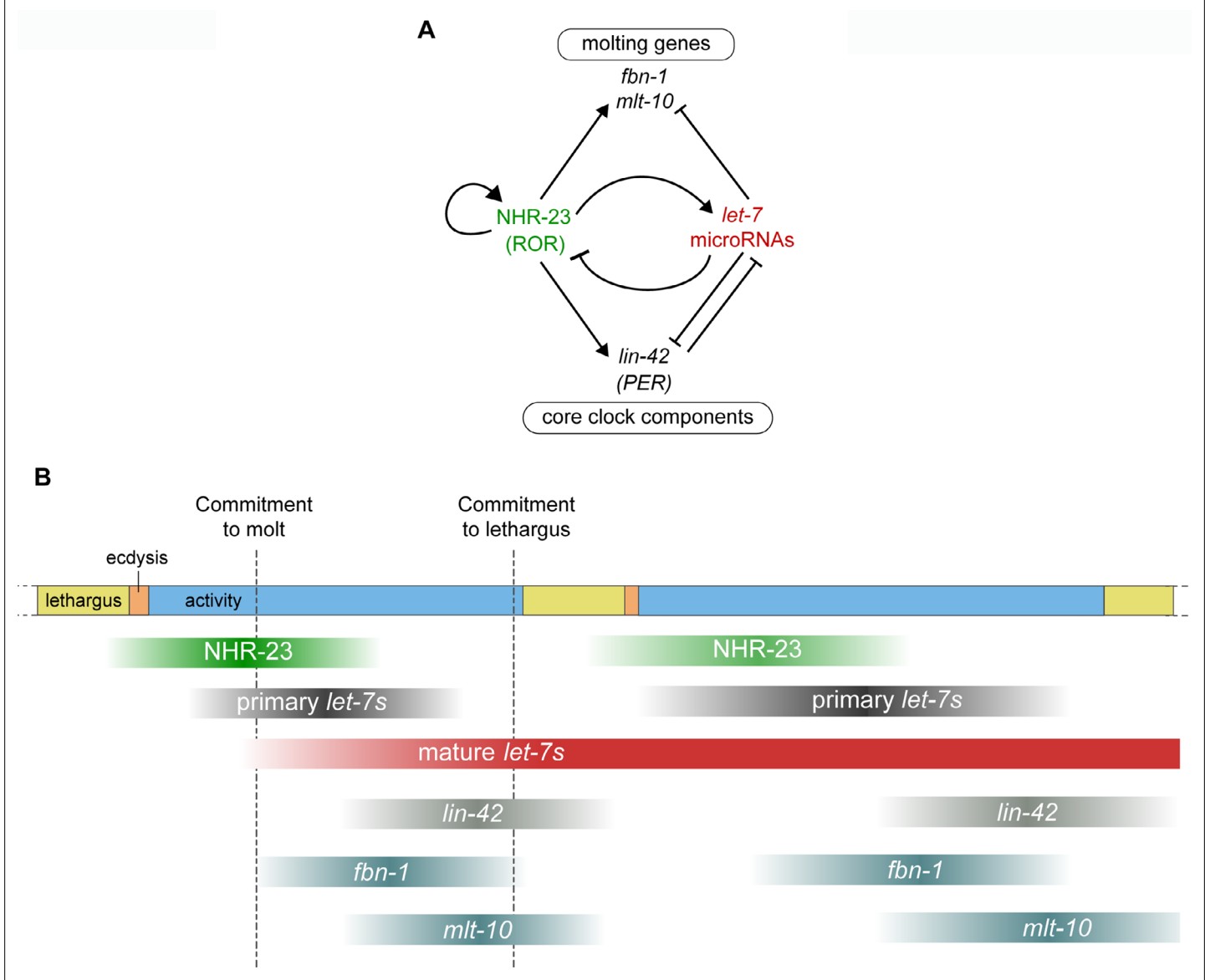

**Figure 11.** Negative feedback between NHR-23 and *let-7s* regulates the gene expression dynamics of core clock components and molting genes, and thus unites the molting timer with unidirectional transitions in the life cycle of *C. elegans*. (**A**) Genetic pathways investigated in this study. NHR-23 promotes transcription of *let-7s* and *let-7s*, in turn, inhibit the accumulation of *nhr-23* transcripts, setting up a negative feedback loop. NHR-23 can autoregulate the transcription of its own gene product. NHR-23 and *let-7* together regulate levels of core clock component, *lin-42* and molting genes, *mlt-10* and *fbn-1*. Arrowheads depict positive regulation and bars indicate negative regulation. (**B**) Schematic summarizing the expression profiles and relative timing of NHR-23, primary *let-7s*, mature *let-7s* and clock-controlled genes (CCGs), investigated in this study, over two molts. The top bar depicts progression of the molting cycle, as color coded in *Figure 1*. Gradient shading in green depicts the rise and fall of NHR-23 protein levels during each larval stage. Similar gradients shaded in dark gray and red depict the levels of primary *let-7s* and mature *let-7s*, respectively. Lighter gradients in shades of gray depict the levels of *lin-42*, *fbn-1*, and *mlt-10*. Dampening of NHR-23 oscillations during the second cycle relative to the first cycle has been indicated. The timepoints at which the depicted gene products rise and fall were estimated based on findings of this report and other experimental evidence (*Hendriks et al., 2014*; *Kim et al., 2013*).

In theory, the intrinsic rates of ascent of *nhr-23* and the *let-7s* transcripts, and the time difference between the accumulation of NHR-23 and accumulation of the *let-7s* together likely impact the amplitudes of the expression curves of multiple CCGs and, by extension, the temporal organization of critical phases of the molting cycle such as cuticle synthesis, lethargus, and ecdysis. Future experiments such as comprehensive analysis of new RNA-seq datasets generated from *nhr-23(ΔLCS)*, *let-7(scRORE)*, and *let-7* family mutants employing the bioinformatic strategies presented in this study

would be exciting, as it would allow for the identification of specific CCGs that drive the transitions between different phases of molting among the ~3700 genes that cycle in expression throughout larval development (*Hendriks et al., 2014*; *Kim et al., 2013*; *Meeuse et al., 2020*).

## *let-7*-mediated dampening of *nhr-23* levels sets the number of oscillations

All species of nematodes studied to date, including numerous species that are parasitic in mammals, molt 4 and only four times. However, the fundamental basis of this seemingly invariant limitation among nematodes is not known, despite the long-standing recognition of supernumerary molts associated with loss-of-function mutations in *let-7* and other heterochronic genes. We propose that the balance between the activity of NHR-23 and the activity of the *let-7*s controls the finite number of molts. In our model, NHR-23 is a positive effector and the *let-7*s are negative regulators of molting. Consistent with this model, both *nhr-23(gf)* and *let-7(lf)* mutants undergo extra molts. As larvae develop from one stage to the next, the amplitude of NHR-23 expression gradually declines, while the levels of the *let-7s* gradually increase, culminating in the extinction of the cycle in adulthood (*Figure 11B*). We have shown that *let-7*-mediated repression is at least partially responsible for the dampening of *nhr-23* expression (*Figure 6A*, *Figure 6—figure supplement 1*). Gradual reduction of positive autoregulation by NHR-23 through successive larval stages may be a second factor that could contribute to the dampening of *nhr-23* expression. Additional factors that function in the cessation of the molting cycle, perhaps through dampening of *nhr-23* expression or via an independent pathway, may include the transcription factors LIN-29 and MAB-10 (*Harris and Horvitz, 2011*; *Hayes et al., 2006*).

## NHR-23 and *let-7* act together with other feedback loops to regulate developmental timing by a possibly conserved mechanism

We have shown that NHR-23 and *let-7* are key components of a biological clock that regulates the pace of molting. Previous studies have characterized ~3700 genes that exhibit oscillatory patterns of gene expression coupled to the molting cycle (*Hendriks et al., 2014*; *Kim et al., 2013*; *Meeuse et al., 2020*). These ~3700 genes have been proposed to form a massive genetic oscillator that could act as a developmental clock during *C. elegans* development (*Meeuse et al., 2020*; *Tsiairis and Großhans, 2021*). The interdependence of the molting cycle and the proposed developmental clock remain unknown (*Tsiairis and Großhans, 2021*). The extent to which the pace of the molting cycle timer sets the pace of the theoretical developmental clock, and vice versa, is unclear. However, given the significant coupling of the molting cycle and the proposed developmental clock, the NHR-23–*let-7* genetic oscillator may contribute to the pace of the *C. elegans* developmental clock.

Using transgenic reporters, we showed that the transcriptional activation of *let-7* by NHR-23 and the post-transcriptional inhibition of *nhr-23* by *let-7* may be conserved in the context of mammalian homologs (*Figure 10*). RORs and mammalian *let-7* both regulate the expression of key clock components in the hepatic circadian clock. Specifically, RORs promote the expression of the clock components *Bmal1* and *Cry1*, as well as CCGs *Elovl3* and *Cyp8b1*, in both the livers of mice and cultured human liver cell lines (*Takeda et al., 2012*; *Zhang et al., 2017*). Liver-specific genetic disruption of only RORγ, or in combination with RORα, alters the levels of serum cholesterol, HDL and LDL, and liver triglycerides relative to wild-type mice (*Takeda et al., 2014*; *Zhang et al., 2017*). Thus, our work on the molting cycle timer may have implications for mammalian circadian clocks and related disorders of sleep and metabolism.

Our findings are consistent with the emerging concept that microRNA-mediated feedback loops increase the robustness of numerous gene regulatory networks and related outcomes, including cell fate decisions, stress responses, and developmental trajectories. The NHR-23–*let-7* genetic oscillator integrates the molting cycle timer with the heterochronic pathway in *C. elegans*, representing an elegant and possibly conserved mechanism of regulating developmental timing.

## Materials and methods
### Key Resources Table
Please see Appendix 1—key resources table.

## Working with *C. elegans*

*C. elegans* strains used in this study are described in the Key Resources Table. *C. elegans* were cultivated, preserved, observed, and transformed using standard methods (*Stiernagle, 2006*). Strains were grown at 25°C unless otherwise specified. Newly hatched worms were developmentally synchronized by passage through starvation-induced, L1-stage diapause. Briefly, eggs were isolated by lysis of gravid hermaphrodites in sodium hypochlorite, suspended in M9 buffer supplemented with 5 µg/ml cholesterol, and incubated for 16–24 h with rotational aeration. Hatchlings were then plated on solid nematode growth medium (NGM) seeded with *E. coli* strain OP50-1, HT115(DE3), or HB101, as indicated. One to two hundred hatchlings were routinely plated on 6-cm NGM plates; ten to fifteen thousand hatchlings, on 10-cm NGM plates seeded with 10-fold concentrated bacteria; 25,000 hatchlings on 15-cm NGM plates seeded with 10-fold concentrated bacteria.

## Bacteria-mediated feeding RNA-interference

*E. coli* HT115(DE3) transformed with the relevant RNA-interference (RNAi) clones from the Ahringer *C. elegans* genome RNAi library (*Kamath and Ahringer, 2003*) were cultured, plated on solid NGM supplemented with 8-mM isopropyl β-D-1-thiogalactopyranoside (IPTG, Laguna Scientific), and incubated for 16–24 h at 25° C, allowing for IPTG-induced expression of dsRNAs. Worms used as controls were fed bacteria transformed with the empty vector pPD129.36 (a gift from Andy Fire, Stanford University). Because the *nhr-23* RNAi clone corresponds to three constitutive exons at the 3′ end of the *nhr-23*, the dsRNA made by this clone targets all six isoforms of *nhr-23* annotated in WS273.

To knockdown *nhr-23* during a specific larval stage and circumvent larval arrest during a preceding molt, hatchlings were fed control bacteria for a certain interval as elaborated below, harvested, washed three times in M9 buffer, and then divided into two samples. Next, larvae in the control sample were fed bacteria transformed with the empty vector; larvae in the test sample were fed bacteria that expressed *nhr-23* dsRNAs (*Kamath and Ahringer, 2003*). Hatchlings destined to become test subjects in longitudinal studies of newly emerged L2s, L3s, and L4s (*Figures 2B, C and 4F*, *Supplementary file 1*) were initially fed control bacteria for 0, 6, and 14 h, respectively. To collect large time samples of synchronized larvae for RT-qPCR experiments, hatchlings were initially fed control bacteria for 16 h (L2, L3, and L4) and 24 h (L3, L4, and adults) and then split into test and control samples, as above (*Figure 3D–G*).

## Longitudinal studies of molting-associated biorhythms

This section provides additional information about the collection, analysis, and presentation of data in *Figures 2, 4 and 6* and *Supplementary file 1*. Cohorts of larvae molting to the stage of interest were isolated from synchronized populations, singled in 12-well NGM-RNAi plates, and observed for 5–60 s at regular 1 h intervals, using a Zeiss M²BioDiscovery microscope. L4s and older worms were observed at 300-fold magnification; L3s and younger worms, at 600-fold magnification. At each time sample, each subject was classified as active or lethargic based on the observation of defined target behaviors. Molting-defective (Mlt) and ruptured through the vulva (Rup) worms were identified by conventional criteria (*Reinhart et al., 2000*).

The longitudinal studies represented in *Figure 2* included video recording the head of the worm using a Sony HDR-XR500V or Nikon D500 camera attached to the microscope. Later, the number of pharyngeal contractions (pumps) in a 15 s recorded interval was counted while the video recording was viewed at 4-fold reduced speed using iMovie version 10.11.2. Pumping rates (Hz) determined by three independent counts of selected video recordings fell within 95% of the mean, validating this method. High, medium, and low levels of activity were then graded post hoc on a one-way standard scale defined by the SDs and mean pumping rate of all age-matched, wild-type time samples. As an example, wild-type young adults pumped at 3.9 ± 1.1 Hz (mean ± SD). The activity levels of nearly all worms that reawakened from lethargi associated with the L4-to-adult molt were therefore graded as high, medium, or low if the worm pumped at greater than, or equal to, 2.8 Hz, between 2.8 and 1.7 Hz, or less than 1.7 Hz, respectively. A reasonable exception to this system was made if sinusoidal locomotion was obvious but no pharyngeal pumps were captured on video. In this scenario, the worm was scored as active at a low level. This exception applied to only 8 out of 56 time samples of *nhr-23* single knockdowns and 14 out of 84 time samples of *nhr-23(RNAi) let-7(n2853)* double mutants. Among animals that reawakened from lethargi associated with the L3-to-L4 molt, the same exception

applied to 20 out of 120 time samples of *nhr-23* single knockdowns and 20 out of 180 time samples of *nhr-23(RNAi) let-7(n2853)* double mutants. The longitudinal studies represented in *Figures 3 and 6* did not involve video recordings. Instead, high versus low levels of activity were assigned based on the direct observation of continuous versus sporadic pharyngeal pumps during the time sample.

## Detection and characterization of supernumerary lethargi and molts

To score quiescence among populations of young adults, synchronized hatchlings were released from starvation-induced diapause by plating on 10-fold concentrated lawns of *E. coli* OP50-1 at a density of 200–400 worms per 10-cm NGM plate. For each strain of interest, six distinct clutches were plated at 12 h intervals, facilitating the later evaluation of time samples covering a 72 h interval. As described, worms were observed by light microscopy and scored as quiescent or active at regular 3 h intervals, 54 to 120 h post-release from diapause. For related longitudinal studies, quiescent adults were selected and singled in 12-well NGM plates seeded with thin lawns of bacteria. Each cohort of animals was then either observed at regular 2 h intervals (*Figure 7*), or video-recorded for 15–30 s, at regular 2 h intervals, with a Nikon D500 camera. All the previously described scoring rubrics were applied. In addition, ecdysis was recognized by the execution of one or more of the following idiosyncratic movements: rotation on the long axis (flipping), bilateral contraction and relaxation on the long axis, and elevation plus semicircular rotation of the head. Aberrant molts were scored based on the observation of puckered sections of cuticle along the body, or the adherence of partly shed cuticle fragments to the body. If a particular animal had passed through lethargus, then the following behaviors were also considered evidence of an aberrant molt: pharyngeal spasms, incomplete pumps wherein the grinder failed to close, and incomplete flips that resulted in twists or kinks along the body. Detection of a shed cuticle, or parts thereof, on the culture plate was recorded separately. The latter categories were not mutually exclusive. The absence of a supernumerary molt was inferred if the animal was active and superficially normal at the endpoint.

## Construction of fusion genes and transgenic strains

The sequences of all oligonucleotides used in this study are specified in *Supplementary file 5*. All DNA oligonucleotides were from Integrated DNA Technologies (IDT). The bicistronic reporters used to detect regulatory elements within 3′ UTRs were constructed by Gibson Assembly (NEB) using standard methods. Phusion High-Fidelity DNA Polymerase (NEB) was used for PCR amplifications. The resulting plasmids contained the pBR322 backbone of Fire Lab vectors; the *dpy-7* promoter, which corresponds to nucleotides 7,537,914–7,538,219 of *C. elegans* Chr. X (NC_003284); the synthetic intron embedded in primer HM01; the coding sequence for *tandem (td) tomato*, which was isolated from Addgene plasmid #30,530 (a gift from Gerhart Ryffel); one of the test 3′ UTRs described below; and an *SL2::gfp::unc-54* 3′ UTR cassette. The gene-specific 3′ UTRs from *C. elegans* comprised nucleotides amplified from Chr. I (NC_003279) as follows: *nhr-23*, 7,220,953–7,221,820; *unc-54*, 14,855,909–14,856,180; *lin-41*, 9,334,850–9,335,964 (*Mangone et al., 2010*). According to *Roach et al., 2020*, the longest 3′ UTR of *nhr-23* may extend to 7,221,835. But as the additional 16 nucleotides do not contain a putative LCS, the shorter 3′ UTR was utilized. Deletions within the *nhr-23* 3′ UTR reporter (cloned in pHR017) were created using a Q5 Site-Directed Mutagenesis Kit (NEB) and verified by Sanger sequencing (Genewiz Inc). Additionally, a 565-bp fragment of the 3′ UTR of *H. sapiens RORβ* (chr9:74689171–74689705; GRCh38/hg38), a 256-bp fragment of the 3′ UTR of *M. musculus RORα* (chr9:69380941–69381196 GRCm39/mm39), as well as derivatives lacking both LCSs, were ordered as gBlocks Gene Fragments from IDT and fused to *tdTomato*. To generate distinct extrachromosomal arrays harboring each bicistronic reporter, mixtures of the corresponding plasmid (1 ng/µl), the cotransformation marker *ttx-3::gfp* (40 ng/µl), and filler DNA pRS316 (59 ng/µl) were microinjected into the gonads of wild-type hermaphrodites. Transgenic progeny and unique descendent strains were isolated by standard methods.

A transcriptional reporter for *M. musculus Mirlet7a-1* was generated by using fusion PCR to combine the 3000-bp region upstream of the mature *let-7-a-1* microRNA (chr13:48538273–48541272; GRCm38/mm10) with *gfp* (pPD95.75). The resulting PCR product was first cloned into the Topo vector pCR-Blunt-II-Topo (Thermo Fisher Scientific) to generate pRA46. The strain ARF431 was generated by coinjecting pRA46 (1 ng/µl), *ttx-3*::*gfp* (40 ng/µl, and pRS316 [59 ng/µl] into wild-type hermaphrodites).

The strain ARF422 was made by first crossing *wgIs43* hermaphrodites with *let-7(mg279) mir-84(tm1304)* males. After singling F2's from the crosses, we screened for *wgIs43* homozygotes among the F3 generation. Only the strains that were homozygous for *wgIs43* were selected and screened for *let-7(mg279); mir-84(tm1304)* homozygotes. The transgene *wgIs43* was obtained from OP43 and *let-7(mg279); mir-84(tm1304)* was obtained from ARF249.

The strain ARF432 was generated by injecting construct 4271 (*Kostrouchova et al., 1998*) at a concentration of 5 ng/µl, together with the coinjection marker *ttx-3::gfp* (40 ng/µl), and pRS316 (45 ng/µl) into wild-type hermaphrodites. Transgenic lines were isolated by standard methods.

## CRISPR/Cas9-mediated editing of *C. elegans* genes

The CRISPR/Cas9 system was used essentially as described (*Paix et al., 2015*) to delete the endogenous LCS from the 3′ UTR of *nhr-23*, generating the allele *nhr-23(aaa20)*. Briefly, wild-type hermaphrodites were microinjected with a mixture containing the following: *nhr-23* crRNA (400 ng/µl), tracrRNA (1 µg/µl), *dpy-10* crRNA (160 ng/µl, GE Dharmacon), *dpy-10* ssODN (13.75 ng/µl, IDT), and CAS9 protein (500 ng/µl, PNA Bio) in HEPES (4-(2-hydroxyethyl)-1-piperazineethanesulfonic acid) buffer pH 7.5 (Sigma-Aldrich) supplemented with 0.025 µM KCl (Sigma-Aldrich). Injected hermaphrodites (P0s) were singled and screened for Dumpy (Dpy) or Roller (Rol) offspring (F1s), both phenotypes associated with mutations in *dpy-10*. One hundred F1s were singled from a selected P0. Genotyping the F1s and their descendants (F2s) identified two strains homozygous for identical chromosomal deletions of precisely the 21 nucleotides comprising the LCS. One *nhr-23(aaa20-ΔLCS)* strain was backcrossed to N2 three times prior to phenotypic analysis. No edits in the *dpy-10* gene were found in the backcrossed strain (ARF414).

To construct *xk22*, wild-type hermaphrodites were injected with *nhr-23* crRNA oHG202 (40 µM, IDT Alt-R CRISPR crRNA), *nhr-23::3xflag* repair template (120 ng/µl, IDT Ultramer DNA oligo), *dpy-10* crRNA (5.6 µM, IDT Alt-R CRISPR crRNA), *dpy-10* repair template (12 ng/µl, IDT Ultramer DNA oligo), tracrRNA (40 µM, IDT Alt-R CRISPR-Cas9 tracrRNA), and Cas9 (15.5 µM, stock at 40 µM in 20 mM HEPES-KOH pH 7.5, 150 mM KCl, 10% glycerol, 1 mM DTT [Dithiothreitol] from Berkeley QB3 MacroLab). All reagents were diluted in IDT duplex buffer. The crRNA and repair template both target sequences encoding the C-terminus of NHR-23, which is common to all predicted isoforms. Injected hermaphrodites were singled and F1 offspring were screened for the same phenotypes described above. One hundred and twenty F1s were singled from plates that had a high penetrance of Dpy and Rol phenotypes. Genotyping the F1s identified three lines that had *3xflag* inserted precisely before the stop codon of the *nhr-23* gene. One *nhr-23(xk22)* line was backcrossed to N2 five times to generate QK159. No edits in the *dpy-10* gene were found in QK159.

To construct the *let-7(scRORE1,2)* and *let-7(scRORE1,3)* strains, the ROREs were serially scrambled. *let-7(scRORE1)* was first made by injecting wild-type hermaphrodites with crRNA oHG287(40 µM, IDT Alt-R CRISPR crRNA), repair template oHG293 (120 ng/µl, IDT Ultramer DNA oligo), and other components as described above. Injected hermaphrodites were singled and F1 offspring were screened for the same phenotypes described above. Genotyping the F1s identified several lines that had *RORE1* scrambled in the *let-7* promoter (*let-7(scRORE1)*). The *dpy-10* mutation was outcrossed from one line. To construct *let-7(scRORE1,2)*, this *let-7(scRORE1)* line was then injected with crRNA oHG282 (40 µM, IDT Alt-R CRISPR crRNA), repair template oHG367 (120 ng/µl, IDT Ultramer DNA oligo), and other components as described above. Injected hermaphrodites were singled and F1 offspring were screened for the same phenotypes described above. Genotyping the F1s identified three lines that had *RORE1* and *RORE2* scrambled in the *let-7* promoter. These lines were backcrossed to N2 three times to generate three independent *let-7(scRORE1,2)* lines: QK201 [*let-7(xk41)*], Q202 [*let-7(xk43)*], and QK203 [*let-7(xk44)*]. To construct *let-7(scRORE1,3)*, *let-7(scRORE1)* was injected with crRNA oHG278 (40 µM, IDT Alt-R CRISPR crRNA), repair template oHG291 (120 ng/µl, IDT Ultramer DNA oligo), and other components as described above. Injected hermaphrodites were singled and F1 offspring were screened for the same phenotypes described above. Genotyping the F1s identified two lines that had *RORE1* and *RORE3* scrambled in the *let-7* promoter. These lines were backcrossed to N2 three times to generate two independent *let-7(scRORE1,3)* lines: QK198 [*let-7(xk39)*] and QK199 [*let-7(xk42)*].

## Quantitative fluorescence microscopy

*C. elegans* were anesthetized with 2.5% NaN₃ (vol/vol) in M9 buffer, mounted on 2% agarose pads, and observed using a Zeiss Axioplan compound microscope with an attached Hamamatsu Orca ER CCD camera. The image acquisition and analysis software package Volocity 6.3 (Perkin Elmer) were used to control the microscope and digital camera and also to measure average fluorescence intensities within selected regions of interest (ROIs). In particular experiments, transgenic animals were staged partly by DIC microscopy and imaged during the L3-to-L4 or L4-to-adult molts. Molting animals were identified by occlusion of the buccal cavity (*Monsalve et al., 2011*). Stereotypical rearrangements of vulva precursor cells demarcated early versus late substages of the L3-to-L4 molt. The presence of a lumen in the incipient vulva demarcated early versus late substages of the L4-to-adult molt (*Gupta et al., 2012*; *Van Buskirk and Sternberg, 2007*).

To measure GFP signals associated with the both the *C. elegans let-7p::nls-gfp* transcriptional reporter (*Kai et al., 2013*) and the *M. musculus let-7p::gfp* reporter, worms were imaged at ×400 total magnification. For the *C. elegans let-7* transcriptional reporter, both DIC and fluorescence images of the lateral epidermis were acquired – the latter with an exposure time of 25 ms. Three nuclei in hyp7 and three in the seam were traced from the DIC image of each worm. The average fluorescence intensity within each nucleus was then measured and corrected for background signal. The average values for both hyp7 and seam nuclei (per worm) were used in further statistical analysis. For the *M. musculus let-7* transcriptional reporter, the pharynx was imaged in both the DIC and fluorescence channels. An exposure time of 200 ms was used to capture the GFP signal. Three ROIs in the pharynx were traced from the DIC image of each worm. As stated above, the average fluorescence intensity within each traced ROI was measured, corrected for background signal and used in further statistical analysis.

Signals associated with tdTomato and GFP expressed from bicistronic reporters for regulatory elements within 3′ UTRs were measured using similar approaches. In this case, three distinct ROIs with areas of 40–70 µm² were manually selected per worm; each ROI included approximately equal areas of the nucleus and cytoplasm. In addition, multiple images of tdTomato and GFP were automatically captured over a range of exposure times. The average fluorescence intensity of each ROI was measured and plotted versus the exposure time. Values within the linear range of the assay were then used to determine the ratiometric signal (tdTomato/GFP) for each ROI. The average ratiometric value of all three ROIs per worm was used for subsequent statistical analysis. Notably, the morphology of the vulva was abnormal in a subset (≤10%) of animals that expressed any bicistronic reporter. Because the phenotype precluded staging by the abovementioned criteria, this subset of animals was excluded from the analysis.

Measurement and analysis of the GFP signal from the NHR-23::GFP reporter were done exactly as described for the *C. elegans let-7::nls-gfp* reporter above, except that an exposure time of 200 ms was used to capture the fluorescence signal.

## Chromatin immunoprecipitation coupled with quantitative PCR (ChIP-qPCR)

Animals grown at 25°C were collected as a ~500 µl packed pellet in M9. The animals were nutated for 30 min at room temperature in 12 ml of 2.6% (vol/vol) formaldehyde in autoclaved DI water for live crosslinking. To quench the reaction, 600 µl of 2.5 M glycine was added and the worms incubated on the nutator for another 5 min. The samples were then washed three times in water and flash-frozen in liquid nitrogen. Frozen pellets were ground twice, for 1 min each, in a Retsch MM400 CryoMill at 30 Hz in liquid nitrogen-chilled stainless steel cryomill chambers, producing a frozen powder of partially lysed worms. The powder was resuspended and further lysed in 2 ml of RIPA (radioimmunoprecipitation assay) buffer (1× phosphate-buffered saline [PBS], 1% [vol/vol] NP40, 0.5% sodium deoxycholate, and 0.1% sodium dodecyl sulfate [SDS]), supplemented with the HALT Protease and Phosphatase Inhibitor Cocktail (Thermo Fisher Scientific), for 10 min at 4°C. To shear the chromatin, samples were sonicated in a Bioruptor Pico (Diagenode) for 3 min (30 s ON/30 s OFF cycles), three times, at 4°C. A 20 µl aliquot of the sample was treated with Proteinase K for 10 min and then subjected to phenol chloroform extraction, as described below. The concentration of the aliquot was determined using a Qubit Fluorometer 3.0 (Invitrogen). Based on the initial concentration of the aliquot, the chromatin sample was diluted to 20–30 ng/µl. To check the extent of shearing, the same aliquot was run on an agarose gel. The sample was processed and analyzed further, provided that

the DNA smear centered around 200 bp. Of the total amount of chromatin that remained, 10% was used as the input sample (stored at 4°C) and 90% was subject to immunoprecipitation. Every 10 μg of chromatin was incubated with 2 μg of mouse M2 anti-FLAG monoclonal antibodies (Sigma-Aldrich) overnight at 4°C on a nutator. Next, samples were incubated with 1.5 mg of affinity-purified sheep anti-mouse IgG antibodies covalently attached to superparamagnetic Dynabeads M-280 (Invitrogen) for 2 h at 4°C. Thereafter, complexes bound to the beads were separated three times from the super-natant and washed in 800 μl LiCl buffer (100 mM Tris–HCl pH 7.5, 500 mM LiCl, 1% [vol/vol] NP40, and 1% sodium deoxycholate). The resulting immunoprecipitates were de-crosslinked by incubation with 80 μg of Proteinase K in 400 μl of worm lysis buffer (100 mM Tris–HCl pH 7.5, 100 mM NaCl, 50 mM EDTA (ethylenediaminetetraacetic acid), and 1% SDS) at 65°C for 4 h; the input samples also underwent the same treatment in parallel. Residual proteins were removed from both ChIP and input samples by phenol–chloroform extraction. Briefly, 400 μl of phenol–chloroform–isoamyl alcohol pH 8.0 (Sigma-Aldrich) was added to each sample. The sample was vortexed vigorously and centrifuged at 15,000 × $g$ for 5 min at 4°C. The top layer was transferred to a new tube and DNA was precipitated by incubating with 1 ml of 0.3 M ammonium acetate (Sigma-Aldrich) in ethanol for 1 h at −30°C. The resulting DNA pellet was washed twice in 100% ethanol and resuspended in Tris–EDTA, pH 8.0. Prior to use as a template for qPCR, the entire DNA sample was treated with RNase A for 1 h at 37°C.

Quantitative PCR for promoter regions of interest was performed with Absolute Blue SYBR Green (Thermo Scientific) using a CFX96 Real Time System Thermocycler (BioRad) as per the manufacturers' instructions, with custom primers described in *Supplementary file 5*. The Ct value for each IP sample was first normalized to the Ct value for the respective input sample. The log 2 transformed fold-change values for samples derived from QK159[*nhr-23::3xflag*] were then normalized to the respective N2 sample. Three biological replicates, each with two technical replicates, were completed for each amplicon of interest, as specified in corresponding figure legends. Pairwise statistical comparisons of the fold enrichment of a given amplicon in samples from QK159[*nhr-23::3xflag*] versus N2 were made by two-way ANOVA with Bonferroni's correction for multiple comparisons.

For L3 ChIP-qPCR samples, hypochlorite prepped embryos were directly plated on HB101 and animals were collected after 29 h at 25°C as a semi-synchronous population. For L4 ChIP-qPCR samples in *Figure 3C* and *Figure 3—figure supplement 1D*, hypochlorite prepped embryos were nutated in M9 buffer for 24 h. L1 diapause worms were plated on HB101 and collected after 32 h at 25°C as a synchronous population of mid-L4 worms. For L4 ChIP-qPCR samples in *Figure 4B* and *Figure 4—figure supplement 1*, hypochlorite prepped embryos were directly plated on HB101 and collected after 35 h at 25°C as a semisynchronous population.

## Isolation of RNA

RNA was extracted from developmentally synchronized *C. elegans* as described (*McCulloch and Rougvie, 2014*). Samples of ~1500 worms were collected at regular 2 h intervals. Because the strains seemed to develop at different rates, light microscopy was used to count the fraction of pumping (active) versus nonpumping (lethargic) animals in each sample prior to collection ($n$ = 50–100). Lethargic phases were empirically identified post hoc by troughs in the proportion of pumping animals. Related graphs in *Figure 6A*, *Figure 8B, C* and *Figure 6—figure supplement 1A* include 14 time samples encompassing three lethargic and two active phases per strain. Pellets containing worms (~100 μl) were re-suspended in 4 volumes of TRIzol (Thermo Fisher Scientific) and 1 volume of glass beads 400–625 μm in diameter (Sigma). The suspensions were vortexed, flash frozen, and thawed thrice. Samples were then mixed with 0.17 volumes of 24:1 chloroform:isoamyl alcohol (OmniPur) and centrifuged. The aqueous layer was collected, mixed with an equal volume of 5:1 acid phenol:chlo-roform (Thermo Fisher Scientific), and centrifuged again. After collection of the top layer, RNA was extracted by precipitation with ice-cold isopropanol (Sigma) and GlycoBlue (Thermo Fisher Scientific). The concentration of RNA in each time sample was measured using a NanoDrop 2000 (Thermo Fisher Scientific). Thereafter, 5 μg of total RNA per sample was treated with 2 U of TURBO DNase (Thermo Fisher Scientific) for 1 h.

## Quantitative RT-PCR

The sequences of gene-specific RT primers and identifiers for TaqMan assays used in this research are provided in *Supplementary file 5*. To quantify levels of primary *let-7* and *ama-1* transcripts in the

abovementioned extracts, we processed 50 ng of RNA using a High-Capacity cDNA Reverse Transcription Kit (Thermo Fisher Scientific). Reaction mixtures of 15 µl included random primers, dNTPs, RNaseOUT, and reverse transcriptase, per the manufacturer's guidelines. To quantify levels of mature *let-7* and the U18 small nucleolar RNA (snoRNA), we processed RNA with the same kit but used gene-specific rather than random primers. Three volumes of nuclease-free water were added to completed RT reactions. Next, we set up TaqMan assays (Thermo Fisher Scientific) in 96-well plates, in triplicate. Per the manufacturer's instructions, each reaction included TaqMan Universal PCR Master Mix, no AmpErase UNG, gene-specific primers, and 1.3 µl of the preceding RT product in a volume of 20 µl. Reactions ran on a Stratagene MX3000P (Agilent Genomics). To measure levels of protein-coding transcripts, 1 µg of RNA was reverse transcribed using the enzyme Transcriptor (Roche). Each reaction mixture (20 µl) also included hexadeoxynucleotide primers (Promega), dNTPs and RNasin (Promega). Four volumes of nuclease-free water were added to completed RT reactions. TaqMan assays were performed as described using 2 µl of the RT product as template in a volume of 10 µl.

The amount of template used in each TaqMan assay gave Ct values in the linear range of 21–36. In nearly all cases, technical replicates gave Ct values within 95% of the mean and the mean Ct value was used in subsequent analyses. Separate TaqMan reactions using templates made in the absence of reverse transcriptase produced no detectable PCR products, confirming the amplification of RNA rather than genomic DNA. As described, the levels of transcripts of interest were normalized to the levels of *ama-1* mRNAs or U18 snoRNAs within each sample, which were quantified in parallel TaqMan assays. For studies of gene expression over several developmental stages, the normalized values for each time sample were further standardized to the mean of all time samples derived from mock-treated or wild-type animals.

## RNA extraction and RT-qPCR for *let-7(scRORE)* mutants

Hypochlorite prepped embryos were synchronized and plated on HB101. The development was tracked by monitoring pharyngeal pumping as described above. Samples of ~8000 worms were collected every 2 h, starting at 18 h after plating at 25°C in TRIzol (Thermo Fisher Scientific). Following three freeze–thaw cycles, 1-bromo-3-chloropropane was added and the RNA in the resulting aqueous phase was precipitated by incubating with isopropanol for 2 h at −30°C. Samples were then spun at 21,000 × *g* for 30 min at 4°C to pellet the RNA. The pellet was washed with 75% ethanol thrice and then resuspended in water. cDNA synthesis for primary *let-7* was done using SuperScript III Reverse Transcriptase (Invitrogen). 250 ng of RNA was used for cDNA synthesis in the Eppendorf Mastercycler Pro S6325. Quantitative PCR for *pri-let-7* and *eft-2* was performed with Absolute Blue SYBR Green (Thermo Scientific) on the CFX63 Real Time System Thermocyclers (BioRad) using custom primers as listed in *Supplementary file 5*. The cycle numbers for *pri-let-7* were normalized to respective cycle numbers for *eft-2*. Two biological replicates with two technical replicates were done. The values were all normalized to the average of the four readings for the N2 sample. Two-tailed Student's *t*-test was done to evaluate p values. RT-qPCR for *mlt-10* was used to validate the age-matched synchronous populations across the genotypes.

TaqMan synthesis for mature *let-7* was done using probes synthesized by Applied Biosystems. 100 ng of RNA was used for TaqMan Synthesis using High capacity cDNA Reverse Transcription Kit (Thermo Fisher Scientific). Quantitative PCR for *let-7* and *U18* was performed using TaqMan Universal Master Mix, No AmpErase UNG (Thermo Fisher Scientific) on the CFX63 Real Time System Thermocyclers (BioRad). The cycle numbers for *let-7* were normalized to respective cycle numbers for U18. Two biological replicates with two technical replicates were done. The values were all normalized to the average of the four readings for the N2 sample. Two-tailed Student's *t*-test was done to evaluate p values.

## Counting seam cell nuclei

Hypochlorite prepped embryos were nutated and hatched over 24 h in M9 buffer. L1 diapause worms were plated on HB101 at 25°C. Animals were scored between 40 and 44 h after plating. Worms were immobilized in 50 mg/ml levamisole on a 2% agarose pad on a slide. The number of Pscm::GFP expressing cells in each worm was counted under the Zeiss Axio Zoom V16 Fluorescence Stereo Scope.

## MetaCycle analysis of gene expression curves

The MetaCycle 1.2.0 package was used to calculate the amplitude and phase of expression of the genes listed in *Figure 6*, *Figure 8*, *Figure 9*, *Figure 6—figure supplement 1A*, *Figure 8—figure supplement 1*, *Figure 9—figure supplement 1*, and *Supplementary file 3*. The normalized levels of transcripts of each gene, derived from the analysis described above were provided to MetaCycle. For each gene, the expression curves recorded from the L2-to-L3 molt until the L3-to-L4 molt, were considered as corresponding to the L3 stage. Similarly, expression curves recorded from the L3-to-L4 molt until the L4-to-adult molt were considered as corresponding to the L4 stage. Gene expression curves recorded in the L3 stage were analyzed separately from those recorded during the L4 stage. Additionally, expected periods of 8 and 10 h were used for analysis of the L3- and L4-stage data, respectively.

## Identification of conserved *cis*-regulatory elements in homologous genes

DNA sequences corresponding to the upstream regulatory region, first intron and 3′ UTR for each nematode gene of interest were retrieved from WormBase v.264 (WS264) and saved as SnapGene v.4 (GSL Biotech) files. The upstream sequences extracted from WormBase included all nucleotides between the transcriptional start site of the gene of interest and the nearest protein-coding gene. Particular sequences were extended or shortened based on gene models, ESTs, and transcriptional start sites archived in WS264. If the gene of interest lacked an annotated 3′ UTR, then we initially retrieved 1 kb of sequence downstream of the stop codon. Particular 3′ UTR sequences were revised based on ESTs and polyA sites that are archived in WS264 but not yet incorporated in current gene models.

Both the upstream regulatory regions of vertebrate homologs of *let-7* and the 3′ UTRs of vertebrate homologs of *nhr-23/RORs* were retrieved from the UCSC genome browser. Three human genes, two mouse genes, and six zebrafish genes encode mature miRNAs identical in sequence to *C. elegans let-7*. We extracted 3 kb of sequence upstream of each *let-7* homolog, except in the case of *H. sapiens let-7a-3*, wherein the promoter has been experimentally delimited to 1 kb of upstream sequence (*Wang et al., 2012*). For a given gene, the longest 3′ UTR was selected if multiple 3′ UTRs existed. The 3′ UTR sequences were individually and systematically validated by comparison with EST; only those genes with annotated 3′ UTRs supported by ESTs were included in further analyses.

## Finding CCGs regulated by NHR-23 and *let-7s*

Genes were determined to be 'involved in molting' based on the literature. For example, if mutations in a particular gene caused a molting-defective phenotype, the gene was considered to be involved in molting (*Frand et al., 2005*). Similarly, if inactivation of the gene had an effect on lethargus, the gene was also considered to be involved in the molting cycle. Genes were annotated as 'oscillatory' based on published RNA-Seq studies (*Hendriks et al., 2014*; *Kim et al., 2013*); therein, genes whose expression at 8 to 10 h intervals was significantly correlated ($p < 0.05$) were considered to be cycling in expression.

To identify ROR response elements that might function as transcriptional enhancers of miRNAs or protein-coding genes of interest, we searched the upstream regulatory sequences and/or first introns for instances of the consensus response element 5′-(A/G)GGTCA-3′ on both the coding and anti-coding strands of DNA. *Figures 3A and 8A* and *Figure 3—figure supplement 1A, B* depict the results of these computational searches. To accurately calculate the probability of an RORE occurring by chance, we first used the k-mer counting software program DSK (*Rizk et al., 2013*) to find that the reference genome of *C. elegans*, which comprises 100.2 mega bases, includes 41,203 distinct instances of the consensus RORE. For non-nematodes, the expected frequency was the chance of either six-nucleotide sequence appearing in a longer oligonucleotide; this frequency is approximately 1 per 1 kb.

Regions of *C. elegans* chromosomal DNA occupied by NHR-23 in vivo were identified on the modENCODE *C. elegans* Genome Browser (v. 2.48). The two relevant datasets archived therein were ChIP-Seq of strain OP43 cultivated at 20°C and harvested during the L2 or L3 stage. Most genomic regions where NHR-23 binding was significantly enriched were detected in the dataset collected from L3-stage larvae, however, we do not discriminate between the two stages in our analysis. The

upstream regulatory sequences and/or first intron for each gene of interest were viewed in this browser. Regions of significant enrichment ('peaks') were identified by Z-scores ≥2 (*Celniker et al., 2009*; *Gerstein et al., 2010*). Sequences extracted and aligned with the upstream regulatory regions and/or first intron as above, adjusting for differences in the related chromosomal coordinates between WS220 and WS264.

Evidence of direct or indirect regulation of transcript levels by NHR-23 – that is, expression of the gene was at least 1.2-fold reduced in *nhr-23(RNAi)* versus control larvae – was either detected by Affymetrix microarrays (*Kouns et al., 2011*), or shown in prior publications (*lin-42a/b*, *nas-36*) (*Frand et al., 2005*; *Monsalve, 2013*).

Targets of NHR-23 followed two out of the three following criteria: (1) the upstream regulatory region and/or first intron contained ChIP-seq NHR-23 peaks (*Celniker et al., 2009*; *Gerstein et al., 2010*); (2) the same region contained more ROREs than predicted by chance alone; and (3) Expression was 1.2-fold lower in *nhr-23* knockdowns than mock-treated larvae.

The software RNAhybrid (*Rehmsmeier et al., 2004*) was used to detect sequences partially complementary to the 21 nt mature *let-7* in the 3′ UTRs of annotated homologs of *nhr-23* in the genomes of *H. sapiens*, *M. musculus*, *D. rerio*, and *C. briggsae*. Mature *C. elegans let-7*, which is identical to human *let-7a*, was used as the query sequence. No more than one mismatched nucleotide within the *let-7* seed sequence was tolerated for the prediction of LCSs in this report.

Targets of *let-7* fulfilled both of the following criteria: (1) LCSs, with up to one mismatch in the seed region, were detected in the 3′ UTR more often than, or equal to, the number predicted by chance alone (*Rehmsmeier et al., 2004*) and (2) ALG-1 co-IP the 3′ UTR, on the coding strand of the gene by iCLIP-Seq (*Broughton et al., 2016*).

## Quantification and statistical analyses

The software package Volocity 6.3 (Perkin Elmer) was used to both acquire fluorescence micrographs and measure the signal intensity of selected ROIs. The software package GraphPad Prism v6.0h was used for all statistical tests except for those done on data from ChIP-qPCR experiments. Statistical tests for the ChIP-qPCR experiments were done using R Studio version 1.1.463 and R version 3.5.2. The software package MetaCycle 1.2.0 was used to calculate the amplitude and phase of expression for the cycling genes. Samples sizes for all experiments, statistical analyses, and outcomes thereof are included within each figure and its legend.

## Acknowledgements

We thank former and current members of the Frand and Kim Labs for helpful suggestions. Ruhi Patel would like to thank Dr. Hilary Coller, Dr. Feng Guo, Dr. Tracy Johnson, and Dr. Steve Jacobsen for their continued support and excellent intellectual contributions to the manuscript. The American Cancer Society (RSG-12-149-01-DDC to ARF), the National Science Foundation (IOS1258218 to ARF), and the National Institutes of Health (R01 GM129301 awarded to JKK) supported this research. Some strains were provided by the *Caenorhabditis* Genetics Center (CGC), which is funded by the NIH Office of Research Infrastructure Programs (P40 OD010440). Himani Galagali and John K Kim dedicate this study to their colleague, Gregory G Fuller.

## Additional information

### Funding

| Funder | Grant reference number | Author |
| --- | --- | --- |
| American Cancer Society | RSG-12-149-01-DDC | Alison R Frand |
| National Science Foundation | IOS1258218 | Alison R Frand |
| National Institutes of Health | R01 GM129301 | John K Kim |

| Funder | Grant reference number | Author |
| --- | --- | --- |

The funders had no role in study design, data collection, and interpretation, or the decision to submit the work for publication.

## Author contributions
Ruhi Patel, Himani Galagali, Conceptualization, Resources, Data curation, Formal analysis, Validation, Investigation, Visualization, Methodology, Writing – original draft, Writing – review and editing; John K Kim, Conceptualization, Supervision, Funding acquisition, Visualization, Methodology, Project administration, Writing – review and editing; Alison R Frand, Conceptualization, Data curation, Formal analysis, Supervision, Funding acquisition, Validation, Investigation, Visualization, Methodology, Writing – original draft, Project administration, Writing – review and editing

## Author ORCIDs
Ruhi Patel http://orcid.org/0000-0002-8612-0016
Himani Galagali http://orcid.org/0000-0001-8983-6397
John K Kim http://orcid.org/0000-0001-9838-3254
Alison R Frand http://orcid.org/0000-0001-5972-989X

## Decision letter and Author response
Decision letter https://doi.org/10.7554/eLife.80010.sa1
Author response https://doi.org/10.7554/eLife.80010.sa2

# Additional files

## Supplementary files
• Supplementary file 1. Metrics of the molting biorhythm associated with specific genotypes. The active, lethargic, and wake-to-wake intervals are defined in the text. The values derived from longitudinal studies of stage-specific cohorts of singled, isogenic worms. The top row of each section corresponds to the same-day cohort of singled, wild-type worms. Dashes (–) beneath 'RNAi' indicate continuous cultivation of the worms on *E. coli* HT115(DE3). '*N*' is the cumulative sample size from two independent trials. All p values were generated by pairwise comparisons between individual metrics tabulated for a specific cohort of test subjects and also for the same-day, age-matched cohort of control subjects: ****p ≤ 0.0001, ***p ≤ 0.001, *p ≤ 0.05, ordinary one-way ANOVA with Bonferroni's correction for multiple comparisons. Entries in the top row of each subsection correspond to six distinct cohorts of control subjects. By order of first appearance in the table, the strains tested were N2, QK509 [*let-7(n2853)*], GR1395 [*mgIs49*], GR1436 [*let-7(mg279)*], ARF249 [*let-7(mg279); mir-84(tm1304)*], QK201 [*let-7(xk41)*], QK203 [*let-7(xk44)*], QK198 [*let-7(xk39)*], QK199 [*let-7(xk42)*], OP43 [*wgIs43*], ARF414 [*nhr-23(aaa20)*] and VT1066 [*mir-48 mir-241(nDf51); mir-84(n4037)*]. Notably, both QK509 [*let-7(n2853)*] and the ancestral strain MT7626 [*let-7(n2853)*] developed at an accelerated pace: 71% of QK059 hatchlings and 79% of MT7626 hatchlings transited the larval stages and emerged as young adults within 42 h of cultivation with food, as compared with 12% of N2 hatchlings (*N* = 100, p ≤ 0.0001, chi-square test).

• Supplementary file 2. LCSs found in selected nematode and vertebrate homologs of *ROR*. Entries correspond to sites shown in *Figure 10D*. The number of nucleotides between the 3' end of each LCS and the stop codon is indicated. The thermostability of every RNA duplex between a prospective LCS and mature *let-7*, as predicted by RNAhybrid, was lower than the predicted thermostability (−29 kcal/mol) of duplexes between the functional LCS in the 3' UTR of *lin-41* and *let-7* (*Rehmsmeier et al., 2004*). The 3' UTRs were supported by ESTs archived in WBcel235/ce11, WBPS9, GRCh38/hg38, GRCm38/mm10, and GRCz10/danRer10.

• Supplementary file 3. Metrics of the expression curves of *nhr-23* in the indicated genetic backgrounds. As described in Materials and methods, Metacycle was used to calculate the amplitude and phase of expression of the waveforms. The peak values and the slope of rise and decay were obtained by manual calculation.

• Supplementary file 4. Evaluation and classification of clock-controlled genes as direct targets of NHR-23, *let-7*s, neither, or both. The bioinformatic approaches and criteria for assignment of queries to categories are described in Materials and methods. The name and WormBase accession number of each gene are listed. '# Obs./# Exp.' stands for the number of observed DNA or RNA response elements divided by the number of elements predicted by chance alone. The down arrows denote

downregulation of the query transcript in *nhr-23(RNAi)* animals as compared with wild-type controls. The '+' symbol in column 12 denotes identification of the transcript in ALG-1 iCLIP datasets in vivo (***Broughton et al., 2016***). The symbol '〰' indicates that expression of the gene oscillates across larval development. Relevant datasets are identified in the text, Materials and Methods, and Key Resources Table.

• Supplementary file 5. Oligonucleotides used in this study. DNA or RNA sequences appear in the first column. For primers and gene blocks used to construct a particular bicistronic reporter for *cis*-regulatory elements in a 3′ UTR of interest, the resulting plasmid and corresponding extrachromosomal array are identified in the 'application' column. All seven reporters and respective transgenic strains of *C. elegans* are further described in The Key Resources Table.

• MDAR checklist

## Data availability

All data generated during this study are included in the manuscript. Additionally, the following datasets from previously published genome-wide studies were analyzed in the paper: NHR-23 L2 Stage ChIP-Seq - GSE46774 NHR-23 L3 Stage ChIP-Seq - GSE48709 nhr-23(RNAi) microarray - GSE32031 RNA-Seq of developing *C. elegans* larvae - GSE49043, GSE52910 ALG-1 iCLIP-Seq - SRA: SRP078361.

The following previously published datasets were used:

| Author(s) | Year | Dataset title | Dataset URL | Database and Identifier |
|---|---|---|---|---|
| Celniker SE | 2009 | Snyder_NHR-23_GFP_L2 | https://www.ncbi.nlm.nih.gov/geo/query/acc.cgi?acc=GSE46774 | NCBI Gene Expression Omnibus, GSE46774 |
| Celniker SE | 2009 | Identification of Transcription Factor NHR-23::GFP Binding Regions in L3 | https://www.ncbi.nlm.nih.gov/geo/query/acc.cgi?acc=GSE48709 | NCBI Gene Expression Omnibus, GSE48709 |
| Dh Kim, Grün D, van Oudenaarden A | 2013 | Dampening of expression oscillations by synchronous regulation of a microRNA and its target | https://www.ncbi.nlm.nih.gov/geo/query/acc.cgi?acc=GSE49043 | NCBI Gene Expression Omnibus, GSE49043 |
| Hendriks GJ, Gaidatzis D, Aeschimann F, Grosshans H | 2014 | Extensive oscillatory gene expression during *C. elegans* larval development | https://www.ncbi.nlm.nih.gov/geo/query/acc.cgi?acc=GSE52910 | NCBI Gene Expression Omnibus, GSE52910 |
| Broughton JP, Lovci MT, Huang JL, Yeo GW, Pasquinelli AE | 2016 | *C. elegans* ALG-1 iCLIP raw sequencing reads | https://www.ncbi.nlm.nih.gov/sra/?term=SRP078361 | NCBI Sequence Read Archive, SRP078361 |
| Kouns NA, Nakielna J, Behensky F, Krause MW, Kostrouch Z, Kostrouchova M | 2011 | Expression data in *C. elegans* L2 larvae after nhr-23 inhibition and in controls | https://www.ncbi.nlm.nih.gov/geo/query/acc.cgi?acc=GSE32031 | NCBI Gene Expression Omnibus, GSE32031 |

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

# Appendix 1

## Appendix 1—key resources table

| Reagent type (species) or resource | Designation | Source or reference | Identifiers | Additional information |
|---|---|---|---|---|
| Gene (*Caenorhabditis elegans*) | *nhr-23* | WormBaseWS272 | *nhr-23*; C01H6.5 | |
| Gene (*C. elegans*) | *let-7* | WormBaseWS272 | *let-7*; C05G5.6 | |
| Gene (*C. elegans*) | *lin-42* | WormBaseWS272 | *lin-42*; F47F6.1 | |
| Gene (*C. elegans*) | *mir-84* | WormBaseWS272 | *mir-84*; B0395.4 | |
| Gene (*C. elegans*) | *mir-48* | WormBaseWS272 | *mir-48*; F56A12.3 | |
| Gene (*C. elegans*) | *mir-241* | WormBaseWS272 | *mir-241*; F56A12.4 | |
| Gene (*C. elegans*) | *col-19* | WormBaseWS272 | *col-19*; ZK1193.1 | |
| Gene (*C. elegans*) | *unc-54* | WormBaseWS272 | *unc-54*; F11C3.3 | |
| Gene (*C. elegans*) | *lin-41* | WormBaseWS272 | *lin-41*; C12C8.3 | |
| Gene (*C. elegans*) | *mlt-10* | WormBaseWS272 | *mlt-10*; C09E8.3 | |
| Gene (*C. briggsae*) | *nhr-23* | WormBase ParaSite WBPS9 | *nhr-23*; WBGene00040598 | |
| Gene (*D. melanogaster*) | HR3 | Aug. 2014 (BDGP Release 6+ISO1 MT/dm6) | NM_001259307.3 | Coordinates - chr2: 10,203,995–10,227,957 GenBank Accession EC058128, C0278043, EC080062, AY094723 |
| Gene (*X. tropicalis*) | RORA | Jul. 2016 (Xenopus_tropicalis_ v9.1/xenTro9) | NM_001079195.1 | Coordinates - chr3: 83,889,614–83,948,330 GenBank mRNAs BC123953 |
| Gene (*D. rerio*) | mirlet7a-1 | Sep. 2014 (GRCz10/danRer10) | NR_029976.1 | |
| Gene (*D. rerio*) | mirlet7a-2 | Sep. 2014 (GRCz10/danRer10) | NR_029977.1 | |
| Gene (*D. rerio*) | mirlet7a-3 | Sep. 2014 (GRCz10/danRer10) | NR_029978.1 | |
| Gene (*D. rerio*) | mirlet7a-4 | Sep. 2014 (GRCz10/danRer10) | NR_029979.1 | |
| Gene (*D. rerio*) | mirlet7a-5 | Sep. 2014 (GRCz10/danRer10) | NR_039088.1 | |
| Gene (*D. rerio*) | mirlet7a-6 | Sep. 2014 (GRCz10/danRer10) | NR_029981.1 | |
| Gene (*D. rerio*) | RORAB | Sep. 2014 (GRCz10/danRer10) | NM_201067.1 | Coordinates - chr7: 29,267,488–29,300,423 GenBank Accession BC051158 |
| Gene (*D. rerio*) | RORB | Sep. 2014 (GRCz10/danRer10) | NM_001082856.1 | Coordinates - chr5: 24,761,303–24,776,994 GenBank Accession CK396220, CK679114, EB861010, B092168, EF107093 |
| Gene (*D. rerio*) | RORC | Sep. 2014 (GRCz10/danRer10) | NM_001082819.1 | Coordinates - chr8: 18,851,532–18,862,800 GenBank Accession EB954643, CA472317, CN51179, CN511196, EB893435, EF107094 |
| Gene (*G. gallus*) | RORA | Nov. 2011(ICGSC Gallus_ gallus-4.0/galGal4) | NM_001289887.1 | Coordinates - chr10: 4,187,635–4,536,712 GenBank Accession BU453250, BU298761 |
| Gene (*G. gallus*) | RORB | Nov. 2011(ICGSC Gallus_ gallus-4.0/galGal4) | NM_205093.1 | Coordinates - chrZ: 36,402,033–36,535,271 GenBank Accession Y08638 |
| Gene (*M. musculus*) | Mirlet7a-1 | Dec. 2011(GRCm38/mm10) | NR_029725.1 | |
| Gene (*M. musculus*) | Mirlet7a-2 | Dec. 2011(GRCm38/mm10) | NR_029726.1 | |
| Gene (*M. musculus*) | RORA | Dec. 2011(GRCm38/mm10) | NM_013646 | Coordinates - chr9: 68,653,786–69,388,246 GenBank Accession AK043990, AK035351, AK087905, AK163917 |
| Gene (*M. musculus*) | RORB | Dec. 2011(GRCm38/mm10) | NM_146095 | Coordinates - chr19: 18,930,605–19,111,196 GenBank Accession AK159011, BC058269 |
| Gene (*M. musculus*) | RORC | Dec. 2011(GRCm38/mm10) | NM_011281 | Coordinates - chr3: 94,377,432–94,398,276 GenBank Accesion AJ132394 |

*Appendix 1 Continued on next page*

Appendix 1 Continued

| Reagent type (species) or resource | Designation | Source or reference | Identifiers | Additional information |
|---|---|---|---|---|
| Gene (*H. sapiens*) | *MIRLET7A1* | Dec. 2013 (GRCh38/hg38) | NR_029476.1 | |
| Gene (*H. sapiens*) | *MIRLET7A2* | Dec. 2013 (GRCh38/hg38) | NR_029477.1 | |
| Gene (*H. sapiens*) | *MIRLET7A3* | Dec. 2013 (GRCh38/hg38) | NR_029478.1 | |
| Gene (*H. sapiens*) | *RORA* | Dec. 2013 (GRCh38/hg38) | NM_134261 | Coordinates - chr15: 60,488,284–61,229,302 GenBank Accesion AL832164, AK055969, BC008831 |
| Gene (*H. sapiens*) | *RORB* | Dec. 2013 (GRCh38/hg38) | NM_006914 | Coordinates - chr9: 74,497,335–74,693,177 GenBank Accesion BX647070, AK125162 |
| Gene (*H. sapiens*) | *RORC* | Dec. 2013 (GRCh38/hg38) | NM_005060.3 | Coordinates - chr1: 151,806,071–151,831,802 GenBank Accession - AL834219 |
| Gene (*H. sapiens*) | *PER2* | Dec. 2013 (GRCh38/hg38) | NM_022817.3 | Coordinates - chr2: 238,244,044–238,288,610 GenBank Accession - AB002345 |
| Strain (*E. coli*) | OP50-1 | *Caenorhabditis Genetics Center* (CGC) | *E. coli* OP50-1 | |
| Strain (*E. coli*) | HB101 | *Caenorhabditis Genetics Center* (CGC) | *E. coli* HB101 | |
| Strain (*E. coli*) | DH5α | New England Biolabs | *E. coli* DH5α | |
| Strain (*E. coli*) | vector only' bacteria; control for feeding RNAi | Julie Ahringer; PMID: 12529635 | *E. coli* HT115(DE3)+pPD129.36 | |
| Strain (*E. coli*) | bacteria expressing *nhr-23* dsRNAs | Julie Ahringer; PMID: 12529635 | *E. coli* HT115(DE3)+pPD129.36 with *nhr-23* insert | |
| Strain (*C. elegans*) | wild type; N2 | CGC | N2 | wild type (Bristol) |
| Genetic reagent (*C. elegans*) | *let-7(n2853)* | CGC | MT7626 | *let-7(n2853) X* not out-crossed |
| Genetic reagent (*C. elegans*) | *mir-48(Δ) mir-241(Δ); mir-84(n4037); let-7s* triple knockout | CGC | VT1066 | *nDf51 V; mir-84(n4037) X*. Used in *Figure 6—figure supplement 1B*, *Supplementary file 1*. |
| Genetic reagent (*C. elegans*) | *wgIs43[nhr-23⁺⁺]*; high-copy or increased dosage of *nhr-23* | CGC | OP43 | *unc-119(ed3) III; wgIs43[nhr-23::TY1::EGFP::3xFLAG(92 C12)+unc 119(+)]*. Used in *Figure 6—figure supplement 2A,B,C* |
| Genetic reagent (*C. elegans*) | *mgIs49; mlt-10* transcriptional reporter *strain* | Frand Lab; PMID: 16122351 | GR1395 | *mgIs49[mlt-10p::gfp-pest ttx-3p::gfp] IV*. Used in *Supplementary file 1* |
| Genetic reagent (*C. elegans*) | *let-7(mg279)*; partial loss-of-function *let-7* mutation | Frand Lab; PMID: 17065234 | GR1436 | *mgIs49[mlt-10p::gfp-pest ttx-3p::gfp] IV; let-7(mg279) X*. Used in *Supplementary file 1* |
| Genetic reagent (*C. elegans*) | *let-7(mg279) mir-84(tm1304)* | Frand Lab; PMID: 17065234 | ARF249 | *mgIs49[mlt-10p::gfp-pest ttx-3p::gfp] IV; let-7(mg279) mir-84(tm1304) X*. Used in *Supplementary file 1* |
| Genetic reagent (*C. elegans*) | *let-7p::nls-GFP*; transcriptional reporter for *let-7* | Amy Pasquinelli; PMID: 23201578 | PQ462 | *apIs[let-7bp::NLS::gfp::let-858 3'UTR unc-119(+)] II; unc-119(ed9) III*. Used in *Figure 3—figure supplement 2A,B* |
| Genetic reagent (*C. elegans*) | *let-7(n2853)*; strong loss-of-function *let-7* mutation | Kim Lab | QK059 | *let-7(n2853) X* out-crossed 3 x. Used in *Figure 2B,C*, *Figure 6A, Figure 8B,D, Figure 9C, Figure 6—figure supplement 1A, Figure 8—figure supplement 1A, B, Figure 9—figure supplement 1C, Supplementary file 1* |
| Genetic reagent (*C. elegans*) | *nhr-23::GFP* partial *translational reporter strain* | Marta Kostrouchova; PMID: 9521900 | | |
| Genetic reagent (*C. elegans*) | *nhr-23(aaa20)* CRISPR allele - initial isolate | This report | ARF413 | *nhr-23(aaa20) I* not out-crossed. See "Editing the *C. elegans* Genome" in Materials and Methods. |
| Genetic reagent (*C. elegans*) | *nhr-23(aaa20); nhr-23 3'UTR(ΔLCS)* mutant strain | This report | ARF414 | *nhr-23(aaa20) I* out-crossed 3 x. See "Editing the *C. elegans* Genome" in Materials and Methods. Used in *Figures 6A,B,C–9C, Figure 6—figure supplement 1A, Figure 8—figure supplement 1A,B, Figure 9—figure supplement 1C, Supplementary file 1* |

*Appendix 1 Continued on next page*

*Appendix 1 Continued*

| Reagent type (species) or resource | Designation | Source or reference | Identifiers | Additional information |
|---|---|---|---|---|
| Genetic reagent (*C. elegans*) | *nhr-23::3xflag; nhr-23(xk22)* | This report | QK159 | *nhr-23(xk22) I* out-crossed 5 x. See "Editing the *C. elegans* Genome" in Materials and Methods. Used in *Figure 3C,B, Figure 4B, Figure 9A, Figure 3—figure supplement 1C,D, Figure 4—figure supplement 1.* |
| Genetic reagent (*C. elegans*) | *aaaEx97; unc-54* 3'UTR reporter strain; negative control for bicistronic reporter system | This report | ARF370 | *aaaEx97[dpy-7p::tdtomato::unc-54 3'UTR::SL2::gfp::unc-54 3'UTR +ttx-3::gfp]* See "Construction of Fusion Genes and Transgenic Strains" in Materials and Methods. Used in *Figure 5C,D* |
| Genetic reagent (*C. elegans*) | *aaaEx129; nhr-23* 3'UTR reporter strain | This report | ARF372 | *aaaEx129[dpy-7p::tdtomato::nhr-23 3'UTR::SL2::gfp::unc-54 3'UTR +ttx-3::gfp].* See "Construction of Fusion Genes and Transgenic Strains" in Materials and Methods. Used in *Figure 5C,D,E* |
| Genetic reagent (*C. elegans*) | *aaaEx146; lin-41* 3'UTR reporter strain; positive control for bicistronic reporter system | This report | ARF399 | *aaaEx146[dpy-7p::tdtomato::lin-41 3'UTR::SL2::gfp::unc-54 3'UTR +ttx-3::gfp].* See "Construction of Fusion Genes and Transgenic Strains" in Materials and Methods. Used in *Figure 5C,D* |
| Genetic reagent (*C. elegans*) | *aaaEx131; nhr-23 3'UTR(ΔLCS)* reporter strain | This report | ARF374 | *aaaEx131[dpy-7p::tdtomato::nhr-23 3'UTR(ΔLCS)::SL2::gfp::unc-54 3'UTR +ttx-3::gfp].* See "Construction of Fusion Genes and Transgenic Strains" in Materials and Methods. Used in *Figure 5C,E* |
| Genetic reagent (*C. elegans*) | *aaaEx165; nhr-233'UTR(Δ623–646)* reporter strain | This report | ARF400 | *aaaEx165[dpy-7p::tdtomato::nhr-23 3'UTR(Δ623–646)::SL2::gfp::unc-54 3'UTR +ttx-3::gfp].* See "Construction of Fusion Genes and Transgenic Strains" in Materials and Methods. Used in *Figure 5E* |
| Genetic reagent (*C. elegans*) | *aaaEx166; nhr-233'UTR(Δ227–249)* reporter strain | This report | ARF401 | *aaaEx166[dpy-7p::tdtomato::nhr-23 3'UTR(Δ227–249)::SL2::gfp::unc-54 3'UTR ttx-3::gfp].* See "Construction of Fusion Genes and Transgenic Strains" in Materials and Methods. Used in *Figure 5E* |
| Genetic reagent (*C. elegans*) | *aaaEx130; nhr-23 3'UTR(Δ26–42)* reporter strain | This report | ARF373 | *aaaEx130[dpy-7p::tdtomato::nhr-23 3'UTR(Δ26–42)::SL2::gfp::unc-54 3'UTR ttx-3::gfp].* See "Construction of Fusion Genes and Transgenic Strains" in Materials and Methods. Used in *Figure 5E.* |
| Genetic reagent (*C. elegans*) | *wgIs43; let-7(mg279) mir-84(tm1304)* | This report | ARF422 | *wgIs43[nhr 23::TY1::EGFP::3xFLAG(92 C12)+unc 119(+)]; let-7(mg279) mir-84(tm1304) X.* See "Construction of Fusion Genes and Transgenic Strains" in Materials and Methods. Used in *Figure 6—figure supplement 2C* |
| Genetic reagent (*C. elegans*) | *aaaEx180, mouse let-7p::gfp* transcriptional reporter | This report | ARF431 | *aaaEx180[Mmu-let-7a-1p::gfp ttx-3::gfp].* See "Construction of Fusion Genes and Transgenic Strains" in Materials and Methods. Used in *Figure 10B, C* |
| Genetic reagent (*C. elegans*) | *aaaEx172; Hs. RORB 3'UTR reporter - line 1* | This report | ARF423 (RA101) | *aaaEx172[dpy-7p::tdtomato::Hs. RORB 3'UTR::SL2::gfp::unc-54 3'UTR +ttx-3::gfp].* See "Construction of Fusion Genes and Transgenic Strains" in Materials and Methods. Used in *Figure 10E,F, Figure 10—figure supplement 1B* |
| Genetic reagent (*C. elegans*) | *aaaEx173, Hs. RORB 3'UTR reporter - line 2* | This report | ARF424 (RA102) | *aaaEx173[dpy-7p::tdtomato::Hs. RORB 3'UTR::SL2::gfp::unc-54 3'UTR +ttx-3::gfp].*See "Construction of Fusion Genes and Transgenic Strains" in Materials and Methods. Used in *Figure 10F, Figure 10—figure supplement 1B* |
| Genetic reagent (*C. elegans*) | *aaaEx174; Hs. RORB 3'UTR(ΔLCS)* reporter strain - *line 1* | This report | ARF425 (RA106) | *aaaEx174[dpy-7p::tdtomato::Hs. RORB 3'UTR(ΔLCS)::SL2::gfp::unc-54+3'UTR; ttx-3::gfp].* See "Construction of Fusion Genes and Transgenic Strains" in Materials and Methods. Used in *Figure 10E, F, Figure 10—figure supplement 1B* |
| Genetic reagent (*C. elegans*) | *aaaEx175; Hs. RORB 3'UTR* reporter strain - line 2 | This report | ARF426 (RA107) | *aaaEx175[dpy-7p::tdtomato::Hs. RORB 3'UTR(ΔLCS)::SL2::gfp::unc-54 3'UTR +ttx-3::gfp].* See "Construction of Fusion Genes and Transgenic Strains" in Materials and Methods. Used in *Figure 10F, Figure 10—figure supplement 1C* |
| Genetic reagent (*C. elegans*) | *aaaEx176; Ms. RORA 3'UTR* reporter strain - line 1 | This report | ARF427 (RA104) | *aaaEx176[dpy-7p::tdtomato::Ms. RORA 3'UTR::SL2::gfp::unc-54 3'UTR +ttx-3::gfp].* See "Construction of Fusion Genes and Transgenic Strains" in Materials and Methods. Used in *Figure 10H, Figure 10—figure supplement 1C* |
| Genetic reagent (*C. elegans*) | *aaaEx177; Ms. RORA 3'UTR* reporter strain - line 2 | This report | ARF428 (RA105) | *aaaEx177[dpy-7p::tdtomato::Ms. RORA 3'UTR::SL2::gfp::unc-54 3'UTR +ttx-3::gfp].* See "Construction of Fusion Genes and Transgenic Strains" in Materials and Methods. Used in *Figure 10G,H, Figure 10—figure supplement 1C* |

*Appendix 1 Continued on next page*

*Appendix 1 Continued*

| Reagent type (species) or resource | Designation | Source or reference | Identifiers | Additional information |
|---|---|---|---|---|
| Genetic reagent (*C. elegans*) | *aaaEx178; Ms. RORA 3'UTR(ΔLCS)* reporter strain - line 1 | This report | ARF429 (RA109) | *aaaEx178[dpy-7p::tdtomato::Ms. RORA 3'UTR(ΔLCS)::SL2::gfp::unc-54 3'UTR +ttx-3::gfp]*. See "Construction of Fusion Genes and Transgenic Strains" in Materials and Methods. Used in *Figure 10G, H, Figure 10—figure supplement 1C* |
| Genetic reagent (*C. elegans*) | *aaaEx179; Ms. RORA 3'UTR(ΔLCS)* reporter strain - line 2 | This report | ARF430 (RA110) | *aaaEx179[dpy-7p::tdtomato::Ms. RORA 3'UTR(ΔLCS)::SL2::gfp::unc-54 3'UTR +ttx-3::gfp]*. See "Construction of Fusion Genes and Transgenic Strains" in Materials and Methods. Used in *Figure 10H, Figure 10—figure supplement 1C* |
| Genetic reagent (*C. elegans*) | *aaaEx181; NHR-23::GFP* strain | This report | ARF432 | *aaaEx181[nhr-23p::nhr-23::gfp::unc-54 3'UTR +ttx-3::gfp]*. See "Construction of Fusion Genes and Transgenic Strains" in Materials and Methods. Used in *Figure 9D,E,F* |
| Genetic reagent (*C. elegans*) | *let-7(scRORE1, scRORE3)* CRISPR allele; *let-7(xk39)* | This report | QK198 | *let-7(xk39)X* outcrossed 3 x. See Materials and Methods: Editing the *C. elegans* genome. Used in *Figure 4C,D,F* |
| Genetic reagent (*C. elegans*) | *let-7(scRORE1, scRORE3)* CRISPR allele; *let-7(xk42)* | This report | QK199 | *let-7(xk42)X* outcrossed 3 x. See Materials and Methods: Editing the *C. elegans* genome. Used in *Figure 4F* |
| Genetic reagent (*C. elegans*) | *let-7(scRORE1, scRORE2)* CRISPR allele; *let-7(xk41)* | This report | QK201 | *let-7(xk41) X* out-crossed 3 x. See Materials and Methods: Editing the *C. elegans* genome. Used in *Figure 4C,D,F* |
| Genetic reagent (*C. elegans*) | *let-7(scRORE1, scRORE2)* CRISPR allele; *let-7(xk43)* | This report | QK202 | *let-7(xk43)X* out-crossed 3 x. See Materials and Methods: Editing the *C. elegans* genome. Used in *Figure 4F* |
| Genetic reagent (*C. elegans*) | *let-7(scRORE1, scRORE2)* CRISPR allele; *let-7(xk44)* | This report | QK203 | *let-7(xk44) X* out-crossed 3 x. See Materials and Methods: Editing the *C. elegans* genome. Used in *Figure 4F* |
| Genetic reagent (*C. elegans*) | *wIs54(scm::GFP); let-7(scRORE1, scRORE3)* CRISPR allele; *let-7(xk39)* | This report | QK204 | *wIs54(scm::GFP); let-7(xk39) X*. Used in *Figure 4E* |
| Genetic reagent (*C. elegans*) | *wIs54(scm::GFP); let-7(scRORE1, scRORE3)* CRISPR allele; *let-7(xk42)* | This report | QK205 | *wIs54(scm::GFP); let-7(xk42) X*. Used in *Figure 4E* |
| Genetic reagent (*C. elegans*) | *wIs54(scm::GFP); let-7(mg279)* | This report | QK206 | *wIs54(scm::GFP); let-7(mg279) X*. Used in *Figure 4E* |
| Genetic reagent (*C. elegans*) | *wIs54(scm::GFP); let-7(scRORE1, scRORE2)* CRISPR allele; *let-7(xk41)* | This report | QK208 | *wIs54(scm::GFP); let-7(xk41) X*. Used in *Figure 4E* |
| Genetic reagent (*C. elegans*) | *wIs54(scm::GFP); let-7(scRORE1, scRORE2); let-7(xk43)* | This report | QK209 | *wIs54(scm::GFP); let-7(xk43)X*. Used in *Figure 4E* |
| Genetic reagent (*C. elegans*) | *wIs54(scm::GFP); let-7(scRORE1, scRORE2)* CRISPR allele; *let-7(xk44)* | This report | QK210 | *wIs54(scm::GFP); let-7(xk44) X*. Used in *Figure 4E* |
| Genetic reagent (*C. elegans*) | *wIs54(scm::GFP); let-7(n2853)* | This report | QK036 | *wIs54(scm::GFP); let-7(n2853) X*. Used in *Figure 4E* |
| Genetic reagent (*C. elegans*) | *nhr-23(xk22); let-7(scRORE1, scRORE3)* CRISPR allele; *let-7(xk39)* | This report | QK211 | *nhr-23(xk22)I; let-7(xk39) X*. Used in *Figure 4C, Figure 4—figure supplement 1* |
| Genetic reagent (*C. elegans*) | *nhr-23(xk22); let-7(scRORE1, scRORE2)* CRISPR allele; *let-7(xk41)* | This report | QK212 | *nhr-23(xk22) I; let-7(xk41) X*. Used in *Figure 4C, Figure 4—figure supplement 1* |
| Recombinant DNA reagent | *unc-54* 3'UTR reporter; negative control for bicistronic fusion genes | This report | pHR011 | *dpy-7p::tdtomato::unc-54 3'UTR::SL2::gfp::unc-54 3'UTR*. See "Construction of Fusion Genes and Transgenic Strains" in Methods. |
| Recombinant DNA reagent | *nhr-23* 3'UTR bicistronic reporter construct | This report | pHR017 | *dpy-7p::tdtomato::nhr-23 3'UTR::SL2::gfp::unc-54 3'UTR*. See "Construction of Fusion Genes and Transgenic Strains" in Methods. |

*Appendix 1 Continued on next page*

Appendix 1 Continued

| Reagent type (species) or resource | Designation | Source or reference | Identifiers | Additional information |
|---|---|---|---|---|
| Recombinant DNA reagent | *lin-41* 3'UTR reporter; positive control for bicistronic fusion genes | This report | pHR023 | *dpy-7p::tdtomato::lin-41 3'UTR::SL2::gfp::unc-54 3'UTR*. See "Construction of Fusion Genes and Transgenic Strains" in Methods. |
| Recombinant DNA reagent | *nhr-23* 3'UTR(ΔLCS) bicistronic reporter construct | This report | pHR021 | *dpy-7p::tdtomato::nhr-23 3'UTRΔLCS::SL2::gfp::unc-54 3'UTR*. See section titled "Construction of Fusion Genes and Transgenic Strains" in Methods. |
| Recombinant DNA reagent | *nhr-23* 3'UTR(Δ623–646) bicistronic reporter construct | This report | pHR022 | *dpy-7p::tdtomato::nhr-23 3'UTRΔ623–646::SL2::gfp::unc-54 3'UTR*. See section titled "Construction of Fusion Genes and Transgenic Strains" in Methods. |
| Recombinant DNA reagent | *nhr-23* 3'UTR(Δ227–249) bicistronic reporter construct | This report | pHR026 | *dpy-7p::tdtomato::nhr-23 3'UTRΔ227–249::SL2::gfp::unc-54 3'UTR*. See section titled "Construction of Fusion Genes and Transgenic Strains" in Methods. |
| Recombinant DNA reagent | *nhr-23* 3'UTR(Δ26–42) bicistronic reporter construct | This report | pHR020 | *dpy-7p::tdtomato::nhr-23 3'UTRΔ26–42::SL2::gfp::unc-54 3'UTR*. See section titled "Construction of Fusion Genes and Transgenic Strains" in Methods. |
| Recombinant DNA reagent | *mouse let-7 promoter fused to GFP* | This report | pRA46 | *aaaEx180[Mmu-let-7a-1p::gfp ttx-3::gfp]*. See section titled "Construction of Fusion Genes and Transgenic Strains" in Methods. |
| Recombinant DNA reagent | *Mm.RORA* 3'UTR bicistronic reporter construct | This report | pRA41 | *dpy-7p::tdtomato::Mm RORA 3'UTR::SL2::gfp::unc-54 3'UTR*. See section titled "Construction of Fusion Genes and Transgenic Strains" in Methods. |
| Recombinant DNA reagent | Mm. RORA 3'UTR(ΔLCS) bicistronic reporter construct | This report | pRA43 | *dpy-7p::tdtomato::Mm RORA 3'UTRΔLCS::SL2::gfp::unc-54 3'UTR*. See section titled "Construction of Fusion Genes and Transgenic Strains" in Methods. |
| Recombinant DNA reagent | *Hs. RORB* 3'UTR bicistronic reporter construct | This report | pRA38 | *dpy-7p::tdtomato::Hs RORB 3'UTR::SL2::gfp::unc-54 3'UTR*. See section titled "Construction of Fusion Genes and Transgenic Strains" in Methods. |
| Recombinant DNA reagent | *Hs. RORB* 3'UTR(ΔLCS) bicistronic reporter construct | This report | pRA40 | *dpy-7p::tdtomato::Hs RORB 3'UTRΔLCS::SL2::gfp::unc-54 3'UTR*. See section titled "Construction of Fusion Genes and Transgenic Strains" in Methods. |
| Recombinant DNA reagent | *nhr-23::GFP partial translational reporter* | **Kostrouchova et al., 1998** | plasmid 4271 | *nhr-23p::nhr-23::gfp::unc-543'UTR* |
| Antibody | Monoclonal Anti-Flag M2 antibody | Sigma-Aldrich | F3165 | |
| Peptide, recombinant protein | CAS9 protein | PNA Bio | CP01 | |
| Peptide, recombinant protein | 1 x HALT Protease and Phosphatase Inhibitor | ThermoFisher Scientific | 78443 | |
| Peptide, recombinant protein | Proteinase K | ThermoFisher Scientific | 25530015 | |
| Peptide, recombinant protein | RNase A | ThermoFisher Scientific | 12091021 | |
| Peptide, recombinant protein | RNAsin Plus RNase inhibitor, 1 U/µL | Promega | N2611 | |
| Peptide, recombinant protein | Transcriptor Reverse Transcriptase, 0.5 U/µL | Roche | 3531295001 | |
| Peptide, recombinant protein | KpnI-HF | New England Biolabs | R3142S | |
| Peptide, recombinant protein | NotI-HF | New England Biolabs | R3189S | |
| Commercial assay or kit | Gibson DNA Assembly Kit | New England Biolabs | E5510S | |

*Appendix 1 Continued on next page*

*Appendix 1 Continued*

| Reagent type (species) or resource | Designation | Source or reference | Identifiers | Additional information |
|---|---|---|---|---|
| Commercial assay or kit | TOPO TA Cloning Kit | ThermoFisher Scientific | 450641 | |
| Commercial assay or kit | Phusion High-Fidelity DNA Polymerase | New England Biolabs | M0530S | |
| Commercial assay or kit | Q5 Site-Directed Mutagenesis Kit | New England Biolabs | E0554S | |
| Commercial assay or kit | TURBO DNase Kit | ThermoFisher Scientific | AM1907 | |
| Commercial assay or kit | High-Capacity cDNA Reverse Transcription Kit | ThermoFisher Scientific | 4368814 | |
| Commercial assay or kit | SuperScript III Reverse Transcriptase | Invitrogen | 18080044 | |
| Commercial assay or kit | TaqMan Universal PCR Master Mix, no AmpErase UNG | ThermoFisher Scientific | 4364341 | |
| Commercial assay or kit | TaqMan assay for *lin-42* | ThermoFisher Scientific | CE02593603_M1 | |
| Commercial assay or kit | TaqMan assay for *nhr-23* | ThermoFisher Scientific | CE02405513_G1 | |
| Commercial assay or kit | TaqMan assay for *mlt-10* | ThermoFisher Scientific | CE02426995_M1 | |
| Commercial assay or kit | TaqMan assay for *fbn-1* | ThermoFisher Scientific | CE02449109_G1 | |
| Commercial assay or kit | TaqMan assay for *ama-1* | ThermoFisher Scientific | CE02462732_G1 | |
| Commercial assay or kit | TaqMan assay for primary *let-7* | ThermoFisher Scientific | AJ1RUH0 | |
| Commercial assay or kit | TaqMan assay for primary *mir-84* | ThermoFisher Scientific | AJ5IO2R | |
| Commercial assay or kit | TaqMan assay for primary *mir-48* | ThermoFisher Scientific | AJ205NW | |
| Commercial assay or kit | TaqMan assay for primary *mir-241* | ThermoFisher Scientific | AJ39QT4 | |
| Commercial assay or kit | TaqMan assay for mature *let-7* | ThermoFisher Scientific | 377 | |
| Commercial assay or kit | TaqMan assay for mature *mir-84* | ThermoFisher Scientific | 236 | |
| Commercial assay or kit | TaqMan assay for mature *mir-48* | ThermoFisher Scientific | 208 | |
| Commercial assay or kit | TaqMan assay for mature *mir-241* | ThermoFisher Scientific | 249 | |
| Commercial assay or kit | TaqMan assay for U18 | ThermoFisher Scientific | 1764 | |
| Chemical compound | Isopropyl β-D-1-thiogalactopyranoside (IPTG) | Laguna Scientific | 6055–5 | |
| Chemical compound | TRIzol | ThermoFisher Scientific | 15596026 | |
| Chemical compound | Chloroform: Isoamyl alcohol (24:1) | OmniPur | 3160–450 ML | |
| Chemical compound | Acid Phenol: Chloroform (5:1) | ThermoFisher Scientific | AM9720 | |
| Chemical compound | Isopropanol, 100% | Sigma-Aldrich | I9516 | |
| Chemical compound | GlycoBlue | Sigma-Aldrich | AM9515 | |
| Chemical compound | Random primers, 25 ng/µL | Promega | C1181 | |

*Appendix 1 Continued on next page*

*Appendix 1 Continued*

| Reagent type (species) or resource | Designation | Source or reference | Identifiers | Additional information |
|---|---|---|---|---|
| Chemical compound | dNTPs, 1 mM | New England Biolabs | N0446S | |
| Chemical compound | Novex Tris- Glycine SDS Sample Buffer | ThermoFisher Scientific | LC2676 | |
| Chemical compound | Absolute Blue SYBR Green | ThermoFisher Scientific | AB4322B | |
| Chemical compound | Dynabeads M-280 Streptavidin | ThermoFisher Scientific | 11205D | |
| Software or algorithm | Photoshop 21.0.1 | Adobe | NA | |
| Software or algorithm | Illustrator 24.0.1 | Adobe | NA | |
| Software or algorithm | NCBI BLAST | National Insitutes of Health | NA | |
| Software or algorithm | ImageJ v2.0.0-rc-43/1.50e | National Insitutes of Health | NA | |
| Software or algorithm | SnapGene | GSL Biotech LLC | NA | |
| Software or algorithm | RNAhybrid | PMID:15383676 | NA | |
| Software or algorithm | GraphPad Prism v6.0h | GraphPad Software, Inc | NA | |
| Software or algorithm | Volocity 6.3 | Perkin Elmer | NA | |
| Software or algorithm | R version 3.5.2 | R Studio | NA | |
| Software or algorithm | RStudio version 1.1.463 | R Studio | NA | |

