## [Editor Report]

The manuscript nicely advances our understanding of the roles of heterochronic genes and the NHR-23 nuclear receptor transcription factor in the regulation of the temporal dynamics of molting behavior in *C. elegans* larval development. The data reveals direct regulatory feedback between let-7 family microRNAs and nhr-23, and shows that this circuit contributes to the regulation of developmental pace. The findings should be of interest to the field studying heterochronic genes and microRNAs in developmental timing, and to the broader field of chronobiology, particularly the regulation of complex oscillatory gene regulatory networks.

---

## [Decision Letter]

**Decision letter after peer review:**

[Editors’ note: the authors submitted for reconsideration following the decision after peer review. What follows is the decision letter after the first round of review.]

Thank you for submitting your work entitled "Feedback among a retinoid-related nuclear receptor and *let-7* miRNAs controls the pace and number of molts in *C. elegans*" for consideration by *eLife*. Your article has been reviewed by 2 peer reviewers, one of whom is a member of our Board of Reviewing Editors, and the evaluation has been overseen by a Reviewing Editor and a Senior Editor. The reviewers have opted to remain anonymous.

Our decision has been reached after consultation between the reviewers. Based on these discussions and the individual reviews below, we regret to inform you that your work will not be considered further for publication in *eLife*.

There has been a lot of enthusiasm about the overall problem addressed in your paper, but as you will see detailed in the reviews below, there has been some substantial concern about whether the intriguing claims are really completely supported by the data. Without such explicit support, the advance relative to previous studies was deemed insufficient (even though it was much appreciated that your results confirm and extend previous insights). As you will also see, the reviewers have made suggestions for how to improve the manuscript. If you decided to undertake the substantial revisions suggested, we would be interested in considering such a manuscript, but this would then count as a new submission.

*Reviewer #1:*

In the manuscript "Feedback among a retinoid-related nuclear receptor and let-7 miRNAs controls the pace and number of molts in *C. elegans*", Patel et al. aim to demonstrate a feedback network between retinoid-related nuclear receptor nhr-23 and let-7 miRNA family in the control of the pace/number of molts in C.elegans. First, they perform a quantitative and detailed characterization of the molting pattern throughout development in a number of different genetic manipulations of nhr-23 and/or let-7 family miRNAs and demonstrate that they affect the length/number of the molting cycle. Then, they go on to demonstrate direct regulation between nhr-23 and let-7 family miRNAs. The authors offer too simplistic of a feedback model for something much more complex and what the data suggest (elaborated in more detail below). Much of what the authors convincingly demonstrate in this manuscript, although done in a more detailed/quantitative manner and sometimes using novel reagents, has been previously shown/implicated (Kostrouchova et al., 2001; Frand et al., 2005; Hayes et al., 2006; Celniker et al., 2009). The portion of the manuscript that could be potentially interesting and novel (the mechanisms by which nhr-23/let-7 alter molting cycle length/number, conservation of this regulation in other systems, etc), the authors offer very little insight and experimental support. In its current form, without further experiments/analysis as suggested below, this manuscript does not significantly advance beyond the current understanding on the topic.

1) While the authors convincingly demonstrate that individual genetic manipulations of let-7 and nhr-23 oppositely regulate the pace of the molting cycle, it is not clear how feedback/molecular regulation between let-7 and nhr-23 regulate the pace. Further exploration of the mechanism will greatly enhance the novelty and significance of the manuscript.

a. Genetically, the authors demonstrate that reducing nhr-23 levels delay the pace of larval development/molting cycle while reducing let-7 levels accelerate the pace of larval development/molting cycle. At the molecular level, nhr-23 transcript level is oscillatory, reaching its peak level in the middle of the interval between molts and reaching its lowest level right before a molt (Kostrouchova et al., 2001; current manuscript). In nhr-23 (aaa20-deltaLCS) or let-7 mutant animals, the authors report elevated peak nhr-23 transcripts level, steeper slopes of rise and wider troughs compared to control. How do (which of) these characteristics contribute to a faster pace? Are there uncharacterized features (ie when in the cycle the peak is reached, time in between peaks, etc) that could help explain its effect on the pace? It seems that the slope of the fall is also steeper in the mutant animals compared to control. If so after quantification, how can the faster degradation of nhr-23 be explained in let-7 mutant animals?

b. More optimally, the questions raised above could be addressed by examining downstream molting-related genes. The authors examine 1 such genes (mlt-10) only in mutant animals with elevated nhr-23 levels. The authors should examine additional genes important in each phase of the molting cycle as outlined by the authors in Figure 9D in genetic manipulation of both increased/decreased nhr-23 levels. Similar examination should be done with lin-42. Together, these could provide a satisfactory answer as to the mechanism by which the pace of the developmental/molting cycle are regulated by genetic manipulation of nhr-23 and let-7. Additionally, some of these molting genes should be examined in supernumerary molts in the adult. These could further strengthen any conclusion made about these genes in the regulation of molt timing.

2) Based upon previous reports and the data presented in this manuscript, they suggest a much more complex regulation than the simple model proposed. Further experimentation and discussion are necessary in the manuscript to address these additional relevant/related regulations not incorporated in the current model.

a. Based upon previous reports (Hayes et al., 2006, Hada et al., 2010) that both nhr-23 and nhr-25 are downstream of let-7 family miRNA regulation of molts, it is important to also examine the role of nhr-25 in the study and how it fits into the model.

b. Based on a previous report (Hayes et al., 2006), nhr-23 GFP reporter without 3'UTR also showed de-repression in let-7 mutant. This suggests that regulation of nhr-23 by let-7 is not simply direct at the level of the 3'UTR. The author should consider experiments examining the effect of known downstream heterochronic regulators of let-7 (ie lin-41, lin-29 etc) on nhr-23 or at least address this point in the discussion of the proposed model.

c. In Fig3F, nhr-23 RNAi does not significantly (statistics?) dampen mature let-7 expression until L4/adult stage. Is there stage specificity in terms of nhr-23 regulation of let-7? Additionally, lin-42 (statistics?) also seems to be only significantly regulated by nhr-23 in L4 (Supp Figure 5B). Does this hold any significance? Further discussion of this should be included.

3) In some cases, there are suggestions of some conclusion but these were not explicitly demonstrated. The authors should provide more experimental evidence in these cases.

a. It is suggested that elevated/continuous cyclic expression of nhr-23 beyond the L4/YA molt is responsible for the supernumerary molts, however this correlation was not explicitly demonstrated. At the very least, the authors should take the nhr-23::GFP overexpression strain (wgIs43) and demonstrate that there is cycling of GFP signal that precedes each supernumerary molt.

b Conservation of the cross regulation between nhr-23 and let-7 is suggested by the author through bioinformatics analysis. This is optimally tested in the respective system if technical capabilities allow, but at the very least, the authors should demonstrate that these promoters/3' UTR sequences from other systems can be regulated in the predictive manner in *C. elegans* when transgenically expressed. Such demonstration would add to the novelty of the manuscript.

c. They authors demonstrate nhr-23 binding at the promoter of other let-7 family miRNA (mir-48, 84, 241). Are their levels regulated by nhr-23?

d. It is suggested that nhr-23 auto-regulates itself due to the presence of ROR binding site in its promoter. This should be tested by the in vivo deletion of this site via CRISPR/Cas-9 and examine the resulting nhr-23 expression.

(4) Statistics are not provided in many places in the manuscript/figures. Sometimes even error bars are not apparently in figures. Is there only a single replicate in these cases? The below is not an exhaustive list, the author should go through to make sure every numerical figures have been properly analyzed and provide detail statistical information for other figures as they have done in Supplementary Table 1: Figure 3E/F, Figure 5A, Supp Figure 3, Figure 7B, Supp Figure 5b.

*Reviewer #2:*

In this manuscript, Patel et al., describe a regulatory loop between a Retinoid-Related Orphan Receptor (ROR) in *C. elegans*, NHR-23, and the let-7 microRNA family. The authors examine lethargus and molting behavior in worms with reduced activity or increased activity of NHR-23 along with worms that have reduced activity of the let-7 family (let-7, mir-48, mir-84, and mir-241). They provide compelling evidence that NHR-23 binds to the let-7 (and other family members to a lesser degree) promoter to promote let-7 transcription and activity. Additionally, they provide evidence that let-7 then functions to act through the nhr-23 3' UTR to repress nhr-23 activity and protein accumulation. In the absence of negative regulation by let-7, nhr-23 transcript and protein levels are misregulated showing elevated levels and a temporal shift in its oscillations. In the absence of let-7 negative regulation or in the presence of overexpression nhr-23, worms show aberrant molting behavior. This is also associated with a shift in mlt-10 expression, which is known regulator of molting, and is a predicted shared target of let-7 and NHR-23. Sequence analysis suggests that this feedback mechanism may be conserved in higher organisms.

Overall, the expression data supports a model whereby the feedback loops between nhr-23, let-7, and lin-42/PER can drive the cyclic lethargus and molting behavior. The data and experimental approach are strong and rigorous. However, the authors interpretation and model go beyond what the data can directly support. There are issues throughout the manuscript with the clarity of the writing with instances of vague and confusing language and inconsistent use of terms. For example, at different points in the manuscript, the authors refer to lethargus, sleep phase, and sleep-like phase. The manuscript should be extensively revised to improve the accuracy and clarity of the writing.

1. The authors perform qRT-PCR for nhr-23 and mlt-10 transcripts at specific time points in larval development to track the oscillations in expression. While the oscillations are clear as are the changes in the dynamics/shape of the oscillations in mutant backgrounds, the calculation of amplitude and "rising slope" is not well supported. While such numbers seem helpful in generally describing the overall dynamics, it seems misleading to provide such seemingly precise quantification of the comparisons of expression at different time points. It appears that the data shown is from one biological replicate performed with technical duplicate data points (at least this is my interpretation of what was done as described in the figure legend). qRT-PCR is typically a very sensitive assay with a degree of variability between biological replicates. Thus, the calculated amplitude and rising slope may vary significantly between experiments. This is observed in the authors' data (Figure 5 and Supplemental Figure 3). If there are references to support such calculation of amplitude and rising slope, then these should be provided. It is my opinion that more general, qualitative descriptions of the oscillations in transcript levels would be more appropriate.

2. The description of how CCGs were identified and selected should be described more clearly (line 526). In this section, the sentences in lines 546-548 and line 550-551 are also very unclear.

3. In the Discussion, the authors describe a detailed model of sequential activation of genes involved in cuticle biogenesis and lethargus as regulated by NHR-23 and let-7. While this is a compelling model, the details, including that NHR-23 "schedules" the start of cuticle biogenesis and that the sequential activation is driven by the number of ROREs and the amount of NHR-23 remain quite speculative. The authors should more clearly write this section as a theoretical model. The description of NHR-23 and let-7 controlling the rate of transcript accumulation and decay along with the "evidently short half-life of NHR-23 protein is not sufficiently supported by their data. The authors should also explain what is meant by the biogenesis of the 'sheath'.

4. The observations that ROREs were found upstream of let-7 genes in other organisms and that LCSs were found in other nhr-23/ROR homologs is compelling but doesn't seem sufficient to propose that this regulation is conserved in human tissues (lines 634-636).

[Editors’ note: further revisions were suggested prior to acceptance, as described below.]

Thank you for resubmitting your work entitled "Feedback between a retinoid-related nuclear receptor and the *let-7* microRNAs controls the pace and number of molting cycles in *C. elegans.*" for further consideration by *eLife*. Your revised article has been reviewed by three peer reviewers, and the evaluation has been oveeseen by Marianne Bronner (Senior Editor) and a Reviewing Editor.

The manuscript has been improved but there are some remaining issues that need to be addressed, as outlined below:

Everyone agrees that this paper is of interest for *eLife*, but there is also a broad consensus that additional revisions are required that relate to tampering of claims and various other editorial revisions. No further experimentation is required. You will see all the requested detailed below by the reviewers.

*Reviewer #3 (Recommendations for the authors):*

Patel et al. have significantly improved the manuscript over the last 2 years. They have convincingly demonstrated the cross regulation between nhr-23 and let-7 using a combination of biochemical and functional approaches. They have further strengthened their claims of a possible conservation of the let-7/nhr-23 cross regulation in other species (mouse and human) with additional experiments. However, the paper as presented currently, still contains the following major weaknesses:

(1) Many claims made in the manuscript were broad and generalized when they were only supported in specific instances or often contrary to data presented. A non-exhaustive list is included below. The authors should go through the manuscript to make sure all of their claims are specific/precise and are supported by the data presented, and adjust conclusions accordingly:

a. Line 215-216: "the cohort of nhr-23(RNAi) animals that emerged as L4 entered lethargus later". This is unsupported in Figure 2B as well as Supplementary Table 1. The mean times to enter lethargy were 8.1 and 8.4 hours for control and nhr-23(RNAi) animals respectively, and are not statistically different from one another.

b. Line 225-226: "delayed and protracted lethargi were associated with nhr-23 across three larval stages." According to Figure 2 and Supplementary table 3, while the time spent in lethargy was prolonged across all three stages, the delayed lethargi (time in active) was only statistically different in L3.

c. Line 319-321: "the level of other members of the let-7 family, mir-48, mir-84, mir-241 were similarly reduced in nhr-23 (RNAi) larvae developing across the L3 stage." The levels of mature mir-84 and mir-241 were not significantly altered according to Figure 3 supplement 2D

d. Line 449-450: The authors claimed that nhr-23 levels in L2 peaked at higher levels in mir-48, mir-241, mir-84 triple mutant compared to wild type controls. This statement was not supported by data in Figure 6 supplement 1B/B': mir-48, mir-241, mir-84 triple mutant only exhibited higher nhr-23 level in L3 but not L2.

e. Line 556: The authors claimed similar slope of fbn-1 waveform between nhr-23 RNAi and mock-treated larvae, but this was not supported by data in Supplementary Table 3.

(2) I applaud the authors for including replication of their experiment. However, as related to point 1, the authors often claimed that the results of the second replicates supported those of the first replicates when they did not (Some specific examples included below). The authors should address this in their manuscript why these replicates yielded inconsistent results and how to interpret their conclusions in light of this.

a. As opposed to what the authors claimed in line 564-566, independent replicates of fbn-1 and mlt-10 transcripts in mock vs nhr-23 (RNAi) animals were not reproducible across all developmental stages (see Figure 8A vs Figure 8 Supplement 1A, Figure 8C vs Figure 8 F).

b. Similarly, as opposed to what the authors claimed in line 648-649, independent replicates of lin-42 transcript levels in nhr-23 (LCS deletion) and let-7(n2853) animals did not yield the same results (see Figure 9B vs Figure 9 Supplement 1C). It seems that for the second replicate, the stages were not aligned as they were in the first replicate. If they were aligned in phase as they were for the first replicate, there were be no significant difference between the three experimental groups.

(3) The mechanism by which let-7 and nhr-23 control the pacing of the molting cycle is still under-addressed in the paper. The authors have addressed this partially by some additional data (regulation of lin-42 and fbn-1) as well as discussion of additional complexities beyond let-7/nhr-23. While some aspects of the mechanisms might be outside the scope of the current manuscript and not possible to address experimentally, one key aspect that is not currently addressed in the paper is how different manipulation of let-7 and nhr-23 affects different phases of the molting cycle. As one example, nhr-23 RNAi increases time spent in lethargy but does not affect other parts of the molting cycle, while increasing nhr-23 levels (overexpression or deletion of let-7 LCS sites) decreases time spent in active cycle but does not affect time spent in lethargy. Inclusion of the discussion of these points would be important to help the readers think about the complexity of the regulation, and important to consider in future studies to understand the mechanisms by which these regulators control molting cycle pacing.

On a related note, and as alluded by Reviewer 2 previously, the authors use inconsistent terms throughout the manuscript to refer to different aspects of the molting cycle in the manuscript. Standardization of the terms throughout the manuscript would help to increase clarity.

(4) In supplementary Table 1, only statistics for comparisons to control groups were presented. It would be helpful for the readers to present statistics for comparisons between experimental groups as well.

*Reviewer #4 (Recommendations for the authors):*

Patel et al. present data that provide new and substantive understanding of the roles of heterochronic genes and the NHR-23 nuclear receptor transcription factor in the regulation of the temporal dynamics of molting behavior in *C. elegans* larval development. The findings provide a solid basis for the proposal that direct regulatory feedback between let-7 family microRNAs and nhr-23 contributes to the regulation of developmental pace. Novel findings include that NHR-23 promotes the oscillatory transcription of let-7s genes at each larval stage – likely by direct binding to their promotors – and that let-7s microRNAs inhibit NHR-23 expression via complementary LCE sequences in the nhr-23 3' UTR. Key supportive evidence includes the finding nhr-23(gf) mutations accelerate the pace of the L4, as do let-7s(lf) mutations, supporting a dosage-dependent role for NHR-23 in promoting larval stage progression, downstream of let-7s microRNAs. Intriguing evidence is presented for possible conservation in vertebrates of the mutual regulation of ROR/NHR-23 and let-7 family microRNAs.

Overall, the data robustly support novel findings that merit publication in *eLife*. However, some of the conclusions and data interpretations, as stated in the current manuscript, are not supported, and therefore revisions are recommended, according to the comments below.

(1) Lines 164-166: "The cyclical expression profile of primary let-7 family transcripts is consistent with temporally reiterated, as well as stage-specific, function(s)." Do the authors mean to propose here that the let-7 family primary transcripts are functional, aside from being processed into mature let-7 microRNAs? (See points #7 – #10 below).

(2) Line 215: "The cohort of nhr-23(RNAi) animals that emerged as L4s entered lethargus later and remained lethargic for twice as long as the control cohort." It appears that nhr-23 RNAi animals entered L4 lethargus only very slightly later than wt. If the authors wish to argue that the length of the active phase of the larval stage is lengthened in the nhr-23(RNAi) animals, statistical analysis of the length measurements should be shown, to demonstrate significant differences.

(3) Line 248: "When we combined stage-specific nhr-23(RNAi) with let-7(n2853), the altered pace of molting associated with each single mutant was partially co-suppressed (Figure 2B, C and Supplemental Table 1)." I think that a simpler interpretation of the data would be co-expressed phenotypes (rather than co-suppressed). The active phase of the L4 stage (panel B) and L3 stage (panel C) of let7(lf) and let-7(lf);nhr-23(RNAi) are shorter than the wild type; this would correspond to the let-7(lf) phenotype expressed in both cases. At the same time, the lethargus is lengthened in let-7(lf);nhr-23(RNAi); which would appear to be the nhr-23(RNAi) phenotype. Co-suppression would be an apt interpretation if, in the let-7(lf);nhr-23(RNAi) animals the active phases were (statistically) significantly longer than for let-7(lf) alone, and the length of lethargus were (statistically) significantly shorter than for nhr-23(RNAi) alone (see also point #2 above).

(4) Line 265: "Taken together, these longitudinal data suggest a model whereby NHR-23 accelerates the molting cycle, partly by directly activating the expression of the let-7s, and the let-7s decelerate the cycle, partly by directly repressing the expression of nhr-23." At this point in the narrative, it seems premature to propose a specific molecular model, because the data presented so far do not address mechanism. More appropriate at this stage would be a straightforward interpretation of what the phenotypes suggest about how nhr-23 and let-7 microRNAs impact the molting cycle (independent of underlying mechanism): nhr-23 activity negatively regulates the length of lethargus (and perhaps the active phase as well, pending statistical significance, see , points #2 and #3 above), and let-7 family microRNAs negatively regulate the length of the wake-to-wake interval. From the phenotypes alone, there is no reason to necessarily suggest that nhr-23 and let-7 microRNAs regulate each other. Figure 3 addresses mechanism, so this molecular model should be presented after discussion of the Figure 3 results.

(5) Line 270: "Based on the findings of the longitudinal studies described above, we hypothesized that NHR-23 may directly activate transcription of let-7." The problem with this hypothesis is that it does not seem to logically follow from the longitudinal studies presented in Figure 2; nhr-23 and let-7 microRNAs seem to impact molting cycle in opposite ways (accelerating vs decelerating, respectively), so the simplest hypothesis for a regulation of let-7s by nhr-23 would be repression of let-7s by nhr-23. It would be better to propose (broadly) that let-7s and nhr-23 might mutually regulate each other, and then ask, how? The authors could suggest that the simplest expectation could be mutual negative regulation, and then when the data indicate the opposite -- that nhr-23 positively regulates let-7s – deal with the unexpected by modifying the model.

(6) Line 369: "Therefore, scrambling the ROREs is sufficient to increase the speed of development, consistent with our model that NHR-23-mediated activation of the let-7s normally slows the pace of molting." Here is where the issue discussed in point #5 above, comes to a head. In this scenario, where NHR-23 is required for full expression of let-7, one would have expected the nhr-23(lf) phenotype to correspond to that of reduced let-7s – i.e., accelerated development. So, it is apparent that the wild type function of nhr-23 in accelerating the molting cycle cannot be simply via promoting expression of let-7s microRNAs (which themselves seem to decelerate the molting cycle). The authors present data regarding potential regulation of lin-42 by nhr-23, and also potential autoregulation by nhr-23, which should be brought into play in formulating a final model (see point # 10 below).

Note: Points #7, #8, and #9 below relate to problematic issues surrounding the proposal that the oscillation of let-7s microRNAs is a mechanistic component of a hypothetical oscillator driving the molting cycle. Point #10 offers comments regarding a modified model to accommodate these and other issues.

(7) Line 611: "These findings suggest that partly interdependent waves in the abundance of NHR-23 and the let-7s sculpt the temporal expression profiles of fbn-1, mlt-10 and possibly many additional effectors of the molting timer." Does the model require that the mature microRNAs oscillate? At the core of the authors' model is the idea that the steepness of accumulation of NHR-23 in each stage sets the timing of the onset and completion of lethargus, and hence the pace of the molting cycle. let-7s, through their LCSs in nhr-23 mRNA, can inhibit the rate of nhr-23 accumulation in each cycle, and thereby regulate pace. let-7s mature microRNAs do not need to oscillate to perform this proposed function in shaping the NHR-23 pulse

(8) Line 732: "NHR-23 transcriptionally activates let-7 and let-7 post-transcriptionally represses nhr-23 to form a molecular-genetic oscillator." Has it been shown that let-7 mature microRNA oscillates? Figure 3G suggests not. In the authors' model, is it required that mature let-7s microRNAs oscillate?

(9) Line 784: "Thus, NHR-23 and let-7 form a transcriptional-post-transcriptional negative feedback loop. Within a given larval stage, NHR-23 promotes the expression of the let-7s and the let-7s repress nhr-23. The expression levels of let-7 peak ~1.6 hours after nhr-23, resulting in an intrinsic delay between the accumulation of nhr-23 and that of let-7 (Figure 3D, 3F, 6A). Across development, the let-7s dampen the relative amplitude of nhr-23 expression from one larval stage to the next (Figure 6A, Figure 6 —figure supplement 1A). Furthermore, NHR-23 autoregulates its own expression (Figure 9A, 9D-F). Together, these interconnected feedback loops set up a self-sustained molecular-genetic oscillator that is extinguished in adulthood." The problem is that mature let-7s do not seem to oscillate (e.g., Figure 3G). So, if let-7 and nhr-23 are proposed to be oscillating components of an oscillator, the functional let-7 transcript would need to be the pri-let-7s (not mature let-7s!) How would that work? Could perhaps pri-let-7 bind to nhr-23 protein and somehow influence its transcriptional regulatory activity? This would be an exceedingly interesting scenario, if true, but highly theoretical and arguably beyond the scope of the present paper. Moreover, genetic evidence suggests that the molting cycle proceeds independently of let-7-family function, as animals quadruple-mutant for mir-84, mir-48, mir-241, and let-7 still undergo multiple larval stages (Vadla et al., 2012). [BTW, it is not possible to compare Figure 3D and 3F to 6A and make an assessment of the relative timing of pri-let-7 and nhr-23 accumulation.]

(10) A graphic depiction of a revised model consistent with the data would help enormously. (The diagram in Figure 1C is not adequate, as the particular phenotypic consequences are not indicated.) Elements of the model could include the following:

(10A) Nhr-23 is oscillatory, and levels of NHR-23 matter, there being successive thresholds for initiation and completion of lethargus, respectively.

(10B) NHR-23 controls oscillatory transcription of many genes, including let-7s, and lin-42, and itself (by positive autoregulation), and could be a component of an oscillator driving molting cycles. [Current evidence does not support a role for let-7s microRNAs as clock drivers.]

(10C) let-7s microRNAs accumulate progressively across the L2-L4 stages and dampen nhr-23 expression in two ways:

(i) At each stage, repression by let-7s slows the rise of NHR-23, and hence the let-7s microRNAs exert a slowing effect on larval stage pace by affecting how rapidly NHR-23 levels cross the thresholds referred to in (A) above.

(ii) After the L4 stage, repression by let-7s puts NHR-23 levels below the threshold for initiation of molting, satisfying one of the conditions for cessation of molting (the other being expression of LIN-29.)

Other notes regarding the model:

(10D) The authors propose that the termination of molting after the L4 is a consequence of the progressive dampening of NHR-23 expression across the final larval stages. This is a plausible model, but it is incomplete, as it does not account for previous findings that lin-29 is required for cessation of molting. The authors' finding that nhr-23 LCE mutations result in apparent molting behavior of adults (Figure 7A) suggests that nhr-23 activity could be sufficient for supernumerary molting. That novel finding motivates a model wherein cessation of molting requires two conditions to be filled: upregulation of lin-29 (an indirect consequence of let-7 activity in the L4), and dampening of nhr-23 expression (also via let-7).

(10E) As the authors data suggest (e.g., Figure 8) let-7s microRNAs could be shaping the dynamics of oscillations of many genes – but not because mature let-7s microRNAs are themselves oscillating, but because, as repressors, these microRNAs can affect the relationship between mRNA dynamics and protein dynamics for their targets.

---

## [Author Response]

[Editors’ note: the authors resubmitted a revised version of the paper for consideration. What follows is the authors’ response to the first round of review.]

Reviewer #1:In the manuscript "Feedback among a retinoid-related nuclear receptor and let-7 miRNAs controls the pace and number of molts in *C. elegans*", Patel et al. aim to demonstrate a feedback network between retinoid-related nuclear receptor nhr-23 and let-7 miRNA family in the control of the pace/number of molts in C.elegans. First, they perform a quantitative and detailed characterization of the molting pattern throughout development in a number of different genetic manipulations of nhr-23 and/or let-7 family miRNAs and demonstrate that they affect the length/number of the molting cycle. Then, they go on to demonstrate direct regulation between nhr-23 and let-7 family miRNAs. The authors offer too simplistic of a feedback model for something much more complex and what the data suggest (elaborated in more detail below). Much of what the authors convincingly demonstrate in this manuscript, although done in a more detailed/quantitative manner and sometimes using novel reagents, has been previously shown/implicated (Kostrouchova et al., 2001; Frand et al., 2005; Hayes et al., 2006; Celniker et al., 2009). The portion of the manuscript that could be potentially interesting and novel (the mechanisms by which nhr-23/let-7 alter molting cycle length/number, conservation of this regulation in other systems, etc), the authors offer very little insight and experimental support. In its current form, without further experiments/analysis as suggested below, this manuscript does not significantly advance beyond the current understanding on the topic.1) While the authors convincingly demonstrate that individual genetic manipulations of let-7 and nhr-23 oppositely regulate the pace of the molting cycle, it is not clear how feedback/molecular regulation between let-7 and nhr-23 regulate the pace. Further exploration of the mechanism will greatly enhance the novelty and significance of the manuscript.a. Genetically, the authors demonstrate that reducing nhr-23 levels delay the pace of larval development/molting cycle while reducing let-7 levels accelerate the pace of larval development/molting cycle. At the molecular level, nhr-23 transcript level is oscillatory, reaching its peak level in the middle of the interval between molts and reaching its lowest level right before a molt (Kostrouchova et al., 2001; current manuscript). In nhr-23 (aaa20-deltaLCS) or let-7 mutant animals, the authors report elevated peak nhr-23 transcripts level, steeper slopes of rise and wider troughs compared to control. How do (which of) these characteristics contribute to a faster pace? Are there uncharacterized features (ie when in the cycle the peak is reached, time in between peaks, etc) that could help explain its effect on the pace?

The reviewer brings up an interesting point. The amplitudes, rising slopes and peak values of the *nhr-23* waveforms in *nhr-23(aaa20)* and *let-7(n2853)* mutants are consistently higher across the two stages that are sped up and across both replicates, relative to wild-type animals. To determine which aspects of the *nhr-23* curve is responsible for the faster pace of development of *nhr-23(aaa20)* and *let-7(n2853)* mutants, relative to wild type, we would have to develop a system wherein we could alter one parameter (e.g. amplitude) without changing the other two parameters. Such a system is beyond the scope of the current manuscript and has yet to be developed even in the much more mature field of chronobiology. Moreover, our approach is consistent with approaches used in the field of chronobiology. Studies of the role of core clock genes on rhythms such as sleep-wake and feeding-fasting cycles are routinely conducted by knockdown or deletion of individual genes or *cis*-regulatory elements and documentation of ensuing behaviors (For example, Brown et al., 2005; Konopka and Benzer, 1971; Zheng et al., 1999).

It seems that the slope of the fall is also steeper in the mutant animals compared to control. If so after quantification, how can the faster degradation of nhr-23 be explained in let-7 mutant animals?

We thank the reviewer for this comment. As suggested, we have now quantified the slope of decay of *nhr-23* transcripts. The slope of decay of *nhr-23* transcripts is consistently higher in *let-7(n2853)* mutants than wild type, across both life stages and both biological replicates (p<0.05, one tailed paired t test) (Supplemental Table 3). However, the trend is not clear in *nhr-23(aaa20)* mutants, relative to wild type. The faster decay of *nhr-23* transcripts in *let-7(n2853)* mutants suggests that a different LCS independent mechanism regulates *nhr-23* levels. A different target of *let-7, ­*like *nhr-25* or *lin-42,* may also drive expression and maintain levels of *nhr-23*. Thus, we suggest that the negative side of the feedback loop in this paper prevents faster rise or higher expression of *nhr-23* in the next life stage, rather than the rate of decay in the current life stage.

b. More optimally, the questions raised above could be addressed by examining downstream molting-related genes. The authors examine 1 such genes (mlt-10) only in mutant animals with elevated nhr-23 levels. The authors should examine additional genes important in each phase of the molting cycle as outlined by the authors in Figure 9D in genetic manipulation of both increased/decreased nhr-23 levels. Similar examination should be done with lin-42. Together, these could provide a satisfactory answer as to the mechanism by which the pace of the developmental/molting cycle are regulated by genetic manipulation of nhr-23 and let-7. Additionally, some of these molting genes should be examined in supernumerary molts in the adult. These could further strengthen any conclusion made about these genes in the regulation of molt timing.

We agree and thank the reviewer for these comments. We have now measured the levels of *fbn-1* (Figure 8B and Figure 8 —figure supplement 1A) and *lin-42* (Figure 9C and Figure 9 —figure supplement 1C) transcripts in *nhr-23(aaa20)* and *let-7(n2853)* mutants and age-matched wild-type animals. We have also measured the levels of the same transcripts in *nhr-23(RNAi)* and mock-treated animals (Figure 8A, Figure 9B). We tried to measure the levels of *osm-11* by Taqman RT-qPCR, but we did not detect cyclical expression of *osm-11* as has been reported in past RNA-Seq data sets. This may be because *osm-11* is expressed at relatively low levels and the assay is not sensitive enough to detect the variations. Additionally, we have revised our manuscript to remove what used to be Figure 9D, since we have little data to support the model at this point.

2) Based upon previous reports and the data presented in this manuscript, they suggest a much more complex regulation than the simple model proposed. Further experimentation and discussion are necessary in the manuscript to address these additional relevant/related regulations not incorporated in the current model.a. Based upon previous reports (Hayes et al., 2006, Hada et al., 2010) that both nhr-23 and nhr-25 are downstream of let-7 family miRNA regulation of molts, it is important to also examine the role of nhr-25 in the study and how it fits into the model.

While both *nhr-23* and *nhr-25* regulate the molting cycle, we chose to investigate the interaction between NHR-23 and *let-7* in great detail. However, to answer the reviewer’s question, there is no binding peak (Celniker et al., 2009) or consensus binding site for NHR-25/FTF (TCAAGGTCA) (Galarneau et al., 1996) in the minimal promoter element of *let-7*. Therefore, it is unlikely that NHR-25 binds upstream of *let-7*. There are 3 NHR-25 binding peaks upstream of *nhr-23* (Celniker et al., 2009)*,* but only one peak contains a single NHR-25/FTF consensus binding site. NHR-25, could therefore, have some role in activating *nhr-23* gene expression.

As per our analysis in Figure 8 and Supplemental Table 4, *nhr-25* may be a shared target of NHR-23 and *let-7* microRNAs. The promoter of *nhr-25* has 3 RORE sites (Supplemental Table 4) and is enriched in the modENCODE NHR-23 ChIP-seq (Celniker et al., 2009). The 3’ UTR of *nhr-25* harbors a LCS in its 3’UTR and is sensitive to levels of *let-7* (Hayes et al., 2006)*.* We have included a new discussion of the above in the text.

b. Based on a previous report (Hayes et al., 2006), nhr-23 GFP reporter without 3'UTR also showed de-repression in let-7 mutant. This suggests that regulation of nhr-23 by let-7 is not simply direct at the level of the 3'UTR. The author should consider experiments examining the effect of known downstream heterochronic regulators of let-7 (ie lin-41, lin-29 etc) on nhr-23 or at least address this point in the discussion of the proposed model.

We thank the reviewer for bringing up this point. In the Results section of the previous submission, we had mentioned that the *wgIs43[nhr-23::GFP]* reporter that we use includes the *nhr-23* 3’ UTR, unlike the reporter used in Hayes *et al.,* 2006. In this submission, we have included a new paragraph in the Discussion section elaborating on both the 3’ UTR dependent and 3’ UTR independent regulation of *nhr-23* by *let-7*. Notably, as shown in Figure 6A and Figure 6 —figure supplement 1A, the levels of *nhr-23* at adulthood in *let-7(n2853)* is more elevated than in *nhr-23(aaa20-∆LCS).* This suggests that *let-7* may be regulating *nhr-23* by more than one pathway. *let-7* represses the expression of the RNA binding protein *lin-41,* and this promotes the translation of the LIN-41 target, *lin-29* (Reinhart et al., 2000; Slack et al., 2000)*. nhr-23* expression is elevated in *lin-29* adults and it has been suggested that LIN-29 represses the transcription of *nhr-23* (Harris and Horvitz, 2011). Therefore, the LIN-29 mediated repression of *nhr-23* could represent the 3’ UTR independent mechanism by which *let-7* regulates *nhr-23*. However, the elucidation of these additional pathways are beyond the scope of this paper.

c. In Fig3F, nhr-23 RNAi does not significantly (statistics?) dampen mature let-7 expression until L4/adult stage. Is there stage specificity in terms of nhr-23 regulation of let-7? Additionally, lin-42 (statistics?) also seems to be only significantly regulated by nhr-23 in L4 (Supp Figure 5B). Does this hold any significance? Further discussion of this should be included.

We appreciate the reviewer for this observation. We now provide evidence that *nhr-23(RNAi)* is associated with significantly lower levels of mature *let-7* in both the L3 and L4 stages, i.e., in all stages that mature *let-7* is detectable (Figure 3D-G). We also show evidence that *lin-42* is significantly downregulated in *nhr-23(RNAi)* L3 and L4 stage larvae, relative to mock-treated animals (Figure 9B and Figure 9 —figure supplement 1C).

3) In some cases, there are suggestions of some conclusion but these were not explicitly demonstrated. The authors should provide more experimental evidence in these cases.a. It is suggested that elevated/continuous cyclic expression of nhr-23 beyond the L4/YA molt is responsible for the supernumerary molts, however this correlation was not explicitly demonstrated. At the very least, the authors should take the nhr-23::GFP overexpression strain (wgIs43) and demonstrate that there is cycling of GFP signal that precedes each supernumerary molt.

We show the following data in support of the hypothesis that continued expression of *nhr-23* is associated with supernumerary molts: (1) We have found evidence that *nhr-23* transcripts peak again in *nhr-23(aaa20-∆LCS)* adults, but not in wild-type adults (Figure 6A). (2) We show that *nhr-23(aaa20-∆LCS)* adults go through extra molts (Figure 7C). We did not track the expression of NHR-23::GFP in *nhr-23(wgIs43)* as it is hard to predict exactly when adults undergo supernumerary molts and this makes reproducible meaningful comparisons very challenging. Additionally, as mutants that undergo aberrant supernumerary molts die, analysis of cycling of NHR-23::GFP is not feasible.

b Conservation of the cross regulation between nhr-23 and let-7 is suggested by the author through bioinformatics analysis. This is optimally tested in the respective system if technical capabilities allow, but at the very least, the authors should demonstrate that these promoters/3' UTR sequences from other systems can be regulated in the predictive manner in *C. elegans* when transgenically expressed. Such demonstration would add to the novelty of the manuscript.

We thank the reviewer for this comment. We have added new data to show potential conservation of the cross regulation. Specifically, we use heterologous expression systems to show that the *M. musculus* let-7a promoter is downregulated in *nhr-23(RNAi)* larvae, relative to control larvae. Additionally, we show that fragments of the 3' UTRs of *H. sapiens RORB* and *M. musculus* RORA to downregulate the expression of TdTomato in the bicistronic reporter system, in an LCS-dependent manner.

c. They authors demonstrate nhr-23 binding at the promoter of other let-7 family miRNA (mir-48, 84, 241). Are their levels regulated by nhr-23?

We thank the reviewer for this suggestion. We performed RT-qPCR to measure the levels of the primary transcripts of *mir-48, mir-84* and *mir-241* (Figure 3 —figure supplement 2C) during the L3 stage in control and *nhr-23(RNAi)* animals. As expected, the levels of *pri-mir-48, pri-mir-84* and *pri-mir-241* exhibit one peak during the L3 stage in the control sample. The peak for all 3 microRNAs were attenuated in the *nhr-23(RNAi)* sample, suggesting that NHR-23 promotes transcription of *pri-mir-48, pri-mir-84* and *pri-mir-241*. We also measured the levels of mature *mir-48, mir-84* and *mir-241* during the same stage (Figure 3 —figure supplement 2D).

d. It is suggested that nhr-23 auto-regulates itself due to the presence of ROR binding site in its promoter. This should be tested by the in vivo deletion of this site via CRISPR/Cas-9 and examine the resulting nhr-23 expression.

We agree that experimental evidence for the autoregulation of *nhr-23* was required. However, the in vivo deletion of the 8 RORE sites upstream of *nhr-23* (Figure 9 —figure supplement 1C) by CRISPR/Cas9 is experimentally challenging. Therefore, we devised two alternate strategies to test the autoregulation of *nhr-23*. We performed ChIP-qPCR during L3 and validated that the *nhr-23* promoter was enriched in NHR-23::3xFLAG IPs (Figure 9A). We also measured the fluorescence in the hyp and seam cell nuclei of transgenic *nhr-23::gfp::unc-54 3’ UTR* reporter line in control and *nhr-23(RNAi)* conditions (Figure 9D-F). The *nhr-23::gfp::unc-54 3’ UTR* reporter does not contain the last two and half exons of endogenous *nhr-23,* and hence should be unresponsive to the RNAi clone. The NHR-23::GFP fluorescence was reduced in both hyp and seam cell nuclei under *nhr-23* RNAi conditions.

4) Statistics are not provided in many places in the manuscript/figures. Sometimes even error bars are not apparently in figures. Is there only a single replicate in these cases? The below is not an exhaustive list, the author should go through to make sure every numerical figures have been properly analyzed and provide detail statistical information for other figures as they have done in Supplementary Table 1: Figure 3E/F, Figure 5A, Supp Figure 3, Figure 7B, Supp Figure 5b.

We thank this reviewer for pointing this out. We now provide statistics for all pairwise or multiple comparisons that were performed in our study.

Reviewer #2:In this manuscript, Patel et al., describe a regulatory loop between a Retinoid-Related Orphan Receptor (ROR) in *C. elegans*, NHR-23, and the let-7 microRNA family. The authors examine lethargus and molting behavior in worms with reduced activity or increased activity of NHR-23 along with worms that have reduced activity of the let-7 family (let-7, mir-48, mir-84, and mir-241). They provide compelling evidence that NHR-23 binds to the let-7 (and other family members to a lesser degree) promoter to promote let-7 transcription and activity. Additionally, they provide evidence that let-7 then functions to act through the nhr-23 3' UTR to repress nhr-23 activity and protein accumulation. In the absence of negative regulation by let-7, nhr-23 transcript and protein levels are misregulated showing elevated levels and a temporal shift in its oscillations. In the absence of let-7 negative regulation or in the presence of overexpression nhr-23, worms show aberrant molting behavior. This is also associated with a shift in mlt-10 expression, which is known regulator of molting, and is a predicted shared target of let-7 and NHR-23. Sequence analysis suggests that this feedback mechanism may be conserved in higher organisms.Overall, the expression data supports a model whereby the feedback loops between nhr-23, let-7, and lin-42/PER can drive the cyclic lethargus and molting behavior. The data and experimental approach are strong and rigorous. However, the authors interpretation and model go beyond what the data can directly support. There are issues throughout the manuscript with the clarity of the writing with instances of vague and confusing language and inconsistent use of terms. For example, at different points in the manuscript, the authors refer to lethargus, sleep phase, and sleep-like phase. The manuscript should be extensively revised to improve the accuracy and clarity of the writing.1. The authors perform qRT-PCR for nhr-23 and mlt-10 transcripts at specific time points in larval development to track the oscillations in expression. While the oscillations are clear as are the changes in the dynamics/shape of the oscillations in mutant backgrounds, the calculation of amplitude and "rising slope" is not well supported. While such numbers seem helpful in generally describing the overall dynamics, it seems misleading to provide such seemingly precise quantification of the comparisons of expression at different time points. It appears that the data shown is from one biological replicate performed with technical duplicate data points (at least this is my interpretation of what was done as described in the figure legend). qRT-PCR is typically a very sensitive assay with a degree of variability between biological replicates. Thus, the calculated amplitude and rising slope may vary significantly between experiments. This is observed in the authors' data (Figure 5 and Supplemental Figure 3). If there are references to support such calculation of amplitude and rising slope, then these should be provided. It is my opinion that more general, qualitative descriptions of the oscillations in transcript levels would be more appropriate.

Studying oscillatory gene expression systems that regulate developmental rhythms poses a great challenge. We have shown that genetic manipulations of *let-7* and *nhr-23* change the pace of development. While RT-qPCR provides enough sensitivity to detect the differences in the levels of gene expression, comparing the values across genotypes in a meaningful way is not trivial (Tsiairis and Großhans, 2021). Metacycle allows us to calculate the following metrics that are directly comparable: slopes of accumulation and decay, peak values, amplitude, and phase of expression of the transcripts in each life stage. Such metrics are routinely calculated in studies of biological oscillators and associated rhythms and thus, we include them in the paper (Novák and Tyson, 2008). We have also edited the Results sections to include qualitative descriptions, as suggested by the reviewer.

In the RT-qPCR charts shown in Figures 6, 8 and 9, the symbols and error bars represent the mean and range from three technical replicates in one biological replicate. The related figure supplements show similar charts for an independent biological replicate. These details are described in the associated figure legends.

2. The description of how CCGs were identified and selected should be described more clearly (line 526). In this section, the sentences in lines 546-548 and line 550-551 are also very unclear.

We thank the reviewer for this comment, and we have made appropriate changes to the above mentioned sections.

3. In the Discussion, the authors describe a detailed model of sequential activation of genes involved in cuticle biogenesis and lethargus as regulated by NHR-23 and let-7. While this is a compelling model, the details, including that NHR-23 "schedules" the start of cuticle biogenesis and that the sequential activation is driven by the number of ROREs and the amount of NHR-23 remain quite speculative. The authors should more clearly write this section as a theoretical model. The description of NHR-23 and let-7 controlling the rate of transcript accumulation and decay along with the "evidently short half-life of NHR-23 protein is not sufficiently supported by their data. The authors should also explain what is meant by the biogenesis of the 'sheath'.

We thank the reviewer for this comment. We have updated the manuscript with a description of the sheath. We have also revised the Discussion section to more clearly state that the current model is theoretical and have called out sections of the model that are supported by data in this paper.

4. The observations that ROREs were found upstream of let-7 genes in other organisms and that LCSs were found in other nhr-23/ROR homologs is compelling but doesn't seem sufficient to propose that this regulation is conserved in human tissues (lines 634-636).

As mentioned above, we now show in Figure 10 that the GFP reporter driven by the *M. musculus* let-7a promoter is downregulated in *nhr-23(RNAi)* larvae, relative to control larvae. Additionally, we now show that fragments of the 3' UTRs of *H. sapiens RORB* and *M. musculus* RORA downregulate the expression of TdTomato in the bicistronic reporter system, in an LCS-dependent manner. This strengthens the argument that the feedback loop may be conserved in mammals.

References:

Brown, S.A., Ripperger, J., Kadener, S., Fleury-Olela, F., Vilbois, F., Rosbash, M., and Schibler, U. (2005). PERIOD1-Associated Proteins Modulate the Negative Limb of the Mammalian Circadian Oscilllator. Science *308*, 691–693.

Celniker, S.E., Dillon, L.A.L., Gerstein, M., Gunsalus, K.C., Henikoff, S., Karpen, G.H., Kellis, M., Lai, E.C., Lieb, J.D., MacAlpine, D.M., et al. (2009). Unlocking the secrets of the genome. Nature 927–930.

Galarneau, L., Paré, J.F., Allard, D., Hamel, D., Levesque, L., Tugwood, J.D., Green, S., and Bélanger, L. (1996). The alpha1-fetoprotein locus is activated by a nuclear receptor of the *Drosophila* FTZ-F1 family. Molecular and Cellular Biology *16*, 3853–3865.

Harris, D.T., and Horvitz, H.R. (2011). MAB-10/NAB acts with LIN-29/EGR to regulate terminal differentiation and the transition from larva to adult in *C. elegans*. Development *138*, 4051–4062.

Hayes, G.D., Frand, A.R., and Ruvkun, G. (2006). The mir-84 and let-7 paralogous microRNA genes of *Caenorhabditis elegans* direct the cessation of molting via the conserved nuclear hormone receptors NHR-23 and NHR-25. Development *133*, 4631–4641.

Konopka, R.J., and Benzer, S. (1971). Clock mutants of *Drosophila melanogaster*. Proc Natl Acad Sci USA *68*, 2112–2116.

Novák, B., and Tyson, J.J. (2008). Design principles of biochemical oscillators. Nature Reviews Molecular Cell Biology *9*, 981–991.

Reinhart, B.J., Slack, F.J., Basson, M., Pasquinelli, A.E., Bettinger, J.C., Rougvie, A.E., Horvitz, H.R., and Ruvkun, G. (2000). The 21-nucleotide let-7 RNA regulates developmental timing in *Caenorhabditis elegans*. Nature *403*, 901–906.

Slack, F.J., Basson, M., Liu, Z., Ambros, V., Horvitz, H.R., and Ruvkun, G. (2000). The lin-41 RBCC gene acts in the *C. elegans* heterochronic pathway between the let-7 regulatory RNA and the LIN-29 transcription factor. Molecular Cell *5*, 659–669.

Tsiairis, C., and Großhans, H. (2021). Gene expression oscillations in *C. elegans* underlie a new developmental clock. Curr Top Dev Biol *144*, 19–43.

Zheng, B., Larkin, D.W., Albrecht, U., Sun, Z.S., Sage, M., Eichele, G., Lee, C.C., and Bradley, A. (1999). The mPer2 gene encodes a functional component of the mammalian circadian clock. Nature *400*, 169–173.

[Editors’ note: what follows is the authors’ response to the second round of review.]

The manuscript has been improved but there are some remaining issues that need to be addressed, as outlined below:Everyone agrees that this paper is of interest for eLife, but there is also a broad consensus that additional revisions are required that relate to tampering of claims and various other editorial revisions. No further experimentation is required. You will see all the requested detailed below by the reviewers.Reviewer #3 (Recommendations for the authors):Patel et al. have significantly improved the manuscript over the last 2 years. They have convincingly demonstrated the cross regulation between nhr-23 and let-7 using a combination of biochemical and functional approaches. They have further strengthened their claims of a possible conservation of the let-7/nhr-23 cross regulation in other species (mouse and human) with additional experiments. However, the paper as presented currently, still contains the following major weaknesses:1) Many claims made in the manuscript were broad and generalized when they were only supported in specific instances or often contrary to data presented. A non-exhaustive list is included below. The authors should go through the manuscript to make sure all of their claims are specific/precise and are supported by the data presented, and adjust conclusions accordingly:a. Line 215-216: "the cohort of nhr-23(RNAi) animals that emerged as L4 entered lethargus later". This is unsupported in Figure 2B as well as Supplementary Table 1. The mean times to enter lethargy were 8.1 and 8.4 hours for control and nhr-23(RNAi) animals respectively, and are not statistically different from one another.

We thank the reviewer for this comment and have made appropriate changes to the text. The relatively delayed entry of the *nhr-23(RNAi)* cohort, as compared with the control cohort, is more obvious when comparing the times that individual worms within each cohort entered lethargus, rather than comparing the average time differences between the active intervals of *nhr-23(RNAi)* and wild-type larvae. Thus, we now more precisely state these results as follows in the text:

“Approximately 50% (8 out 17 animals) of the cohort of *nhr-23(RNAi)* animals entered lethargus 9–10 h after emerging as L4s as compared with 19% of the control cohort (3 out of 16 animals). Strikingly, the cohort of *nhr-23(RNAi)* animals remained lethargic for twice as long as the control cohort.” (lines 222-225)

b. Line 225-226: "delayed and protracted lethargi were associated with nhr-23 across three larval stages." According to Figure 2 and Supplementary table 3, while the time spent in lethargy was prolonged across all three stages, the delayed lethargi (time in active) was only statistically different in L3.

We have also changed the concluding remarks in this section to emphasize the protracted lethargus and de-emphasize the delay in lethargus. However, we would like to point out again that, although the differences in the mean times to enter lethargy are not significantly different between the *nhr-23(RNAi)* and control cohorts for stages besides the L3 stage, we did observe differences when comparing the proportion of individual larvae that enter lethargus at a particular time (see response to comment 1a above). Accordingly, we have modified the text to read as follows:

“Thus, *nhr-23(RNAi)* animals developing through three larval stages entered lethargus slightly later and remained lethargic for much longer than wild-type animals.” (lines 233-235)

c. Line 319-321: "the level of other members of the let-7 family, mir-48, mir-84, mir-241 were similarly reduced in nhr-23 (RNAi) larvae developing across the L3 stage." The levels of mature mir-84 and mir-241 were not significantly altered according to Figure 3 supplement 2D

We thank the reviewer for this comment. We have made textual changes (line 343) to emphasize that only the primary transcripts of miR-48, miR-84 and miR-241 were significantly reduced in the developing *nhr-23(RNAi)* larvae. However, the observation that mature levels of miR-48, miR-84 and miR-241 did not show significant change, despite changes in primary transcript levels, is accurate and may be explained by the precedent that only a fraction of primary and precursor *let-7* is processed to form mature *let-7* (Bracht et al., 2004). Similarly, it is possible that the reduced levels of primary miR-48, miR-84 and miR-241 under *nhr-23(RNAi)* conditions is still sufficient to produce wild type levels of mature microRNAs*.*

d. Line 449-450: The authors claimed that nhr-23 levels in L2 peaked at higher levels in mir-48, mir-241, mir-84 triple mutant compared to wild type controls. This statement was not supported by data in Figure 6 supplement 1B/B': mir-48, mir-241, mir-84 triple mutant only exhibited higher nhr-23 level in L3 but not L2.

We thank the reviewer for pointing this out and have changed the related text to more accurately describe the data (lines 474-480). Only the rising slope of the *nhr-23* waveform was 3-fold higher during the L2 stage in *mir-48 mir-241 mir-84* triple mutants, as compared with the controls. Additionally, the phase of the *nhr-23* waveform was slightly advanced in *mir-48 mir-241 mir-84* triple mutants relative to the controls (Supplemental Table 3).

e. Line 556: The authors claimed similar slope of fbn-1 waveform between nhr-23 RNAi and mock-treated larvae, but this was not supported by data in Supplementary Table 3.

We have corrected this error (lines 586-588).

2) I applaud the authors for including replication of their experiment. However, as related to point 1, the authors often claimed that the results of the second replicates supported those of the first replicates when they did not (Some specific examples included below). The authors should address this in their manuscript why these replicates yielded inconsistent results and how to interpret their conclusions in light of this.a. As opposed to what the authors claimed in line 564-566, independent replicates of fbn-1 and mlt-10 transcripts in mock vs nhr-23 (RNAi) animals were not reproducible across all developmental stages (see Figure 8A vs Figure 8 Supplement 1A, Figure 8C vs Figure 8 F).

The charts showing the levels of *fbn-1* and *mlt-10* transcripts in *nhr-23(RNAi)* and mock-treated animals (Figures 8A and 8C) include data from two independent replicates. The dots and error bars represent the mean and range from two biological replicates.

If the reviewer is referring to the independent replicates of *fbn-1* and *mlt-10* transcripts in wild type, *let-7(n2853)* and *nhr-23(aaa20)* mutants (Figure 8B vs Figure 8 Supplement 1A and Figure 8C vs Figure 8 Supplement 1B), then we have changed the text to be precise about the metrics that were reproducible in both stages and both biological replicates. Specifically, we now refer to the time at which the peak values of *fbn-1* expression and *mlt-10* expression were reached in the respective strains. While the value for the phase of gene expression calculated by Metacycle (Supplemental Table 3) may not have been the same across biological replicates, the trends remain the same. Accordingly the text now reads as follows:

“Peak values of *fbn-1* expression were detected slightly earlier in both *nhr-23(∆LCS)* and *let-7(n2853)* mutants developing through the L4 stage, relative to control animals (Figure 8B); this finding was replicated in a second, independent trial (Figure 8 —figure supplement 1A).” (lines 592-595)

“Again, peaks in *mlt-10* expression were detected slightly earlier during the L4 stage, in both mutants relative to wild-type animals, in two independent trials (Figure 8D and Figure 8 —figure supplement 1B).” (lines 603-605)

b. Similarly, as opposed to what the authors claimed in line 648-649, independent replicates of lin-42 transcript levels in nhr-23 (LCS deletion) and let-7(n2853) animals did not yield the same results (see Figure 9B vs Figure 9 Supplement 1C). It seems that for the second replicate, the stages were not aligned as they were in the first replicate. If they were aligned in phase as they were for the first replicate, there were be no significant difference between the three experimental groups.

We have now specifically stated the differences that were observed in both biological replicates in the text. Additionally, we made an error in the placement of the box indicating the L3-to-L4 and L4-to-Adult lethargi for the wild-type samples in the second biological replicate (in Figure 9 Supplement 1C). We have fixed the error and it can now be seen that the developmental stages of the different genotypes are aligned. Regardless, as above, we are now more specific about the metrics that are reproducible across biological replicates. We also emphasize the time at which the peak value of gene expression is achieved, rather than looking at the absolute value of the phase calculated by Metacycle. The text has been changed as follows:

“The peak levels of *lin-42* expression were 1.5-fold higher in both *let-7(n2853)* and *nhr-23(∆LCS)* mutants, in the L3 stage, relative to the age-matched control animals (Figure 9C). We also detected earlier peaks in *lin-42* expression in both *nhr­-23(∆LCS)* and *let-7(n2853)* mutants developing through the L4 stage, across two independent replicates (Figure 9C, Figure 9 —figure supplement 1C, Supplemental Table 3).” (lines 674-679)

3) The mechanism by which let-7 and nhr-23 control the pacing of the molting cycle is still under-addressed in the paper. The authors have addressed this partially by some additional data (regulation of lin-42 and fbn-1) as well as discussion of additional complexities beyond let-7/nhr-23. While some aspects of the mechanisms might be outside the scope of the current manuscript and not possible to address experimentally, one key aspect that is not currently addressed in the paper is how different manipulation of let-7 and nhr-23 affects different phases of the molting cycle. As one example, nhr-23 RNAi increases time spent in lethargy but does not affect other parts of the molting cycle, while increasing nhr-23 levels (overexpression or deletion of let-7 LCS sites) decreases time spent in active cycle but does not affect time spent in lethargy. Inclusion of the discussion of these points would be important to help the readers think about the complexity of the regulation, and important to consider in future studies to understand the mechanisms by which these regulators control molting cycle pacing.

We thank the reviewer for this comment. However, *nhr-23(RNAi)* and *let-7* mutations do alter the time spent in the active and lethargic intervals, respectively. As stated earlier in our responses to comments 1a and 1b, the prolonged active interval of *nhr-23(RNAi)* larvae in the L4 stage is more obvious when comparing the time that individual *nhr-23(RNAi)* and mock-treated larvae entered the L4-to-Adult lethargus (Figure 2A and lines 222-225). The average active interval of the L3 stage *nhr-23(RNAi)* cohort is significantly longer than the mock-treated cohort (Supplemental Table 1). The average time that *let-7(n2853)* mutants spend in the L4 lethargus is significantly lower than wild type (Supplemental Table 1). Additionally, 72% of *let-7(n2853)* spent only 1 h in the L3-to-L4 lethargus, as opposed to 40% of the control cohort (Figure 2C). Importantly, these phenotypes were mutually suppressed in the *nhr-23(RNAi) let-7(n2853)* double mutant (Figure 2B and 2C and lines 252-272). Our new figure (Figure 11) and lines 835-873 in the Discussion section both summarize our current model of the mechanism by which *nhr-23* and the *let-7s* control the speed of the molting cycle. As suggested by the reviewer, we have also included the following section section describing future experiments that could address the mechanism by which NHR-23 and let-7s regulate specific aspects of the molting cycle:

“Future experiments such as comprehensive analysis of new RNA-seq datasets generated from *nhr-23(∆LCS)*, *let-7(scRORE)*, and *let-7* family mutants employing the bioinformatic strategies presented in this study would be exciting, as it would allow for the identification of specific clock-controlled genes that drive the transitions between different phases of molting among the ~3700 genes that cycle in expression throughout larval development (Hendriks et al., 2014; Kim et al., 2013; Meeuse et al., 2020).” (lines 878-883).

On a related note, and as alluded by Reviewer 2 previously, the authors use inconsistent terms throughout the manuscript to refer to different aspects of the molting cycle in the manuscript. Standardization of the terms throughout the manuscript would help to increase clarity.

We agree and have better defined terms and standardized their use for clarity.

4) In supplementary Table 1, only statistics for comparisons to control groups were presented. It would be helpful for the readers to present statistics for comparisons between experimental groups as well.

We have included the pairwise statistics for the wake-to-wake interval in the Author response table 1 . However, we chose to only present the statistics for comparison to control groups in the manuscript (Supplemental Table 1) for simplicity and because we believe that those are the biologically relevant comparisons.

**Author response table 1. sa2table1:** Multiple comparison of p values (one-way ANOVA with Bonferroni correction) for wake-to-wake intervals between the genotypes/conditions in the row versus column.

L4 (Figure 2B)	wild type (N2)	wild type, *nhr-23* RNAi	*let-7(n2853)*	
wild type, *nhr-23* RNAi	****			
*let-7(n2853)*	****	****		
*let-7(n2853), nhr-23* RNAi	n.s.	****	****	
				
L4 (Figure 4F)	wild type (N2)	*let-7 (xk41-scRORE1,2)*	*let-7 (xk44-scRORE1,2)*	*let-7 (xk39-scRORE1,3)*
*let-7(xk41-scRORE1,2)*	***			
*let-7(xk44-scRORE1,2)*	*	n.s.		
*let-7(xk39-scRORE1,3)*	***	n.s.	n.s.	
*let-7(xk42-scRORE1,3)*	***	n.s.	n.s.	n.s.
				
L4 (Figure 6B)	wild type (N2)	*wgIs43[nhr-23++]*		
				
*wgIs4 [nhr-23++]*	**			
*nhr-23*(*aaa20-∆LCS*)	****	n.s.		
				
L3 (Figure 2C)	wild type (N2)	wild type, *nhr-23* RNAi	*let-7(n2853)*	
wild type, *nhr-23* RNAi	******			
*let-7(n2853)*	*	****		
*let-7(n2853), nhr-23* RNAi	******	****	****	
				
L3 (Figure 6C)	wild type (N2)	*wgIs43[nhr-23++]*		
*wgIs43 [nhr-23++]*	******			
*nhr-23* (*aaa20-∆LCS*)	********	*		
				
L2	wild type (N2)	wild type, *nhr-23* RNAi	*mir-48 mir-241(nDf51); mir-84(n4037)*	
wild type, *nhr-23* RNAi	****			
*mir-48 mir-241(nDf51); mir-84(n4037)*	n.s.	****		
*mir-48 mir-241(nDf51); mir-84(n4037),**nhr-23* RNAi	****	n.s.	****	

Reviewer #4 (Recommendations for the authors):Patel et al. present data that provide new and substantive understanding of the roles of heterochronic genes and the NHR-23 nuclear receptor transcription factor in the regulation of the temporal dynamics of molting behavior in *C. elegans* larval development. The findings provide a solid basis for the proposal that direct regulatory feedback between let-7 family microRNAs and nhr-23 contributes to the regulation of developmental pace. Novel findings include that NHR-23 promotes the oscillatory transcription of let-7s genes at each larval stage – likely by direct binding to their promotors – and that let-7s microRNAs inhibit NHR-23 expression via complementary LCE sequences in the nhr-23 3' UTR. Key supportive evidence includes the finding nhr-23(gf) mutations accelerate the pace of the L4, as do let-7s(lf) mutations, supporting a dosage-dependent role for NHR-23 in promoting larval stage progression, downstream of let-7s microRNAs. Intriguing evidence is presented for possible conservation in vertebrates of the mutual regulation of ROR/NHR-23 and let-7 family microRNAs.Overall, the data robustly support novel findings that merit publication in eLife. However, some of the conclusions and data interpretations, as stated in the current manuscript, are not supported, and therefore revisions are recommended, according to the comments below.1) Lines 164-166: "The cyclical expression profile of primary let-7 family transcripts is consistent with temporally reiterated, as well as stage-specific, function(s)." Do the authors mean to propose here that the let-7 family primary transcripts are functional, aside from being processed into mature let-7 microRNAs? (See points #7 – #10 below).

We did mean to suggest that primary *let-7* transcripts might have a function besides being processed into mature microRNAs. However, this is speculative and not supported by data, and therefore, we have removed this statement (line 169).

2) Line 215: "The cohort of nhr-23(RNAi) animals that emerged as L4s entered lethargus later and remained lethargic for twice as long as the control cohort." It appears that nhr-23 RNAi animals entered L4 lethargus only very slightly later than wt. If the authors wish to argue that the length of the active phase of the larval stage is lengthened in the nhr-23(RNAi) animals, statistical analysis of the length measurements should be shown, to demonstrate significant differences.

We thank the reviewer for this comment. As stated in our response to Reviewer #3 comment 1a, the relatively delayed entry of the *nhr-23(RNAi)* cohort, as compared with the control cohort, is more obvious when comparing the times that individual worms of each cohort entered lethargus, rather than comparing the average time differences between the active intervals of *nhr-23(RNAi)* and wild-type larvae (lines 222-225).

3) Line 248: "When we combined stage-specific nhr-23(RNAi) with let-7(n2853), the altered pace of molting associated with each single mutant was partially co-suppressed (Figure 2B, C and Supplemental Table 1)." I think that a simpler interpretation of the data would be co-expressed phenotypes (rather than co-suppressed). The active phase of the L4 stage (panel B) and L3 stage (panel C) of let7(lf) and let-7(lf);nhr-23(RNAi) are shorter than the wild type; this would correspond to the let-7(lf) phenotype expressed in both cases. At the same time, the lethargus is lengthened in let-7(lf);nhr-23(RNAi); which would appear to be the nhr-23(RNAi) phenotype. Co-suppression would be an apt interpretation if, in the let-7(lf);nhr-23(RNAi) animals the active phases were (statistically) significantly longer than for let-7(lf) alone, and the length of lethargus were (statistically) significantly shorter than for nhr-23(RNAi) alone (see also point #2 above).

We appreciate the reviewer’s comment and agree that “cosuppression” is not the appropriate term. We now simply state that the phenotypes of single mutants are suppressed in the *let-7(lf);nhr-23(RNAi)* double mutant.

Additionally, some of the behavioral phenotypes, as well as their suppression in the double mutant, become more obvious when comparing the times that individual worms of each cohort entered and exited lethargus, rather than comparing the average active and lethargic intervals. As stated in our response above to Reviewer #3, comment 3, *nhr-23(RNAi)* and *let-7* mutations alter the time spent in the active and lethargic intervals, respectively. The prolonged active interval of *nhr-23(RNAi)* larvae in the L4 stage is more obvious when comparing the time that individual *nhr-23(RNAi)* and mock-treated larvae entered the L4-to-adult lethargus (lines 222-225). Also, the average active interval of the L3 stage *nhr-23(RNAi)* cohort is significantly longer than the mock-treated cohort (Supplemental Table 1). Similarly, the average time that *let-7(n2853)* mutants spend in the L4 lethargus is significantly lower than wild type (Supplemental Table 1). Furthermore, 72% of *let-7(n2853)* spent only 1 h in the L3-to-L4 lethargus, as opposed to 40% of the control cohort (Figure 2C). These phenotypes were mutually suppressed in *nhr-23(RNAi) let-7(n2853)* mutants (lines 258-272).

4) Line 265: "Taken together, these longitudinal data suggest a model whereby NHR-23 accelerates the molting cycle, partly by directly activating the expression of the let-7s, and the let-7s decelerate the cycle, partly by directly repressing the expression of nhr-23." At this point in the narrative, it seems premature to propose a specific molecular model, because the data presented so far do not address mechanism. More appropriate at this stage would be a straightforward interpretation of what the phenotypes suggest about how nhr-23 and let-7 microRNAs impact the molting cycle (independent of underlying mechanism): nhr-23 activity negatively regulates the length of lethargus (and perhaps the active phase as well, pending statistical significance, see , points #2 and #3 above), and let-7 family microRNAs negatively regulate the length of the wake-to-wake interval. From the phenotypes alone, there is no reason to necessarily suggest that nhr-23 and let-7 microRNAs regulate each other. Figure 3 addresses mechanism, so this molecular model should be presented after discussion of the Figure 3 results.

We agree and we have made textual edits to present the logic of our conclusions (lines 280-284).

5) Line 270: "Based on the findings of the longitudinal studies described above, we hypothesized that NHR-23 may directly activate transcription of let-7." The problem with this hypothesis is that it does not seem to logically follow from the longitudinal studies presented in Figure 2; nhr-23 and let-7 microRNAs seem to impact molting cycle in opposite ways (accelerating vs decelerating, respectively), so the simplest hypothesis for a regulation of let-7s by nhr-23 would be repression of let-7s by nhr-23. It would be better to propose (broadly) that let-7s and nhr-23 might mutually regulate each other, and then ask, how? The authors could suggest that the simplest expectation could be mutual negative regulation, and then when the data indicate the opposite -- that nhr-23 positively regulates let-7s – deal with the unexpected by modifying the model.

We appreciate the reviewer’s comment. We have made textual edits to reflect that NHR-23, as a transcription factor, could, in principle, regulate *let-7* transcription either positively or negatively. Our findings then showed that NHR-23 promotes *let-7* transcription.

6) Line 369: "Therefore, scrambling the ROREs is sufficient to increase the speed of development, consistent with our model that NHR-23-mediated activation of the let-7s normally slows the pace of molting." Here is where the issue discussed in point #5 above, comes to a head. In this scenario, where NHR-23 is required for full expression of let-7, one would have expected the nhr-23(lf) phenotype to correspond to that of reduced let-7s – i.e., accelerated development. So, it is apparent that the wild type function of nhr-23 in accelerating the molting cycle cannot be simply via promoting expression of let-7s microRNAs (which themselves seem to decelerate the molting cycle). The authors present data regarding potential regulation of lin-42 by nhr-23, and also potential autoregulation by nhr-23, which should be brought into play in formulating a final model (see point # 10 below).Note: Points #7, #8, and #9 below relate to problematic issues surrounding the proposal that the oscillation of let-7s microRNAs is a mechanistic component of a hypothetical oscillator driving the molting cycle. Point #10 offers comments regarding a modified model to accommodate these and other issues.

We thank the reviewer for the thorough analysis of our model of how NHR-23 and *let-7s* regulate the pace of the molting cycle. We visualize the effect of NHR-23 and *let-7s* on the speed of the molting cycle, keeping in mind that the two molecules act in a feedback loop, rather than in a linear genetic pathway. Additionally, both molecules likely regulate several shared target genes in opposing ways; many shared target genes are probably linked to the molting cycle. Thus, although the levels of *let-7* transcripts would be lower in *nhr-23(RNAi)* animals, so would the levels of several shared target genes because of the decrease in the levels of the transcriptional activator, NHR-23. As such, several processes associated with molting (lethargus, cuticle remodeling, arousal, and ecdysis) are either delayed or might not occur.

On the other hand, in *let-7(n2853)* mutants, the expression of *nhr-23* and shared target genes would be higher than wild-type, possibly because of both increased levels of the NHR-23 transcriptional activator and decreased levels of the *let-7* repressor. As such, the molting cycle is accelerated in *let-7(n2853)* mutants relative to wild-type animals. When *nhr-23(RNAi)* is combined *let-7(n2853)*, one would predict that the expression of *nhr-23* would be lower than in *let-7(n2853)* mutants, but higher than in *nhr-23(RNAi)* single mutants (because of absence of the repressor *let-7*) and as such the expression of shared target genes would also be at an intermediate level between the two single mutants. The pace of the molting cycle of the *nhr-23(RNAi) let-7(n2853)* double mutants would be somewhere in between each single mutant.

We also thank the reviewer for the suggestion to incorporate our data on the regulation of *lin-42* by NHR-23 and *let-7s*, as well as the autoregulation of NHR-23 into the model. We have now incorporated both points in our discussion (lines 835-873), as well as in the new model figures in Figure 11 A and B.

7) Line 611: "These findings suggest that partly interdependent waves in the abundance of NHR-23 and the let-7s sculpt the temporal expression profiles of fbn-1, mlt-10 and possibly many additional effectors of the molting timer." Does the model require that the mature microRNAs oscillate? At the core of the authors' model is the idea that the steepness of accumulation of NHR-23 in each stage sets the timing of the onset and completion of lethargus, and hence the pace of the molting cycle. let-7s, through their LCSs in nhr-23 mRNA, can inhibit the rate of nhr-23 accumulation in each cycle, and thereby regulate pace. let-7s mature microRNAs do not need to oscillate to perform this proposed function in shaping the NHR-23 pulse8) Line 732: "NHR-23 transcriptionally activates let-7 and let-7 post-transcriptionally represses nhr-23 to form a molecular-genetic oscillator." Has it been shown that let-7 mature microRNA oscillates? Figure 3G suggests not. In the authors' model, is it required that mature let-7s microRNAs oscillate?9) Line 784: "Thus, NHR-23 and let-7 form a transcriptional-post-transcriptional negative feedback loop. Within a given larval stage, NHR-23 promotes the expression of the let-7s and the let-7s repress nhr-23. The expression levels of let-7 peak ~1.6 hours after nhr-23, resulting in an intrinsic delay between the accumulation of nhr-23 and that of let-7 (Figure 3D, 3F, 6A). Across development, the let-7s dampen the relative amplitude of nhr-23 expression from one larval stage to the next (Figure 6A, Figure 6 —figure supplement 1A). Furthermore, NHR-23 autoregulates its own expression (Figure 9A, 9D-F). Together, these interconnected feedback loops set up a self-sustained molecular-genetic oscillator that is extinguished in adulthood." The problem is that mature let-7s do not seem to oscillate (e.g., Figure 3G). So, if let-7 and nhr-23 are proposed to be oscillating components of an oscillator, the functional let-7 transcript would need to be the pri-let-7s (not mature let-7s!) How would that work? Could perhaps pri-let-7 bind to nhr-23 protein and somehow influence its transcriptional regulatory activity? This would be an exceedingly interesting scenario, if true, but highly theoretical and arguably beyond the scope of the present paper. Moreover, genetic evidence suggests that the molting cycle proceeds independently of let-7-family function, as animals quadruple-mutant for mir-84, mir-48, mir-241, and let-7 still undergo multiple larval stages (Vadla et al., 2012). [BTW, it is not possible to compare Figure 3D and 3F to 6A and make an assessment of the relative timing of pri-let-7 and nhr-23 accumulation.]

We thank the reviewer for these very astute observations #7–#9. Consistent with previous reports (McCulloch and Rougvie, 2014; Van Wynsberghe et al., 2011), we did not found significant oscillations in the levels of mature *let-7*. Mature *let-7* accumulates starting at early L3 and, most likely, associates with the Argonaute ALG-1. Argonaute proteins can stabilize microRNAs (e.g. Grishok et al., 2001; Winter and Diederichs, 2011) and this may partly explain the lack of oscillations in the level of mature *let-7*. However, as all of our assays were done in whole worms, we cannot rule out the possibility that mature *let-7* levels may oscillate in specific tissues.

As the reviewer points out, mature *let-7* levels need not oscillate to function as an oscillator. As we have shown, *let-7* affects the timing of the rhythmic expression of its targets, *fbn-1* and *mlt-10*. Non oscillatory *bantam* and *miR-279* microRNAs regulate the rhythmic expression of circadian rhythm-associated genes and related periodic behaviors in flies (Kadener et al., 2009; Luo and Sehgal, 2012; Vodala et al., 2012).

We have included a paragraph in the Discussion section to reflect the above ideas (lines 853-863). We have also included some of these ideas in the new Figure 11, as recommended by the reviewer.

We also agree with the reviewer that we cannot compare the timing of expression across different experiments and we have removed the absolute value of the time difference between the peaks. However, as indicated in Author response figure 1, expression of *nhr-23* peaks before *primary let-7* when measured in the same experiment*.* We also cannot rule out that other potential regulatory mechanisms by which NHR-23 and *let-7* interact and as the reviewer points out, it is beyond the scope of the current paper.

**Author response image 1. sa2fig1:** RT-qPCR of *pri-let-7* and *nhr-23* measured in the same experiment in wild type animals indicates that *nhr-23* expression peaks before *pri-let-7*.

(10) A graphic depiction of a revised model consistent with the data would help enormously. (The diagram in Figure 1C is not adequate, as the particular phenotypic consequences are not indicated.) Elements of the model could include the following:(10A) Nhr-23 is oscillatory, and levels of NHR-23 matter, there being successive thresholds for initiation and completion of lethargus, respectively.(10B) NHR-23 controls oscillatory transcription of many genes, including let-7s, and lin-42, and itself (by positive autoregulation), and could be a component of an oscillator driving molting cycles. [Current evidence does not support a role for let-7s microRNAs as clock drivers.](10C) let-7s microRNAs accumulate progressively across the L2-L4 stages and dampen nhr-23 expression in two ways:(i) At each stage, repression by let-7s slows the rise of NHR-23, and hence the let-7s microRNAs exert a slowing effect on larval stage pace by affecting how rapidly NHR-23 levels cross the thresholds referred to in (A) above.(ii) After the L4 stage, repression by let-7s puts NHR-23 levels below the threshold for initiation of molting, satisfying one of the conditions for cessation of molting (the other being expression of LIN-29.)

We thank the reviewer for their thorough analysis of our data and for specific suggestions for a model figure. Based on this reviewer’s suggestions, we have made a new figure (Figure 11), which shows our model of how NHR-23 and *let-7s* control the pace of the molting cycle as well as the total number of molts. Lines 818-878 in our Discussion section provide specific details about the model. We have also stated that LIN-29 is an additional factor, besides *let-7*, that promotes the cessation of molting, as suggested by the reviewer (lines 894-896).

Other notes regarding the model:(10D) The authors propose that the termination of molting after the L4 is a consequence of the progressive dampening of NHR-23 expression across the final larval stages. This is a plausible model, but it is incomplete, as it does not account for previous findings that lin-29 is required for cessation of molting. The authors' finding that nhr-23 LCE mutations result in apparent molting behavior of adults (Figure 7A) suggests that nhr-23 activity could be sufficient for supernumerary molting. That novel finding motivates a model wherein cessation of molting requires two conditions to be filled: upregulation of lin-29 (an indirect consequence of let-7 activity in the L4), and dampening of nhr-23 expression (also via let-7).

We thank the reviewer for this comment and have made the suggested change (lines 899-901).

(10E) As the authors data suggest (e.g., Figure 8) let-7s microRNAs could be shaping the dynamics of oscillations of many genes – but not because mature let-7s microRNAs are themselves oscillating, but because, as repressors, these microRNAs can affect the relationship between mRNA dynamics and protein dynamics for their targets.

As stated in our response to comment 9 above, and as the reviewer points out, mature *let-7* levels need not oscillate to function as an oscillator. As we have shown, *let-7* affects the timing of the rhythmic expression of its targets, *fbn-1* and *mlt-10*. We have included a paragraph in the Discussion section to reflect the above ideas (lines 835-873).